# Versatile nitrate-respiring heterotrophs are previously concealed contributors to sulfur cycle

Bo Shao[1], Yuan-Guo Xie [2], Long Zhang[3,4], Yang Ruan [5], Bin Liang [6], Ruochen Zhang[7], Xijun Xu[1], Wei Wang [1], Zhengda Lin[1], Xuanyuan Pei[8], Xueting Wang[1], Lei Zhao[1], Xu Zhou[6], Xiaohui Wu[9], Defeng Xing [1], Aijie Wang[6], Duu-Jong Lee [10], Nanqi Ren[1], Donald E. Canfield [11], Brian P. Hedlund [12,13], Zheng-Shuang Hua [2] ✉ & Chuan Chen [1] ✉

Heterotrophic denitrifiers play crucial roles in global carbon and nitrogen cycling. However, their inability to oxidize sulfide renders them vulnerable to this toxic molecule, which inhibits the key enzymatic reaction responsible for reducing nitrous oxide ($N_2O$), thereby raising greenhouse gas emissions. Here, we applied microcosm incubations, community-isotope-corrected DNA stable-isotope probing, and metagenomics to characterize a cohort of heterotrophic denitrifiers in estuarine sediments that thrive by coupling sulfur oxidation with denitrification through chemolithoheterotrophic metabolism. Remarkably, ecophysiology experiments from enrichments demonstrate that such heterotrophs expedite denitrification with sulfur acting as alternative electron sources and substantially curtail $N_2O$ emissions in both organic-rich and organic-limited environments. Their flexible, non-sulfur-dependent physiology may confer competitive advantages over conventional heterotrophic denitrifiers in detoxifying sulfide, adapting to organic matter fluctuations, and mitigating greenhouse gas emissions. Our study provides insights into the ecological role of heterotrophic denitrifiers in microbial communities with implications for sulfur cycling and climate change.

Denitrification converts nitrate into dinitrogen gas via nitrite, nitric oxide (NO), and nitrous oxide ($N_2O$). This process has been widely recognized as a primitive energy-conserving metabolism that profoundly influenced the early evolution of life prior to the oxygenation of the atmosphere[1,2]. Denitrification has significant environmental implications, as it helps regulate the nitrogen content in ecosystems and contributes to the nitrogen balance in the biosphere. Within marine and terrestrial ecosystems, denitrification contributes substantially to the loss of bioavailable nitrogen, therefore constraining global primary productivity, soil fertility, and crop yields[3–5]. Incomplete denitrification is one of the two major sources of $N_2O$[6], a potent greenhouse gas that contributes to climate warming. Additionally, $N_2O$ emissions contribute to ozone layer depletion and acid rain formation[7].

In many estuaries, coastal regions, and marine oxygen-deficient zones, the sulfur cycle plays a crucial role in regulating biogeochemical processes. This is primarily due to the dominance of sulfate as a terminal electron acceptor and the subsequent release of hydrogen sulfide ($H_2S$) from benthic sediments[8,9]. Studies have reported that sulfide, as a toxic chelator, strongly inhibits the copper-dependent metalloenzymes nitrite reductase (NirK) and $N_2O$ reductase (NosZ)[10], adversely impacting denitrifying activity and potentially leading to increased $N_2O$ emissions[11–13]. At present, ubiquitous chemolithoautotrophic denitrifiers are recognized as the dominant biological

contributors responsible for the oxidation of sulfur compounds and detoxification of sulfide[14–19]. Some of these chemolithoautotrophs are also involved in dissimilatory nitrate reduction to ammonium (DNRA), thereby supplying ammonium to support microbial growth of indigenous bacteria[20,21]. Most known heterotrophic denitrifiers are unable to detoxify sulfide due to their absence of enzymes necessary for sulfide oxidation[13,22]. However, recent evidence of bacterial isolates suggests that certain chemolithoheterotrophic denitrifiers, designated here as facultative sulfide-oxidizing heterotrophic denitrifiers (F-SOHDs), can oxidize sulfide, allowing them to thrive in sulfidic environments[23,24]. A notable example is the recently identified chemolithoheterotroph *Arcobacter peruensis*[23], which has been shown to play a significant role during sulfidic events off the coast of Peru. Despite the energetic advantage and detoxification potential of F-SOHDs, their presence and ecological roles in other natural ecosystems remain poorly understood. Thus, questions remain regarding how heterotrophic denitrifier consortia adapt to sulfide stress within complex microbial communities and their ecological impact on regulation of $N_2O$ emissions.

In this study, we investigate the response of heterotrophic denitrifiers to sulfide in the highly productive Songhua River Estuary, a major tributary within the largest water system in East Asia. By employing a combination of microcosm and lab culture experiments, metagenomics, and DNA stable-isotope probing (DNA-SIP) approach, we describe a previously hidden cohort of heterotrophic denitrifiers that use sulfur compounds as supplemental electron donors. This capability allows them to detoxify sulfide and achieve complete denitrification, thereby mitigating the $N_2O$ emission. This work enhances our understanding of heterotrophic denitrifiers capable of sulfide detoxification in aquatic environments, with broader implications for regulating carbon, nitrogen, and sulfur cycles, including the mitigation of $N_2O$ emissions in both natural and engineered system.

## Results

### Both sulfide and organic matter fuel nitrate reduction in estuarine sediments

Estuarine sediments were sampled at the junction of the Songhua River and effluent from municipal treatment in 2019 (Supplementary Fig. 1a). Despite the low concentration of sulfide in situ (<0.02 mM), sulfur cycling was active with 46.2 μM of sulfide produced in 20 mL (~24.2 g) of surface sediments over 52 h (Fig. 1a). These sediments contained $2.17 \pm 0.64 \times 10^4$ copies of *dsrA*, encoding dissimilatory sulfite reductase subunit A, per gram of sediments (Fig. 1b). Sulfide addition (53.8 μM) resulted in a 30.2% increase in nitrate consumption (Fig. 1c). We therefore conjectured that microbes inhabiting the community may be capable of reducing nitrate with sulfide as an electron donor. As the incubation reached a point of relatively slow nitrate reduction, potentially due to the depletion of electron donors, organic matter (OM) was introduced, triggering an instant decrease in nitrate (Fig. 1d). The concomitant increase of dinitrogen gas (94.2 μM), $N_2O$ (167.3 μM), and ammonium (14.6 μM) suggests that OM promoted both complete and incomplete denitrification, as well as DNRA. The high abundance of *narG*, *nirS*, *norB*, *nosZ*, and *nrfA* genes reinforces this inference (Fig. 1b). Thus, provision of the electron donors sulfide and OM simultaneously boost nitrate respiration in the estuary.

### Unexpected sulfur oxidation by heterotrophic denitrifiers

To better understand interactions between the carbon, nitrogen, and sulfur cycles, we conducted DNA-SIP in microcosms with estuarine sediments that were amended with combinations of carbon (C, with both inorganic and organic carbon added), nitrate (N, with $NO_3^-$ added), and sulfide (S, with $S^{2-}$ added): +C + N + S, +C + N, +$C_i$ + N + S, +$C_i$ (only inorganic carbon provided), and +$C_o$ (only organic carbon provided) (Supplementary Fig. 2). To determine whether chemolithoautotrophs or heterotrophs drive nitrate reduction, we conducted

DNA-SIP in microcosms with either inorganic ($^{13}C_i$, $^{13}CO_2$ and bicarbonate-$^{13}C$) or organic composite ($^{13}C_o$, glucose-$^{13}C$, propionate-$^{13}C$, and acetate-$^{13}C$) carbon labeled, along with unlabeled carbon substrates ($^{12}C_i$ and $^{12}C_o$). These incubations led to increases in bulk DNA buoyant density along with 16S rRNA gene copies as determined by qPCR, showing that both inorganic $^{13}C_i$ and organic $^{13}C_o$ were incorporated into cells in both +C + N + S and +C + N microcosms (Fig. 2a, b). In comparison to the heavy fraction without $^{13}C$ labeling in the +C + N + S microcosm, $^{13}C_o$ labeling led to increases of 1.3%, 21.6%, and 3.8% in the relative abundances of *Thauera*, *Azoarcus*, and *Pseudomonas* in the heavy DNA fraction, but these taxa decreased by 0.1%, 6.1%, and 1.5% in relative abundance when provided with $^{13}C_i$, suggesting a possible heterotrophic lifestyle (Fig. 2c). Similar increases (0.7–29.8%) in these taxa were observed in +C + N microcosm accompanied by $^{13}C_o$ assimilation (Fig. 2d). When starved of nitrate, very little increase in these three genera was observed in the heavy DNA fraction regardless of the presence of OM (Fig. 2e). We therefore conjectured that in addition to heterotrophy, they might be also capable of reducing nitrate. Unlike most other studies suggesting that sulfide oxidation coupled to denitrification is primarily driven by chemolithoautotrophs in freshwater, estuarine, and coastal environments[14,25–27], the identified heterotrophic denitrifiers including *Azoarcus* and *Pseudomonas* increased $^{13}C_o$ assimilation in response to sulfide addition, with their relative abundances increasing by 64.1% and 8.0% in ($^{12}C_i + {}^{13}C_o$)+N + S treatment compared to ($^{12}C_i + {}^{13}C_o$)+N treatment (Fig. 2c, d). This appears to suggest that these heterotrophs contribute to both sulfur oxidation and denitrification in our system. Despite this, interestingly, the metabolic trait of these two genera in $^{13}C_o$ assimilation with reliance on nitrate in non-sulfide-involved ($^{12}C_i + {}^{13}C_o$)+N treatment (Fig. 2d, e), indicates that these heterotrophs are capable of denitrification irrespective of sulfide addition. In contrast, the provision of $^{13}C_i$ in +C + N + S and +$C_i$ + N + S microcosms consistently led to 4.5- and 4.6-fold increases in *Thiobacillus* in the heavy fraction (Fig. 2c, f), whereas *Thiobacillus* did not increase in that of the +$C_i$ microcosm (Fig. 2g), suggesting a primarily chemolithoautotrophic lifestyle with reliance on nitrate. Collectively, these findings underscore the metabolic versatility of *Azoarcus* and *Pseudomonas* as heterotrophic denitrifiers capable of sulfur oxidation and nitrate reduction. Meanwhile, *Thiobacillus* has a more specialized chemolithoautotrophic role, highlighting the diverse metabolic strategies present within the microbial community.

In addition to 16S rRNA genes, we tracked the gene abundance of biomarkers for nitrate reduction (*napA*, *narG*), nitrite reduction (*nirS*, *nirK*), nitrous oxide reduction (*nosZ*), sulfide oxidation (*sqr*), and thiosulfate oxidation (*soxB*) along DNA density gradients (Fig. 2h–m and Supplementary Fig. 3). The high proportion of these genes in the heavy fractions in ($^{13}C_i + {}^{12}C_o$)+N + S and ($^{12}C_i + {}^{13}C_o$)+N + S treatments illustrated that both chemolithoautotrophs and heterotrophs contributed to nitrate reduction and sulfur oxidation (Supplementary Fig. 3a–g). Compared to the treatment amended only with $^{13}C_o$, the significantly higher abundance of *sqr* and *nirS* gene copies in heavy fractions of ($^{12}C_i + {}^{13}C_o$)+N + S treatment suggests the possible contribution of heterotrophic denitrifiers to sulfur oxidation (Fig. 2i, j, l, m). Intriguingly, in the ($^{12}C_i + {}^{13}C_o$)+N treatment, both *sqr* and *soxB* were highly enriched in heavy DNA fractions, with 30.9% and 17.8% abundance (Fig. 2i and Supplementary Fig. 3m, n). This indicates that the *sqr*-/*soxB*-encoding heterotrophic denitrifying microorganisms are stimulated when organic carbon and nitrate are provided even without sulfide (Supplementary Fig. 3j–n). These findings suggest that heterotrophic denitrifiers could potentially engage in sulfur oxidation without relying on it, as potential F-SOHDs. By linking element cycling functions inferred from isotope incorporation in the microcosms to relative abundance of individual genera, we found that 60.8% of the denitrifying sulfur-oxidizer community was composed of heterotrophs, while only 12.1% were chemolithoautotrophs (Supplementary

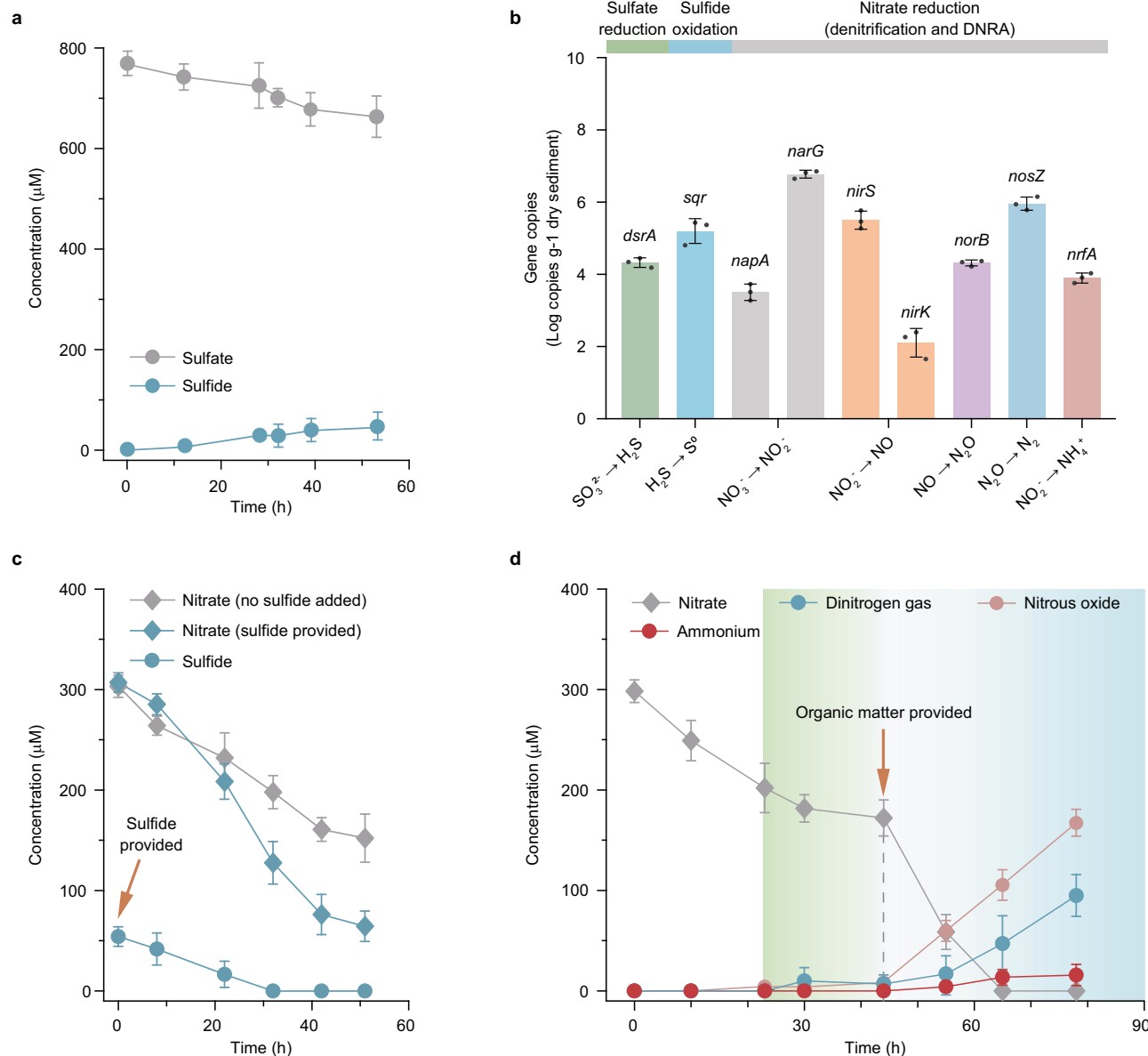

**Fig. 1 | Functional characteristics and injection reaction kinetics of the estuary sediments. a** Assessment of sulfate reduction activity in estuary sediment microcosms (three biological replicates). Sediment was incubated with anoxic overlying water. **b** Quantification of gene copies of biomarkers associated with sulfate reduction, sulfide oxidation, and nitrate reduction in sediments using qPCR. Nitrate reduction includes both denitrification and dissimilatory nitrate reduction to ammonium (DNRA) processes. All gene copies are represented per gram of dry sediment across three experimental replicates. Biomarkers include: *dsrA*, dissimilatory sulfite reductase subunit A; *sqr*, sulfide:quinone oxidoreductase; *napA*, cytochrome nitrate reductase; *narG*, nitrate reductase subunit alpha; *nirS*, cytochrome-cd$_1$ nitrite reductase; *nirK*, copper-containing nitrite reductase; *norB*, nitric oxide reductase subunit B; *nosZ*, nitrous oxide reductase, and *nrfA*, ammonium-producing cytochrome c−552 nitrite reductase. **c** Nitrate reduction facilitated by the provision of nitrate and sulfide (three biological replicates). The blue circles illustrate the consumption of sulfide upon its supplementation. Sulfide levels in the incubation without the supplementation of sulfide are below detection. **d** Nitrate reduction facilitated by the provision of nitrate and organic matter (OM) (three biological replicates). OM is supplied once the nitrate reduction reaches a slow consumption rate. Atom concentrations of sulfur and nitrogen compounds are represented. All data are presented as mean values ± SEM. Source data are provided as a Source Data file.

Fig 4). Compared to the +C + N microcosm, the relative abundance of *Thiobacillus*, the most abundant chemolithoautotroph in the community, decreased by an average of 3.8% in the +C + N + S microcosm (Fig. 2c, d). This suggests that F-SOHDs outcompete chemolithoautotrophs when sulfide is present. Taken together, these microcosm data imply that heterotrophs, rather than obligate chemolithoautotrophs, may dominate sulfur oxidation and denitrification in this ecosystem.

### Metabolic potential of F-SOHDs revealed by metagenomics
To better understand the metabolic potentials of key microbes identified by DNA-SIP, metagenomic sequencing was conducted on heavy DNA fractions from microcosms and natural sediments. A total of 63 high- and medium-quality representative metagenome-assembled genomes (MAGs) were reconstructed (Fig. 3a–c and Supplementary Table 1). Functional annotation revealed that 15 MAGs, primarily from *Desulfobacterota*, *Betaproteobacteria*, *Gammaproteobacteria*, *Bacillota*, and *Bacteroidota*, represent heterotrophic sulfur oxidizers harboring near-complete denitrification or DNRA pathways (Fig. 3a, c). Virtually all of these MAGs encode either periplasmic nitrate reductases (*napAB*) or membrane-bound nitrate reductases (*narGHI*), indicating the widespread potential for nitrate reduction (Supplementary Fig. 5). Approximately 60% of those MAGs harbor one or two nitrite

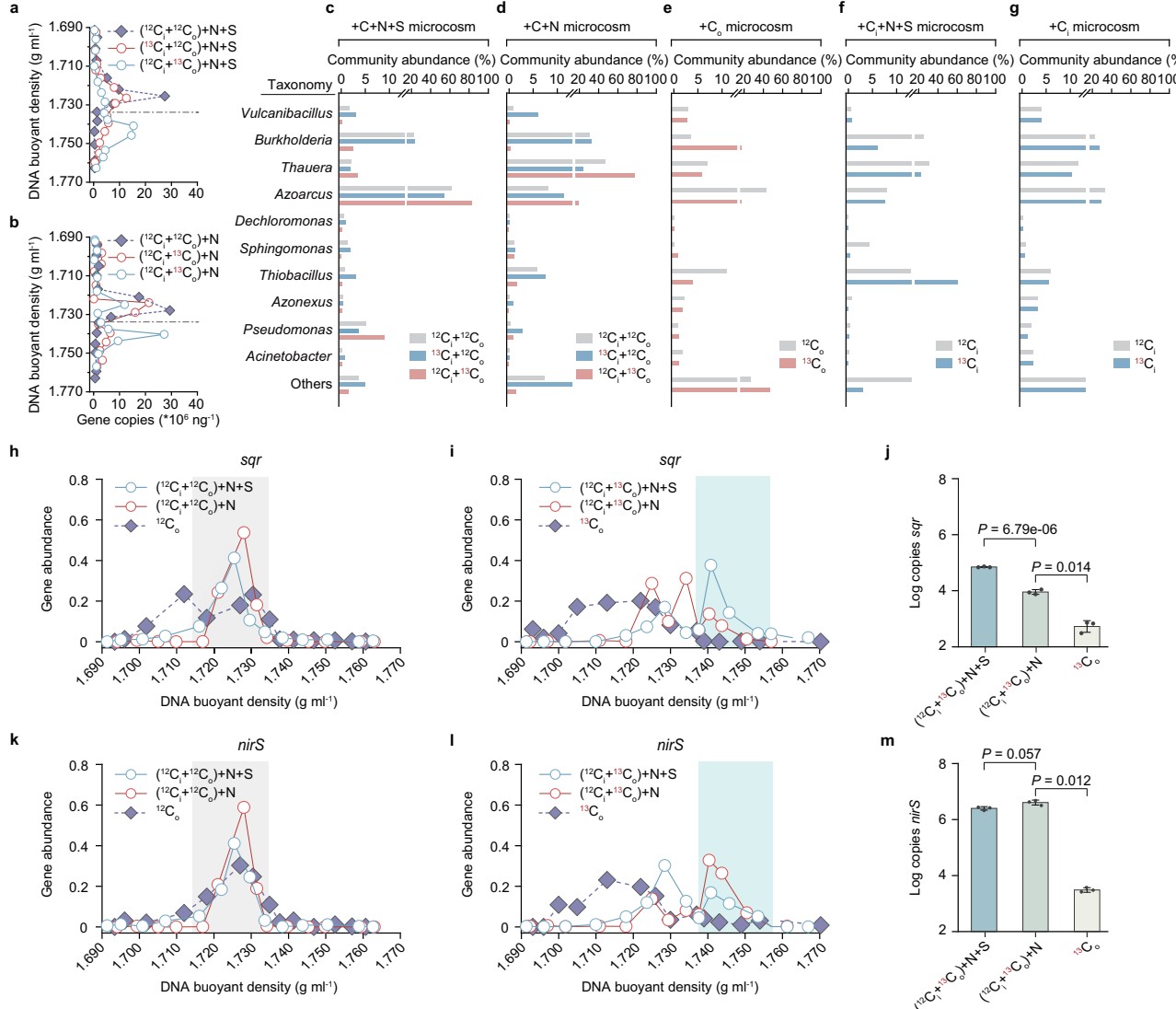

**Fig. 2 | Carbon, nitrogen, and sulfur cycling in estuary sediments revealed by DNA stable-isotope probing (DNA-SIP).** DNA-SIP profiles showing buoyant density distribution of community 16S rRNA genes determined by qPCR for incubations treated with +C + N + S (**a**) and +C + N (**b**). C denotes the provision of both inorganic ($C_i$) and organic ($C_o$) carbon. $C_i$ and $C_o$ represent inorganic carbon ($CO_2$ plus bicarbonate) and organic carbon (glucose, propionate, and acetate) provided, while N and S denote additions of nitrate ($NO_3^-$) and sulfide ($S^{2-}$). The dotted horizontal line indicates the demarcation between heavy and light fractions. **c–g** Microbial community composition of heavy DNA fractions with and without isotope labeling in +C + N + S, +C + N, +$C_o$, +$C_i$ + N + S, and +$C_i$ microcosms, respectively. Only the ten most abundant genera are presented. **h**, **i** Gene abundance of *sqr* along the DNA density gradient for the $^{12}$C-unlabeled and $^{13}$C-labeled incubations.

For each buoyant density fraction, the gene abundance is normalized as a percentage of its copy number relative to the total DNA copy number. Peak gene abundance with and without isotope labeling are highlighted in cyan and gray. **j** Comparison of the total copy numbers of *sqr* gene across three $^{13}C_o$-labeled microcosms (two-tailed unpaired Student's *t* test with 95% confidence intervals from three technical replicates). Data are presented as mean values ± SEM. **k**, **l** Gene abundance of *nirS* along the DNA density gradient for the $^{12}$C-unlabeled and $^{13}$C-labeled incubations. **m** Comparison of the total copy numbers of *nirS* gene across three $^{13}C_o$-labeled microcosms (two-tailed unpaired Student's *t* test with 95% confidence intervals from three technical replicates). Data are presented as mean values ± SEM. Source data are provided as a Source Data file.

reductases encoded by *nirS* and *nirK*, accompanied by *norBC*, *nosZ*, and accessory/regulatory genes (e.g., *nosR*), pointing to denitrification capacity. In contrast to most known heterotrophic denitrifying bacteria that are unable to use sulfur compounds as electron donors[28–30], these MAGs also encode various enzymes for sulfur oxidation (Fig. 3c and Supplementary Fig. 5), the most prevalent being sulfide:quinone oxidoreductase (SQR), a key enzyme that can catalyze sulfide detoxification in nitrate-reducing chemolithoautotrophs[19,31]. Additionally, genes encoding thiosulfate:cyanide sulfurtransferase (TST) and periplasmic *sox* gene clusters encoding for thiosulfate oxidation and sulfane dehydrogenation were also detected[15,32]. Flavocytochrome c sulfide dehydrogenase (FccAB), which has a role in oxidative sulfur

metabolism in photolithoautotrophic and chemolithoautotrophic sulfur bacteria[33,34], was also abundant. The presence of sulfur dioxygenase (ETHE1) in partial MAGs suggests that zero-valent sulfur or polysulfide can also be oxidized to sulfite by glutathione[35].

Unlike well-studied chemolithoautotrophs including *Sulfurimonas*[19], *Thiobacillus*[36], and SUP05 groups[37] that are widely distributed and capable of both denitrification and sulfur oxidation, most denitrifiers in the present study lack known $CO_2$ fixation pathways (Fig. 3c and Supplementary Fig. 5), herein identified as F-SOHDs. Consistent with the absence of known autotrophic pathways, M4.bin9 (*Pseudomonas*), M2.bin44 (*Ralstonia*), M5.bin12 (*Dechloromonas*), M5.bin61 (*Thauera*), M5.bin41 (UBA2357), M4.bin17 (*Azoarcus*), and

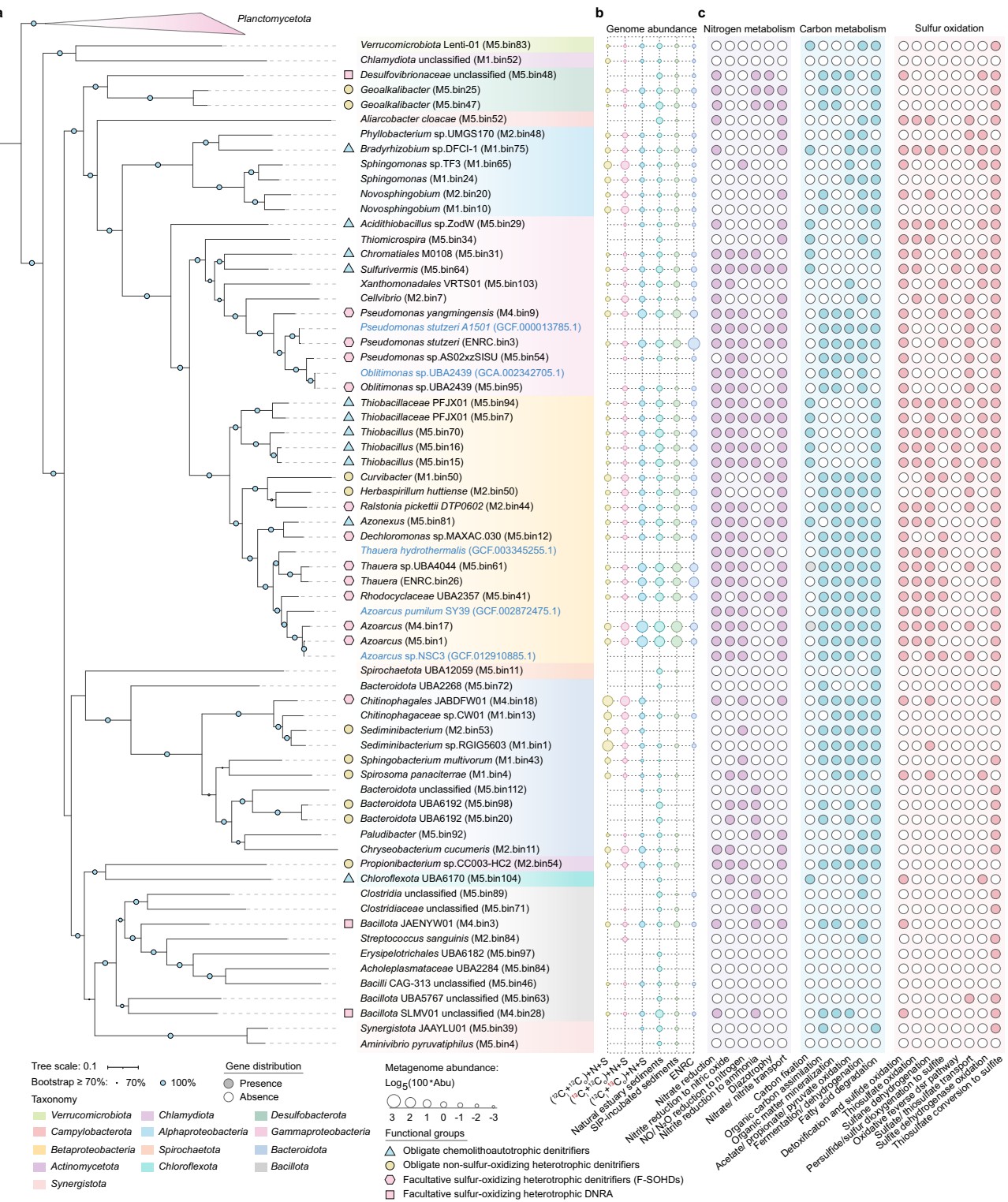

**Fig. 3 | Phylogenetic tree, relative abundance, and gene content of metagenome assembled genomes (MAGs) identified as the facultative sulfide-oxidizing heterotrophic denitrifiers (F-SOHDs). a** Phylogenetic placement of 63 representative MAGs assembled in the present study. Genomes from the phylum *Planctomycetota* were selected as the outgroup. Five complete genomes retrieved from public databases that are closely related to F-SOHDs and have similar inferred functions were employed as reference genomes. **b** Relative abundances of assembled MAGs in natural sediments, short-term isotope-incubated sediments, heavy DNA fractions of DNA-SIP experiments, and enrichment cultures. ENRC represents the metagenome of ENR_C4 enrichment. **c** Presence/absence of genes involved in carbon, nitrogen, and sulfur metabolism. Solid/hollow circles indicate the presence/absence of target pathways. Gray circles within the carbon fixation column indicates that DNA-SIP experiments and abundance shifts confirm the inability to fix carbon despite the presence of *rbcL* gene in corresponding genome. Source data are provided as a Source Data file.

M5.bin1(*Azoarcus*) MAGs were more abundant in metagenomes of heavy DNA fractions amended with [13]$C_o$ than those with [13]$C_i$ and unlabeled-[12]C treatments (Fig. 3b), implying their enrichment on OM and incorporation of [13]$C_o$ to support growth. While these results point to a predominantly heterotrophic lifestyle, some members of these genera might possess the capacity for $CO_2$ fixation under specific conditions, though this possibility cannot be confirmed due to the incomplete genomes reconstructed from the metagenomic sequencing. The detection of genes diagnostic of heterotrophy including fermentations and carbohydrate mineralization in many of the MAGs is also consistent with the heterotrophic lifestyle of these microbes (Supplementary Figs. 5, 6). Most of them also encode enzymes to synthesize acetyl-CoA from acetate via acetyl-CoA synthetase (Acs) and/or acetate kinase-phosphate acetyltransferase (Ack-Pta), both of which could be coupled with the formation of ATP[38,39]. The presence of genes encoding glycoside hydrolases, glycolysis, β-oxidation, and propionyl-CoA production in these MAGs suggests that organic molecules such as carbohydrates, long-chain fatty acids, and propionate can be utilized as carbon sources to conserve energy and support heterotrophic growth[40–42]. Our findings suggest a strong genomic potential for F-SOHDs within these sediments at the study site, supporting their responses in the microcosm experiments. This highlights the importance of previously unrecognized heterotrophic denitrifiers in the cycling of both sulfur and nitrogen. Apart from denitrifiers, several heterotrophs encoding DNRA and sulfur oxidation were also identified (Supplementary Fig. 7). Some ammonium was produced in nitrate respiration assays (Fig. 1d), and *nrfA* genes were uncovered by qPCR (Fig. 1b) and in MAGs (Fig. 3c), confirming DNRA in certain microbes.

### Long-term enrichment cultures confirm the non-sulfur-dependent physiology of F-SOHDs

A custom bioreactor system was used to enrich F-SOHDs from sediments based on alternating medium with and without sulfide (Fig. 4a). Two enrichments named ENR_C4 and ENR_U2 were successfully maintained for 612 days. Kinetic tests showed that approximately 140–300 μM of sulfide, soluble zero-valent sulfur, thiosulfate, and sulfite could be oxidized to sulfate in ENR_C4, accompanied by the consumption of OM and an increase in cell numbers (Supplementary Figs. 8a–I, 9). Though both enrichments reduced nitrate along with OM consumption, the poor net cell growth suggested that microbes inhabiting the ENR_U2 community were only able to oxidize sulfide, but not other sulfur compounds (Supplementary Fig. 10). Two representative MAGs identified as *Pseudomonas* (ENRC.bin3) and *Thauera* (ENRC.bin26), with a cumulative relative abundance of 88.6% in the ENR_C4 enrichment culture (Fig. 3b; Supplementary Fig. 8k), were interpreted to be F-SOHDs. Interestingly, *Thauera* (ENRC.bin26) was enriched in response to sulfide addition and possessed a complete sulfide oxidation pathway despite decreasing in relative abundance in the [13]$C_o$-incorporated DNA fraction (Figs. 2c,d, 3c). We speculate that the pattern of enrichment of *Thauera* (ENRC.bin26) may be attributed to the competition with other microorganisms, such as *Pseudomonas* (ENRC.bin3), under certain environmental conditions. It is possible that high sulfide concentrations lead *Thauera* to be outcompeted by *Pseudomonas*, thereby lowering its relative abundance and contribution to the overall community. This pattern has been observed in benthic marine waters, where the sulfide oxidizers *Arcobacter*, SUP05, and *Sulfurovum* co-exist at low sulfide levels, but *Arcobacter* and *Sulfurovum* dominate over SUP05 when sulfide concentrations exceed 10 μM[10].

To further verify the coupling of denitrification with sulfur oxidation, we performed three consecutive incubations, each with different substrates provided (Fig. 4b). In the first stage, the addition of OM and nitrate into ENR_C4 led to a complete removal of nitrate and rapid decrease in OM. Then in the second stage, sulfide and nitrate were added. About half of the sulfide was completely oxidized to sulfate, which was concomitant with further consumption of both OM and nitrate, consistent with sulfide oxidation coupled with heterotrophic denitrification. When harvested cells from this enrichment were transferred to sulfide-free medium in the third stage, heterotrophic denitrification capacity in the absence of sulfide was restored. This experiment suggests key denitrifiers in enrichments modify their metabolism to adjust to the presence or absence of sulfide. OM was necessary for sulfur oxidation since nitrate was stable in the absence of OM, whereas it decreased upon replenishment of OM (Fig. 4c). This was further supported by the higher net increases in cell numbers when cultures were amended with OM (Student's *t* test, $P = 0.004$ and 0.002; Fig. 4d, e).

### Overlooked significance of F-SOHDs in sulfide detoxification and reducing N₂O emission

To verify the ecological significance of F-SOHDs, we examined biochemical cycling within different laboratory cultures (Supplementary Fig. 11a–x). As OM flux is known to control nitrogen loss[43], experiments were conducted under either OM-rich or OM-limited conditions. Although minimal sulfide was consumed in enrichment cultures of conventional heterotrophic denitrifiers (Fig. 5a and Supplementary Fig. 11f, t), sulfide was completely consumed within 32 h in enrichment cultures dominated by F-SOHDs (enrichments ENR_C4 and ENR_U2), regardless of OM concentrations. The contributions of OM and sulfide as electron donors for denitrification were evaluated (Fig. 5b, c). Analysis of electron flow balance indicated that although OM dominates (Supplementary Fig. 11a, d, p, r), approximately 24.6–27.1% of electrons supporting denitrification could be generated by the oxidation of sulfide to zero-valent sulfur (Fig. 5b). With the provision of sulfide, less OM was consumed by F-SOHDs than by conventional denitrifiers (Fig. 5d). To a certain extent, this implies that despite the necessity of OM for growth, sulfide might be an important supplemental electron donor. When fed with limited OM, two F-SOHDs can theoretically incorporate an additional 34.7–102.4% of electrons from sulfide (Fig. 5c), promoting complete denitrification (Supplementary Fig. 11w, x). With this, we conjecture that, rather than simply suppressing heterotrophic denitrifiers, sulfide might serve as an alternative electron source for certain denitrifiers that can oxidize it to less toxic sulfate/sulfur, thereby mitigating or preventing sulfide toxicity and potentially accelerating the denitrification process. These extra electrons could overcome a bottleneck imposed by limited reducing power under low OM conditions that can lead to incomplete denitrification by conventional heterotrophic denitrifiers. For example, OM-limited conditions led to only 16.3% of nitrate being reduced to dinitrogen gas in conventional heterotrophic denitrifier enrichment (Supplementary Fig. 11s), whereas complete denitrification was observed in an OM-limited culture containing F-SOHDs when they were supplemented with sulfide (Supplementary Fig. 11w).

Interestingly, further experiments suggests that the F-SOHDs could significantly lower the release of the greenhouse gas N₂O. In these experiments, conventional heterotrophic denitrifiers produced 55.8% more N₂O and less dinitrogen gas when exposed to only 4.83 μM sulfide (Fig. 5e, f and Supplementary Fig. 11c–f). This is consistent with the reported observations of excess N₂O accumulation reported by sulfide suppression in the Shinji Estuary[11]. Remarkably, negligible N₂O emission and comparable levels of nitrogen in the form of dinitrogen gas and initial nitrate were detected in two F-SOHD enrichments, suggesting a complete denitrification process when sufficient OM is provided (Fig. 5g and Supplementary Fig. 11o, q). Although slow, chemolithoautotrophic denitrifiers also achieved the full nitrate reduction to dinitrogen gas (Supplementary Fig. 11g). In contrast, in the presence of sulfide, the conventional heterotrophic denitrifiers released more N₂O, accounting for approximately 50% of the reduced nitrogen (Supplementary Fig. 11e). When OM is deficient, the additional

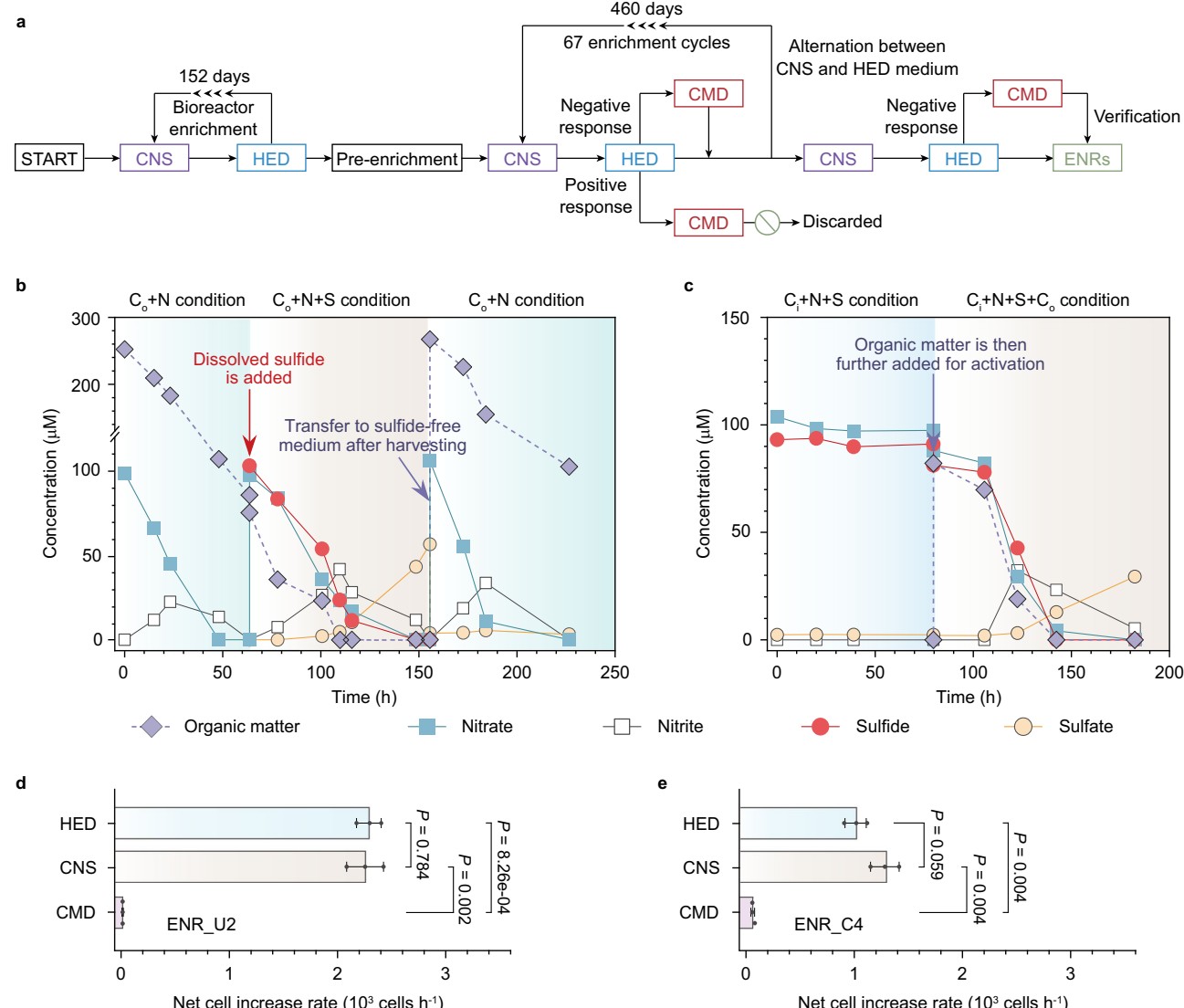

**Fig. 4 | Isotope- and genome-informed enrichment and physiology kinetics of putative F-SOHDs. a** A custom strategy used for the long-term enrichment of F-SOHDs. Estuary sediments (referred to START) served as the inoculum. Given that F-SOHDs can couple denitrification with OM oxidation with or without sulfide as an additional electron donor, the medium altered between the absence and presence of sulfide, iterating for 67 cycles over a duration of 460 days. A 152-day sequencing batch bioreactor was operated to pre-enrich the sediments with relatively high biomass. CMD represents the incubation condition wherein inorganic carbon, nitrate, and sulfide are supplied. Within each dilution transfer cycle, the enrichments generated from HED incubation were introduced into CMD incubation to test chemolithoautotrophic potential. Enrichments demonstrating positive responses to the CMD condition were discarded, while those displaying unfavorable response were inoculated to the next cycle. ENR represents the ultimate enrichments (ENR_C4 and ENR_U2) dominated by F-SOHDs. **b, c** Reaction kinetics of ENR_C4 enrichment with different substrates provided. In the $C_o + N$ and $C_o + N + S$ treatments, OM serves as the sole carbon source, whereas the $C_i + N + S$ treatment relies on inorganic carbon provision. Dissolved sulfide and fresh nitrate are provided upon the depletion of nitrate and nitrite at 63 h. **d, e** Net increase rates of cell numbers of ENR_C4 and ENR_U2 under different incubation conditions (two-tailed unpaired Student's $t$ test with 95% confidence intervals). Data from three biological replicates are presented as mean values ± SEM. Source data are provided as a Source Data file.

electrons released from the complete oxidation of sulfide to sulfate, compared to partial oxidation to sulfur, can give rise to lower 72.6% $N_2O$ emission and 42.0% higher production of dinitrogen gas by F-SOHDs (Supplementary Fig. 11u, w). Our results highlight that F-SOHD is a crucial sink for mitigating greenhouse effects by reducing $N_2O$ emissions through harnessing various substrates as electron donors to support complete denitrification.

## Discussion

DNA-SIP is a cultivation-independent tool used to identify active microorganisms that assimilate specific substrates into biomass. While DNA-SIP technology is not restricted to a single element, understanding the complex interactions between multiple element cycles remains a challenge. By considering variations in isotope uptake within the context of varying carbon, nitrogen, and sulfur resources, we are able to reveal the functional roles of microbes in the cycling of all three elements and have successfully identified a unique group of microorganisms we termed F-SOHDs. Despite allowing for functional inference, it primarily highlights the more active microbes and is limited in its ability to differentiate extremely low-abundance F-SOHDs from other community members. Additionally, it is noted that some heterotrophs might also incorporate inorganic carbon under certain conditions[44] and cross-feeding could also occur, which complicate the interpretation.

Many studies have demonstrated that chemolithoautotrophic denitrifiers, such as *Epsilonproteobacteria* and the *Gammaproteobacteria* SUP05 clade[14,16,19,34], are the prevalent groups that detoxify

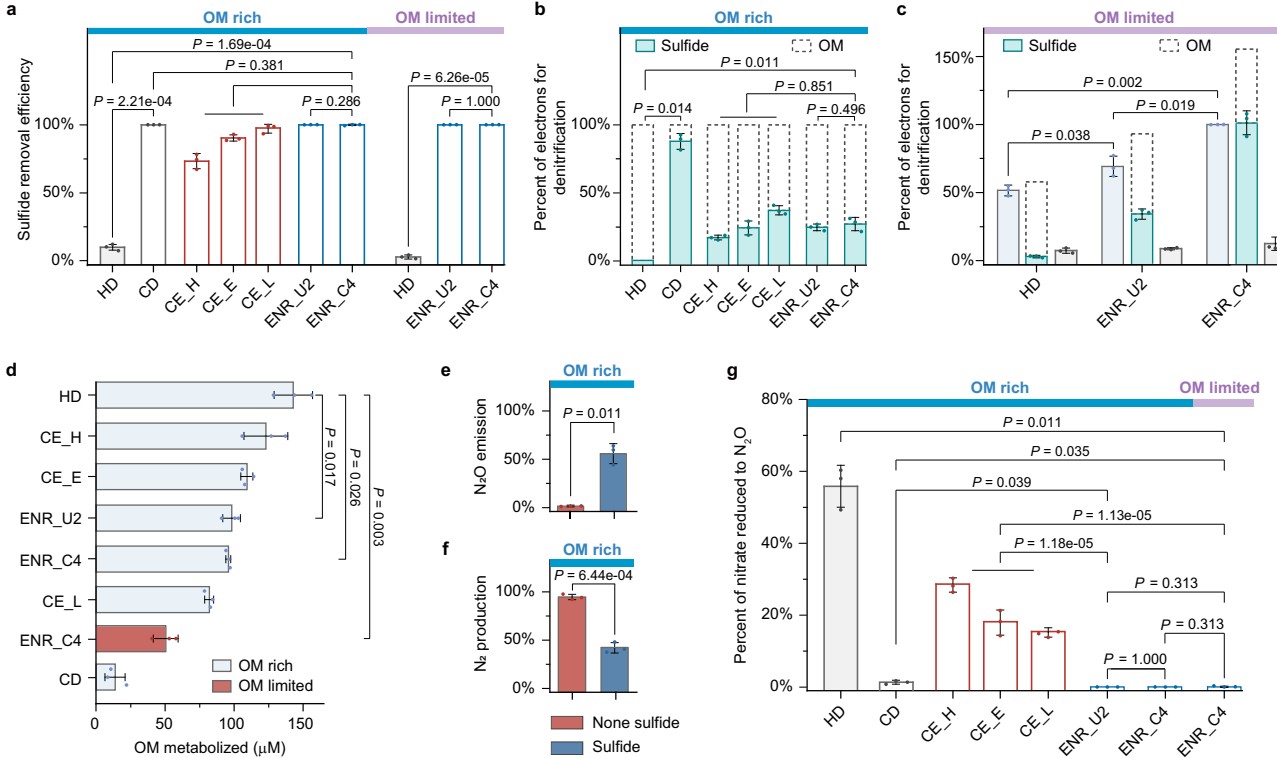

**Fig. 5 | Impact of F-SOHDs on biogeochemical cycling and nitrous oxide (N₂O) emission. a** Sulfide removal in different laboratory cultures under OM-rich and OM-limited conditions. HD is a sediment enrichment culture containing conventional heterotrophic denitrifiers that do not engage in sulfur oxidation. CD is a pure culture of the chemolithoautotrophic sulfur-oxidizing denitrifier *Thiobacillus denitrificans* ATCC 25259. The ENR_U2 and ENR_C4 are two enrichments dominated by F-SOHDs with incomplete and complete sulfur oxidation capacities, respectively. The three different microbial communities were established by combining the HD culture with the CD *T. denitrificans* in ratios of 3:1 (CE_H, high ratio), 1:1 (CE_E, equal ratio), and 1:3 (CE_L, low ratio). Both ENR_U2 and ENR_C4 experiments were replicated three times under OM-limited conditions, and the final concentration of sulfide were consistently below detection limits. **b** Contribution of electrons derived from sulfide (green boxes) and OM (dashed lines) to denitrification process in OM-rich cultures. The calculation is conducted under the assumption that nitrate is fully reduced to dinitrogen gas. **c** Contribution of electrons derived from sulfide (green boxes) and OM (dashed lines) to denitrification in OM-limited cultures. The light blue boxes indicate the completion extent of denitrification (i.e., the percent of electrons produced as a function of the total electrons needed to completely reduce nitrate to dinitrogen). Gray boxes represent the completion extent of denitrification in the absence of OM. **d** OM metabolism by various laboratory cultures under OM-rich and OM-limited conditions. The comparison of OM consumption is conducted under the assumption that total electron donors in each laboratory culture are sufficient for supporting complete denitrification **e**, **f** Emission of N₂O and N₂ by the enrichment culture of conventional heterotrophic denitrifiers in the presence or absence of sulfide under OM-rich conditions. Percentages are shown by nitrogen atoms normalized to the initial concentration of nitrate. **g** N₂O emission by various laboratory cultures under OM-rich and OM-limited conditions. All experimental data from three biological replicates are presented as mean values ± SEM (two-tailed unpaired Student's *t* test with 95% confidence intervals). Source data are provided as a Source Data file.

sulfide in marine, coastal, and terrestrial aquatic ecosystems. In this study, we present compelling evidence that some heterotrophic denitrifiers can also engage in sulfur oxidation, and play an important but previously unrecognized role in multiple biogeochemical cycles in estuarine environments. Unlike chemolithoautotrophs that fix CO₂ into biomass, F-SOHDs are chemolithoheterotrophs that couple heterotrophic respiration to sulfide oxidation and nitrate reduction. Through a combination of microcosm incubation, DNA-SIP, qPCR, and metagenomics, we identified diverse F-SOHDs from estuarine sediment communities, including members of the genera *Pseudomonas*, *Thauera*, *Azoarcus*, *Ralstonia*, *Dechloromonas*, *Oblitimonas*, and *g_UBA2357* (Fig. 3). These findings highlight the functional diversity of microbial communities involved in sulfide oxidation and nitrate reduction in sulfidic environments. Similar microorganisms have also been reported in other habitats. For instance, the marine strain *Arcobacter peruensis*[23], recently isolated from the organic- and sulfide-rich waters of the Peruvian OMZ shelf, couples sulfide oxidation to nitrate reduction while assimilating acetate. Similarly, a strain described as *Thauera* HDD1 demonstrates such metabolic traits in sulfide-rich denitrifying sludge[45]. In wastewater bioreactors, the putative F-SOHD described as *Pseudomonas* C27 oxidizes sulfide to elemental sulfur

using organic carbon as an obligatory substrate[46]. Notably, significant metabolic differences can occur even among related microbes. For example, *Pseudomonas yangmingensis* and *Pseudomonas* H117 could demonstrate chemolithoautotrophic capabilities[24,47], whereas certain *Pseudomonas stutzeri* strains function as denitrifiers, capable of either chemolithoautotrophic sulfide oxidation or heterotrophic thiosulfate oxidation[48–50]. Additionally, two haloalkaliphilic species from soda lakes, *Halomonas mongoliensis* and *Halomonas kenyensis*[51], oxidize sulfide using acetate as a carbon source and nitrous oxide or nitrate as electron acceptors, suggesting they are potential F-SOHDs. While previous studies have reported F-SOHDs in marine and wastewater-related systems at the single strain level, our study provides robust evidence from complex sediment communities. Our findings reveal that F-SOHDs are a persistent component of microbial communities in OM-rich estuaries, with important implications for the sulfur cycle and N₂O reduction.

Sulfide detoxification is crucial for healthy ecosystems and the survival of most denitrifiers. As documented in previous studies[10,30], the high input of organic matter and seasonal oxygen depletion may result in the intermittent release of toxic H₂S from anoxic benthic sediments into the overlying water column, impairing the activity of

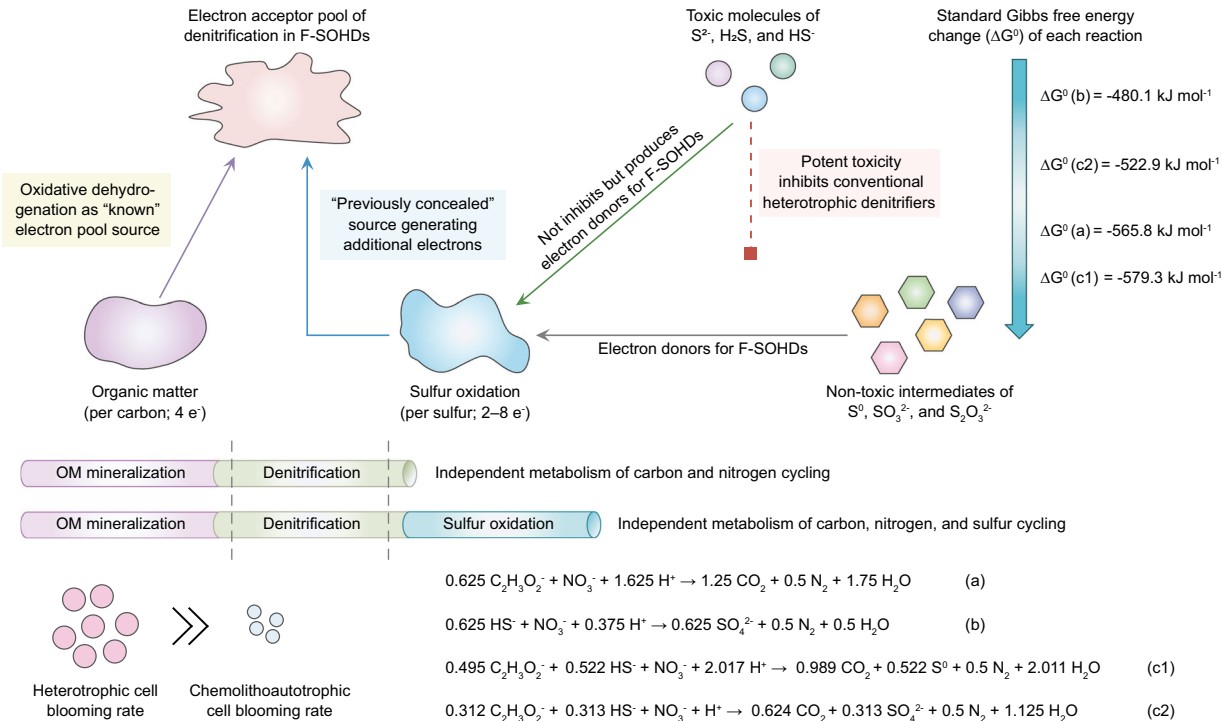

**Fig. 6 | Proposed ecophysiological advantages and environmental adaptations of F-SOHDs.** The red dashed line highlights the toxic inhibition of sulfide on conventional heterotrophic denitrifiers. Purple area indicates the known electron pool generated from OM for heterotrophic denitrifiers, while the blue represents the previously concealed electron pool contributed by toxic sulfide and non-toxic sulfur compounds for F-SOHDs. $\Delta G^0$(a) and $\Delta G^0$(b) denote standard Gibbs free energy change per nitrate reduction for conventional heterotrophic denitrifiers and chemolithoautotrophic sulfur-oxidizing denitrifiers (equations a and b).

$\Delta G^0$(c1) and $\Delta G^0$(c2) represent standard Gibbs free energy change for F-SOHDs with incomplete and complete sulfur oxidation capacities (equations c1 and c2). These calculations were conducted under the standard conditions (298 K, 1 atm, and 1 M concentrations). The formation energies of each substance were obtained from the standard thermodynamic database provided in *CRC Handbook of Chemistry and Physics*[96]. The metabolic rates of cell blooming between F-SOHDs and chemolithoautotrophs are depicted by pink and blue circles.

many microorganisms. This phenomenon, known as sulfidic events, frequently occurs in eutrophic oxygen minimum zone (OMZ) shelf regions[10], where it promotes the release of dissolved phosphate[52], further exacerbating eutrophication[10]. In extreme cases, large amounts of toxic sulfide may diffuse into oxygenated layers, causing mass die-offs of fish and invertebrates[9,10,53]. Remarkably, the F-SOHDs may have developed multiple sulfur oxidation pathways to cope with sulfide stress, a hypothesis that merits further exploration. Such capacity not only enables them to detoxify sulfide molecules, but also to harvest energy to support their growth and functional coupling with denitrification (Fig. 6). This unique physiology appears ideally suited to adapt to the changing environments posed by intermittent sulfidic events. Specifically, F-SOHDs may carry out non-sulfur-dependent heterotrophic denitrification during periods without sulfide, while they respond to sulfidic events by expressing sulfur oxidation pathways. This previously concealed ecophysiology may benefit other biochemical reactions or surrounding animals by relieving sulfide toxicity in anoxic ecosystems[29,54].

NosZ, the only enzyme in the biosphere responsible for converting the greenhouse gas and atmospheric reactant $N_2O$ into inert dinitrogen gas[55], is highly susceptible to environmental pressures[56,57]. Estuarine environments are hotspots for $N_2O$ emission and contribute substantially to the global $N_2O$ budget[58,59]. Previous studies have consistently shown that sulfide broadly inhibits denitrifiers by interacting with NosZ[60], a copper-dependent metalloenzyme that catalyzes the final step of denitrification. Even a minimal concentration of 1.25 μM can completely sequester the [Cu:S] active sites of NosZ enzyme via abiotic copper chelation[10,13,61,62]. This process causes nitrate reduction to terminate at $N_2O$ in microbial cultures, raising concerns that increases in water column sulfide due to increased organic or nutrient

loading could increase $N_2O$-diven greenhouse warming. For microorganisms lacking sulfide oxidation pathways, such as conventional heterotrophic denitrifiers, the inability to detoxify sulfide leads to NosZ inactivation, causing strong accumulation of $N_2O$[13,61]. This sulfide-induced $N_2O$ emission phenomenon has been observed in numerous ecosystems, including eutrophic estuarine and coastal sediments[11], bay and freshwater sediments[63,64], continental shelf waters[12], agricultural soils[65], marine and soil pure cultures[66,67], wastewater treatment-related activated sludges[13,68], and our own experiments (Fig. 5e). However, our enrichment cultures containing F-SOHDs instead showed that sulfide can actually reduce $N_2O$ emissions, suggesting a fundamental difference in physiological properties between F-SOHDs and conventional heterotrophic denitrifiers. By oxidizing sulfide, F-SOHDs prevent it from chelating the copper within Cu-rich NosZ, thereby enabling $N_2O$ reduction to continue. Sulfide oxidation could not only rescue the sulfide-sensitive NosZ [Cu:S] active sites but may simultaneously provide an additional electron donor beyond organic matter to reduce available nitrate to dinitrogen gas rather than to $N_2O$. These two processes confer a dual advantage in reducing the greenhouse gas footprint (Fig. 6 and Supplementary Fig. 12), offering a functional advantage similar to that of chemolithoautotrophs. In addition, sulfide may be a superior electron donor over organic matter since the oxidation of per mole of sulfide-S to sulfate yields twice the number of electrons (8) than the oxidation of per mole of OM-C to $CO_2$ (4). Thus, our results suggest that F-SOHDs may contribute to the reduction of $N_2O$ emissions under certain sulfidic conditions that are associated with high $N_2O$ production rates, although their role might be context-dependent and potentially overlooked in estuarine systems. To our knowledge, the ability of heterotrophs to self-detoxify sulfide and simultaneously reduce $N_2O$ has not been described in other bacteria.

To date, there has been limited knowledge about the distribution and ecological roles of F-SOHDs in natural ecosystems. By scanning genomes in public databases, we also found putative F-SOHDs in hot springs (*Thauera hydrothermalis* GD-2 and *Azoarcus taiwanensis* NSC3), surface seawater (*Azoarcus pumilus* SY39), and a sewage bioreactor (*Oblitimonas* sp. UBA2439) (Fig. 3a, c). These genomes were identified to harbor diverse genes encoding pathways for complete denitrification, OM mineralization, and sulfide/sulfur/thiosulfate oxidation, but lacked genes for carbon fixation. The presence of these microbes beyond estuaries suggests that F-SOHDs may also play crucial roles in mediating carbon, nitrogen, and sulfur cycling in other environments[69–72]. We even anticipate that F-SOHDs outcompete known chemolithoautotrophs due to their ability to thrive in the presence or absence of sulfide, as well as in environments with varying OM levels. This adaptability, coupled with favorable Gibbs free energy for their reactions ($-579.3$ to $-522.9$ kJ nitrate$^{-1}$) (Fig. 6), highlights their potential metabolic advantages. Moreover, the more efficient energetics of acetate into biomass, compared to $CO_2$ fixation via chemolithoautotrophic metabolism[23,44], imparts an additional advantage of lower ATP requirements for F-SOHDs, which could explain their capacity for rapid blooms. Their exceptional performance in sulfide detoxification and greenhouse gas control makes them promising targets for future studies focused on environmental sustainability and industrial applications, particularly under conditions with excessive organic matter. Introducing F-SOHDs into the anoxic denitrification tanks of wastewater treatment plants through bioaugmentation could strengthen the sulfide resistance of heterotrophic denitrifying activated sludge, where the undesired sulfide is often generated in sewer systems[13,68]. This strategy also holds promise for maintaining high efficiency in nitrogen and OM removal and mitigating the greenhouse gas footprint of $N_2O$ emissions. Despite its potential importance, future systematic investigations at larger scales are necessary to quantify the diversity, distribution, and significance of these microorganisms.

## Methods

### Estuary sampling and dynamic incubation

Estuarine sediments were collected in the main nutrient-exchanging hotspot zone belonging to Eastern Asia's largest Songhua River System in August of 2019 (45°49′12.81″N, 126°43′22.51″E). Five replicate sediment cores were sampled at 0–20 cm depth, with a water depth of 3.7 m. The surface sediments contained 48.35% sand, 22.57% total organic matter, 1.34% total nitrogen, 0.07% total phosphorus, and had a pH of 7.06. One overlying water sample was fixed immediately using sulfide antioxidant buffer[73] to prevent sulfide oxidation, while in another sample sulfide was precipitated by zinc chloride to filter for other chemical analyses. The anoxic bottom water contained 0.75–0.98 mM sulfate, 0.04–0.06 mM nitrate, 0.002–0.005 mM nitrite, 0.08–0.13 mM ammonium, and 1.03–1.79 mM total organic carbon at pH 7.13. The concentrations of sulfate, nitrate, and nitrite were measured using an ICS-3000 ion chromatograph (Dionex, USA), with detection limits of 0.2 μM for sulfate, 0.3 μM for nitrate, and 0.2 μM for nitrite. Ammonium was determined by the standard Nessler reagent method, with a detection limit of 1.8 μM. Total organic carbon was determined using a multi N/C 3100 (Analytik Jena, Germany), with a detection limit of 0.4 μM.

Sediment incubation experiments were performed in-situ to test the activity of element cycling in the estuary. Sediments for the measurements of sulfate reduction rate were rapidly filtered on-site through a homogenized mesh and then transferred into 340 mL glass-bottles with overlying water. A homemade assembly, featuring a three-way valve connected to an air bag pre-filled with 15 L pure helium and a vacuum pump, was employed for negative vacuum gas replacement. After inoculation, sediment microcosms were then incubated on-site on a thermostatic shaker at 22 °C before transportation to the lab.

Liquids were periodically sampled and stored on dry ice for later physiochemical analysis. Sulfate reduction activity in estuarine sediments was assessed by sulfate consumption and sulfide production over a per unit time using 20 mL of sediments. To determine relationships between nitrate respiration and sulfur cycle, equivalent volume of sediments was inoculated into two bottles of estuarine water, and then the nitrate concentration was initialized by diluting a stock solution. After deoxygenation, a group of bottles was additionally amended with a mixed sulfur stock solution with a 1:1 molar ratio of sulfate to sulfide to reach a concentration of ~50 μM sulfide compared with the control. While sediments for OM additions were similarly inoculated in anaerobic estuarine water that was initialized with nitrate and then incubated for 44 h at which time nitrate consumption became sluggish based on the limited dissolved organic carbon in the estuarine water. At that point, 800 μM of a composite OM stock solution, comprising equimolar glucose, propionate, acetate, and lactate, was added. The decrease in nitrate concentration and the production of dinitrogen gas and ammonium before and after OM amendments were then compared to determine the rate of nitrate respiration coupled to OM utilization in the estuary. All field sediment incubations were conducted in three repetitions.

### Quantitative PCR of biomarkers

DNA was extracted from 0.25 g of sediment using PowerSoil DNA isolation kit (MoBio, USA). Ten genes involved in sulfate reduction (*dsrA*), sulfide oxidation (*sqr*, *soxB*), nitrate reduction (*napA*, *narG*), denitrification (*nirS*, *nirK*, *norB*, *nosZ*), and DNRA (*nrfA*) were quantified and details regarding the primers are described in Supplementary Table 2. Cell numbers from each density gradient were estimated by counting the absolute 16S rRNA gene copies. All qPCR assays were performed in triplicate in a 20 μl reaction, containing 10 μL of TB-Green Premix Ex Taq (Takara, Japan), 0.4 μL of ROX Reference Dye, 0.4 μL of each forward/reverse primer (10 μM), 2 μL of standardized DNA template (17.5 ng μL$^{-1}$), and 6.8 μL of DNAase-free water. The reactions were carried out on a StepOnePlus ABI 7500 PCR system (Applied Biosystems, USA). DNA concentrations were determined using the Quant-iT PicoGreen dsDNA kit (Thermo Scientific, USA). Recombinant plasmids with a cloned target gene were tenfold diluted for an eight-point standard calibration curve ranging from 10 to10$^8$.

### Isotope incubation experiments

Prior to conducting microcosm experiments, sediments were rinsed with deoxygenated Milli-Q water within the BACTRONI-2 Anaerobic Workstation (ShelLab, USA) to eliminate insoluble sandy particles and extracellular residues[22]. To stimulate microbial activity within the communities, the sediments underwent a dark preincubation in estuarine water, within a headspace of $N_2$ to $CO_2$ gas ratio of 4:1, maintained at 22 °C for a period of 5 days. Microcosms were amended with various combinations of inorganic carbon ($C_i$, $CO_2$ in gas-phase and bicarbonate in liquid-phase), organic carbon ($C_o$, 5 mM mixture of acetate, glucose, and propionate), nitrate (0.30 mM), and sulfide (0.05 mM) along with the basic additions of ammonium (0.10 mM), sulfate (0.75 mM), disodium phosphate (0.10 mM), and HEPES buffer. To delineate the contributions of chemolithoautotrophs and heterotrophs, $^{13}C_i$ (98–99% labeling percent, Sigma-Aldrich, USA) and $^{13}C_o$ (99% labeling percent, Cambridge Isotope Laboratories, USA) were used in SIP incubation experiments (Supplementary Fig. 2). No copper ligand complex was supplied in trace elements to prevent the selective enrichment of denitrifiers with NirK[74]. During incubation, 700 μL of liquid was routinely collected for chemical determination, while 400 μL of headspace was taken for $N_2$, $N_2O$ and methane analysis. Each microcosm was incubated up to three generation times (~10–12 days). During medium replacement after each cycle, a custom-designed hose with on-off switch was connected between microcosm vessel outlet and a washing bottle containing deoxygenated phosphate buffer

solution to maintain the strict anoxic environment[22]. Following a negative pressure exchange, fresh isotope substrates slowly flowed into the incubation bottle, and the headspace was simultaneously refilled with helium gas.

## Function identification and community isotope response correction

Approximately 3 μg of DNA was mixed by vortexing with CsCl solution to achieve a standard density of $1.725\,g\,mL^{-1}$. Isopycnic ultracentrifugation was performed with a VTi 65.2 vertical rotor (Beckman, USA) at 20 °C, $174,000 \times g$ for 55 h. DNA from each fraction of the gradients was collected through sterile water displacement, resulting in 14 individual fractions. Buoyant density was determined by calculating the calibrated refractive index measured with a AR200 refractometer (Reichert, USA). To recover the trace DNA mixed with CsCl solution, DNA purification was performed through precipitation using glycogen (Roche, USA) and polyethylene glycol 6000 (Sigma-Aldrich, USA). Both 16S rRNA and functional genes were quantified across varying DNA buoyant densities in each microcosm. The identified heavy fractions were pooled for subsequent DNA extraction and high-throughput sequencing.

We introduced a approach to identify and quantify functions of individual microbes within a community and their roles in the cycling of all three elements. The fundamental principle of this approach involves comparing variations in the relative abundance of a given microorganism in heavy DNA fractions across different treatments, where the disparity in treatments serves as an indicator of the microbe's metabolism in each of the treatments. For instance, a significant increase in the relative abundance of species A in the heavy fractions of a +C + N + S microcosm compared to a +C + N microcosm would lead to the conclusion that species A is capable of sulfide oxidation. Through simultaneous assessments of community isotope response, the functions of each taxon, encompassing carbon (inorganic carbon fixation or OM assimilation), nitrogen (nitrate reduction), and sulfur (sulfur oxidation) metabolisms, were determined within the complex community. For a given taxon, we performed following comparisons: (1) microbes are considered chemolithoautotrophs if they increase in relative abundances in heavy fractions in both $(^{13}C_i + {}^{12}C_o) + N + S$ compared to $(^{12}C_i + {}^{12}C_o) + N + S$ and $^{13}C_i + N + S$ compared to $^{12}C_i + N + S$ treatments; (2) microbes are considered heterotrophs if they increase in relative abundances in heavy fractions of both $(^{12}C_i + {}^{13}C_o) + N + S$ compared to $(^{12}C_i + {}^{12}C_o) + N + S$ and $(^{12}C_i + {}^{13}C_o) + N$ compared to $(^{12}C_i + {}^{12}C_o) + N$ treatments; (3) microbes are capable of nitrate reduction and sulfur oxidation if they increase in relative abundances in $+C_i + N + S$ compared to $+C_i$ microcosms; (4) microbes have the capacity to oxidize sulfide if they increase in relative abundances in $+C + N + S$ compared to $+C + N$ microcosms. Here, an increase in $(^{13}C_i + {}^{12}C_o) + N + S$ compared to $(^{13}C_i + {}^{12}C_o) + N$ and $(^{12}C_i + {}^{12}C_o) + N + S$ treatments suggests a chemolithoautotroph capable of sulfide oxidation, while an increase in $(^{12}C_i + {}^{13}C_o) + N + S$ compared to $(^{12}C_i + {}^{12}C_o) + N + S$ and $(^{12}C_i + {}^{13}C_o) + N$ treatments suggests a heterotroph capable of sulfide oxidation.

## 16S rRNA gene sequencing

After DNA extraction, the V3–V4 regions of bacterial 16S rRNA gene were amplified by using the primer set 341 F (CCTACGGGNGGCWG-CAG) and 805 R (GACTACHVGGGTATCTAATCC)[75]. Amplicon sequencing was performed on an Illumina MiSeq sequencer to generate $2 \times 300$ bp paired-end reads, with raw reads ranging from 54,985 to 104,902 per sample. Sequences were clustered into OTUs at the 99% sequence similarity level using Mothur (v1.34.3) (https://github.com/mothur/mothur/releases/). Representative sequences were taxonomically assigned by searching against SILVA (v132) rRNA reference database (https://www.arb-silva.de/) and NCBI nucleotide collection (nt) database (https://www.ncbi.nlm.nih.gov/) using the Naïve Bayesian algorithm in RDP classifier. For the enrichment sample, near full-

length 16S rRNA gene was amplified using primer set 27 F (AGRGT-TYGATYMTGGCTCAG) and 1492 R (RGYTACCTTGTTACGA CTT)[76]. After blunt-end-ligation, the SMRTbell library (PacBio, USA) was constructed and then purified for sequencing via dedicated PacBio Sequel II 8 M cells. Circular consensus sequencing (CCS) reads were generated using a PacBio Sequel II sequencer.

## Metagenomic assembly and genome binning

DNA samples from heavy fractions following incubations with $^{13}C_i$-/$^{13}C_o$-labeled compounds and $^{12}C_i$-/$^{12}C_o$-natural abundance compounds, raw sediments, short-term isotope-inoculated sediments, and ENR_C4 enrichments were collected for metagenome sequencing. Trace amounts of heavy DNA in SIP experiments were amplified by using the multiple displacement amplification technique[77]. Chimeras were removed based on the general approach of Chen and colleagues[78] using Phi29 DNA polymerase and S1 nuclease. Metagenomic sequence data for each sample were generated using Illumina HiSeq X Ten instruments (PE150 mode) with an insert size of ~450 bp. Raw reads with adapters, duplicated sequences, and low-quality nucleotides at both ends (average phred value < 20) were preprocessed using homemade scripts[79]. Metagenomic sequences for each sample were de novo assembled using SPAdes[80] (v3.15) with parameters "-k 21, 33, 55, 77, 99, 127 --meta". The sequence depth was calculated by mapping quality reads to the assembled scaffolds using BBMap (https://sourceforge.net/projects/bbmap/) with the following parameters: minid=0.97, local= t. Scaffolds with length ≥2,500 bp were used for binning using MetaBAT2[81], MaxBin2[82], and CONCOCT[83]. Default parameters were applied and both tetranucleotide frequency and sequence depth were considered. The best MAGs were refined using DAS Tool[84] (v1.1.1). MAGs assembled from different samples were combined and dereplicated using dRep[85] to generate representative bins. To optimize the genome quality, clean reads were recruited using BBmap with the same parameters as above. Genome bins were subsequently reassembled by SPAdes with following options: "--careful -k 21,33,55,77,99,127". All bins were manually curated by removing scaffolds with multiple copies of marker genes and abnormal sequence depth. A total of 63 representative genomes with relatively high completeness (>80%) and low contamination (<5%) evaluated by CheckM[86] were retained for further analysis.

Gene calling was conducted for each MAG using Prodigal (v.2.6.3) with the "-p single" option. Functional annotations were performed by searching predicted genes against the KEGG, NCBI-nr, carbohydrate-active enzymes, and eggNOG databases using DIAMOND[87] with $1e^{-5}$ as the E-value cutoff. Predicted genes were also used to search against the KOfam[88] database using kofamscan v1.3.0 for metabolic pathway reconstruction. Genes associated with heterotrophic lifestyle and environmental adaptation were validated by BLASTing against a recent dataset[40]. Relative abundance of MAG was determined by dividing the sum of coverage for each scaffold in a single MAG by the sum of coverage for all scaffolds in the belonging sample.

## Genome- and SIP-informed enrichment of F-SOHDs

A combination of bioreactors and serum bottle cultures were employed to enrich for F-SOHDs. To sustain the relatively high biomass, the cultivation was first conducted in a custom-designed sequencing batch reactor (SBR). Approximately 120 mL of sediments was inoculated into a 950 mL of the SBR for a five-month (-152 days) continuous enrichment (Fig. 4a). Considering the functional flexibility of F-SOHDs, which are able to grow under both sulfide and sulfide-free conditions, the medium was periodically alternated between HED and CNS conditions. Here, HED refers to culture medium containing OM (composite OM, including glucose, propionate, acetate, and fermentative DL-lactate) and nitrate, while CNS represents the same medium supplemented with additional sulfide. During the enrichment process, sterile deionized water was used instead of filtered river water, along

with essential salts and other necessary nutrients (see media composition in the "Isotope incubation experiments" section), to maintain greater control and prevent impurities from interfering with the growth of F-SOHDs. Throughout long-term operation, pure argon gas was continuously bubbled from the bottom of the reactor to expel $CO_2$ released through microbial respiration, thereby preventing the undesired growth of chemolithoautotrophs.

After 152 days of cultivation in a SBR, we switched to a dilution-transfer approach to further purify F-SOHDs. This process was carried out within 120 mL serum bottles sealed with butyl rubber stoppers. Approximately 20 mL of preliminary enrichment from the SBR was inoculated into a 60 mL of CNS medium and following growth was transferred to flasks provided with HED medium. Following growth in HED medium, 2 mL of culture was inoculated in triplicate into anaerobic Hungate tubes containing 18 mL of CMD medium to test for chemolithoautotrophic metabolism. This medium replicates the optimal growth conditions for the well-known chemolithoautotrophic sulfide-oxidizing denitrifier, *Thiobacillus denitrificans* ATCC 25259 (450 T2 medium), where sulfide, nitrate, and bicarbonate are provided. Viable cells were counted using Vi-Cell XR coulter (Beckman, USA) with trypan blue dyeing, and the results were validated through FACSCalibur flow cytometry analysis (Becton Dickinson, USA). For treatments with low cell numbers, the one exhibiting a lowest sulfide oxidation rate was regarded as a negative response to the CMD condition. It was then chosen and inoculated into the fresh CNS and then HED medium at a volume dilution ratio of 1:6 for the next enrichment cycle. Other treatments showing a positive response were discarded in the subsequent enrichment. After 67 cultivation cycles (~460 days), two stable enrichments named ENR_C4 and ENR_U2 were ultimately acquired. For these two enrichments, various sulfur sources, including sulfide, zero-sulfur, thiosulfate, and sulfite, were separately provided for kinetic tests under both heterotrophic and chemolithoautotrophic conditions.

### Physiological experiments

Experiments were conducted with seven different cultures to determine the effect of F-SOHDs on sulfur and nitrogen cycles under controlled conditions through tracking the biochemical transformation and greenhouse gas emission. The cultures were: HD, an enrichment culture containing conventional heterotrophic denitrifiers with no sulfur-oxidation activity (previously enriched from sediments in laboratory); CD, a pure culture of the chemolithoautotrophic sulfur-oxidizing denitrifier *T. denitrificans* ATCC 25259 (purchased from The Global Bioresource Center); ENR_U2 and ENR_C4, two enrichment cultures dominated by F-SOHDs with incomplete and complete sulfide oxidation capacity; and CE_H, CE_E, and CE_L contain mixtures of the HD culture with the CD *T. denitrificans* in ratios of 3:1, 1:1, and 1:3 (synthetic microbial communities). To establish environments with rich and limited OM, we maintained carbon/nitrogen ratios of 3.52 and 0.70. An automatic sampling needle, and two precalibrated ultra-trace $N_2O$-100 and SULF-100 microsensors (Unisense, Denmark) were used for continuous monitoring the concentration of soluble $N_2O$ and $H_2S$ in liquid phase. Greenhouse gases ($N_2O$, $CO_2$, and $CH_4$) in the headspace were analyzed using an Agilent 7890B gas chromatography (USA) equipped with two series-wound packed columns (J&W G3591-81121 and G3591-81004). The detection limits were 0.8 ppm for $CH_4$, 0.2 ppm for $CO_2$, and 0.05 ppm for $N_2O$. Dinitrogen gas was measured using a separate Agilent 7890 A gas chromatography (USA) equipped with a thermal conductivity detector and J&W G3591-81121 packed column, with a detection limit of 2.6 ppm. The completion extent of denitrification process was calculated with the following equation:

$$N'_{com} = \frac{(\Delta NO_3^- - \Delta NO_2^- - \Delta N_2O - \Delta NH_4^+) \times 5 + \Delta NO_2^- \times 2 + \Delta N_2O \times 4 + \Delta NH_4^+ \times 8}{C^i_{NO_3^-} \times 5}$$

(1)

where $N'_{com}$ denotes the completion extent of denitrification. $\Delta NO_3^-$ indicates the net nitrate concentration consumed, and other nitrogen compounds after triangle indicate the net concentration produced. $C^i_{NO_3^-}$ denotes the initial nitrate concentration, and its value multiplied by 5 represents the theoretical total electron demand for complete reduction of nitrate to dinitrogen gas.

The proportion of electrons contributed by sulfide to facilitate the complete denitrification was calculated with the following equation:

$$P^S_{den} = \frac{\Delta S^0 \times 2 + \Delta S_2O_3^{2-} \times 4 + SO_3^{2-} \times 6 + \Delta SO_4^{2-} \times 8}{C^i_{NO_3^-} \times 5}$$

(2)

where $P^S_{den}$ denotes contribution of sulfide, as an electron donor to promote the complete reduction of nitrate to nitrogen. All sulfur compounds after triangles indicate the net produced concentration. The concentrations of the anions sulfate, sulfite, and thiosulfate were determined using an ICS-3000 ion chromatograph (Dionex, USA) equipped with an IonPac AS11-HC analytical column. Sulfide was measured by the standard methylene blue method at a wavelength of 665 nm[89], while zero-valent sulfur was quantified through ultraviolet-visible spectrophotometric analysis (Shimadzu, Japan) at a custom wavelength of 282 nm[90].

### Phylogenetic tree and gene analysis

The phylogenomic tree was inferred based on 31 conserved protein markers[91] identified by AMPHORA2[92]. Genomes belonging to the phylum *Planctomycetota* were downloaded from GenBank (74 available in July 2021) and were chosen as outgroup. For each marker, multiple sequence alignment was performed using MUSCLE[93] (v3.8.31) and poorly aligned regions were trimmed using TrimAL with options "-gt 0.95 -cons 50". Maximum-likelihood-based phylogeny was reconstructed based on concatenated markers using IQ-TREE[94] with ultrafast bootstrapping (-bb 1000 -m LG + R6). The best-fitting model was determined by ModelFinder[95].

### Reporting summary

Further information on research design is available in the Nature Portfolio Reporting Summary linked to this article.

## Data availability

The sequence data of 16S rRNA amplicons, metagenomes, and metagenome-assembled genomes generated in this study have been deposited in the NCBI Sequence Read Archive under Bioproject PRJNA1108931. The detailed information of genomes is provided in Supplementary Table 1. Relevant data are available within the paper and source data files. Source data are provided with this paper.

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

## Acknowledgements

We thank the School of Life Science, Harbin Institute of Technology for technical assistance in microbial analysis. This research was financially supported by the National Natural Science Foundation of China (52321005, A.J.W.), the National Natural Science Foundation of China (52076063, C.C.; 32170014, Z.S.H.; 52100035, X.J.X.; 52400025, W.W.; 52300155, X.T.W.), China Postdoctoral Science Foundation (2023M740917, X.T.W.; 2024M754204, W.W.), the National Key Research and Development Program of China (2023YFC3207203, C.C.), and State Key Laboratory of Urban Water Resource and Environment (Harbin Institute of Technology) (2023DX04, C.C.).

## Author contributions

B.S., Z.-S.H., and C.C. conceived the research and prepared manuscript. B.S., R.C.Z., W.W., Z.D.L., X.Y.P., and X.T.W. contributed to samplings and incubation experiments. L.Z. and Y.R. conducted the stable-isotope probing experiments. Y.-G.X. performed the bioinformatic analyses and data discussion. B.L., X.J.X., L.Z., X.Z., X.H.W., and D.F.X. assisted in the figure visualization and molecular experiments. A.J.W., D.J.L., N.Q.R., Z-S.H., and C.C. jointly supervised the project. D.E.C. and B.P.H. greatly contributed to the writing, discussion, review, and editing of the manuscript.

## Competing interests

The authors declare no competing interests.

## Additional information

¹State Key Laboratory of Urban Water Resource and Environment, School of Environment, Harbin Institute of Technology, Harbin 150090, PR China. ²Chinese Academy of Sciences Key Laboratory of Urban Pollutant Conversion, Department of Environmental Science and Engineering, University of Science and Technology of China, Hefei 230026, PR China. ³College of Life Sciences, Huaibei Normal University, 235000 Huaibei, PR China. ⁴Department of Microbiology, Key Lab of Microbiology for Agricultural Environment, Ministry of Agriculture, College of Life Sciences, Nanjing Agricultural University, Nanjing 210095, PR China. ⁵Jangsu Provincial Key Lab for Solid Organic Waste Utilization, Key Lab of Organic-based Fertilizers of China, Nanjing Agricultural University, Nanjing 210095, PR China. ⁶State Key Laboratory of Urban Water Resource and Environment, School of Civil and Environmental Engineering, Harbin Institute of Technology Shenzhen, Shenzhen 518055, PR China. ⁷School of Civil and Transportation, Hebei University of Technology, Tianjin 300401, PR China. ⁸School of Environmental Engineering, Wuhan Textile University, Wuhan 430073, PR China. ⁹School of Environmental Science and Engineering, Huazhong University of Science and Technology, Wuhan 430074, PR China. ¹⁰Department of Mechanical Engineering, City University of Hong Kong, Tat Chee Avenue, Kowloon, Hong Kong, PR China. ¹¹Nordcee, Department of Biology, University of Southern Denmark, Odense, Denmark. ¹²School of Life Sciences, University of Nevada, Las Vegas, Las Vegas, NV 89154, USA. ¹³Nevada Institute of Personalized Medicine, Las Vegas, NV 89154, USA. ✉e-mail: hzhengsh@ustc.edu.cn; cchen@hit.edu.cn

