## [Peer Review File · Nature Communications]

Versatile nitrate-respiring heterotrophs are previously concealed contributors to sulfur cycle

Corresponding Author: Professor Chuan Chen

Version 0:

Reviewer comments:

Reviewer #1

(Remarks to the Author)

This manuscript found the important contribution of heterotrophic denitrifiers on N₂O reduction in natural ecosystem, especially in estuaries environments. Based on authors data, heterotrophic denitrifiers seemed to exert a more significant role compared to chemolithoautotrophic denitrifiers, which was completely beyond folks's traditional understanding. These findings help folks further comprehend the interaction between C, N and S cycles. Before its publication, the followings should be addressed:

1. Line 88-89, why authors sampled the Estuarine sediments at the junction of the Songhua River and effluent from municipal treatment? And in Fig.S1 the detailed sample site was not provided.
2. line 91, what was the quality of 20 mL of surface sediments?
3. In Fig.1d, the N amounts were balanced or not?
4. line 171-177, please provide more details on this deduction of heterotrophs may dominate sulfur oxidation and denitrification in this ecosystem.
5. line 222-229, author mainly focused on the contribution of denitrification processes, and they also found DNRA process in the sediments. Thus, how about the DNRA process contribution on both NO₃⁻ reduction and S₂⁻ oxidation?
6. line 299-303, more details information are required to support authors's deduction.
7. line 351-353, please provide more similar sites, I do not believe that only 3 similar sample were enough for supporting authors's opinion.

Reviewer #2

(Remarks to the Author)

The study by Shao et al., explores the presence of facultative sulfur-oxidizing heterotrophic denitrifiers, abbreviated by the authors as F-SOHDs, in estuary sediments of the Songhua River System. Specifically, their study employed DNA-SIP experiments, in combination with metagenomic sequencing and enrichment cultures to identify F-SOHD members in a mixed microbial community comprising chemoorganoheterotrophs and chemolithoautotrophs. Using this range of techniques, they identified environmental genera affiliated with *Thauera*, *Pseudomonas*, and *Azoacus* as likely F-SOHDs in their samples. Lastly, they demonstrate using F-SOHD enriched bioreactors, that such members are capable at better suppressing N₂O emissions than their heterotrophic denitrifying counterparts. In turn, this led to the suggestion that F-SOHDs could be useful in industrial settings to mitigate N₂O emissions.

As someone who has previously studied F-SOHDs, which I refer to as chemolithoheterotrophs, I find the work to be of high value as it expands on the potential for chemolithoheterotrophy in the environment. Moreover, the authors employed an impressive range of tools to identify these microbes in situ. Overall, I find the results fairly compelling, albeit the manuscript is

not well-rounded at times, and does tend to overstate claims. Namely, the introduction did not provide the reader with state-of-the-art background information concerning chemolithoheterotrophs. The results I generally find compelling, although I have some points that need further clarification, and the authors should discuss some of the discrepancies in their data. While I don't disagree with all the points raised in the discussion, it does need some more work/reframing, to present a more well-rounded case for chemolithoheterotrophy in the environment. In addition, some errors in their estimates of Gibbs free energy need to be addressed in the discussion. The method section was at times thorough, but other parts were fairly vague, to the point that the reader would not be able to reproduce the experiment without additional clarification. Ending on an optimistic note, the figures are nicely polished, and the data/conclusions (while overstated at times) appear generally sound.

Comments

Lines 40-41: I find this statement to be a little vague, namely because heterotrophic denitrifiers operate in environments where sulfide is present. But what the authors could specify is that sulfide inhibits a key enzymatic reaction that catalyzes the reduction of N₂O to N₂. Hence, heterotrophic denitrifiers are sensitive unless sulfide is removed, either by cooperation with chemolithoautotrophs or with chemolithoheterotrophs. This type of denitrifying community structure is indeed what we find in organic and sulfide-rich ecosystems.

Line 43: This statement comes across as a little too strong as their role in the environment, including aquatic ecosystems, has been noticed by some and even empirically demonstrated (Sorokin 2003; Boltyanskaya et al. 2007; Callbeck et al. 2019; Wang and Shao 2021). As I mentioned above, the value of the study is not in that it discovered sulfide oxidation by heterotrophs in the environment, it is that it demonstrates that numerous microbes in estuary sediments could thrive via a chemolithoheterotrophic metabolism greatly expanding its potential. I would therefore consider re-framing this part.

Lines 48-51: I agree with this statement, but this largely repeats the findings of Callbeck et al., 2019, which I detail a little later in this review. Moreover, their justification of this statement based on their Gibbs free energy calculations appears to be flawed (see later discussion).

Lines 51-53: I'm in favor of this work generally, but this is an overstatement. Firstly, environmental examples of sulfide oxidation by heterotrophs exist, thus the use of "paradigm shift" is over the top. Secondly, I understand that if you work on N₂O you can make a link to climate change, however, to say that his work has "profound implications" for climate change is a stretch too far. For example, it is not the case that the authors have tested/modeled their reactors under various climate change scenarios. In my view, the work is a good piece because it advances the idea that chemolithoheterotrophs form a persistent part of the microbial community in an organic matter-rich estuary.

Lines 62-64: For clarity, denitrification is one of two major processes, nitrification being the other. The statement makes it seem as though denitrification is the only one.

Lines 65-68: Fully agree, for a comprehensive review of sulfur cycling and its influence over the nitrogen cycle in oxygen minimum zones please see (Callbeck et al. 2021).

Lines 71-73: The authors only superficially mention that it impacts N₂O but how is this the case? Some have speculated that sulfide also inhibits the last step of denitrification, as sulfide acts as a Cu-chelator.

Lines 68-70: Certainly, chemolithoautotrophs are well-recognized, but work by Callbeck et al., 2019 has shown that both chemolithoautotrophs, and chemolithoheterotroph (aka. F-SOHD) can co-exist to oxidize sulfide during a sulfidic event off the coast of Peru.

Line 72: I understand the point the authors are making, however, for clarification sulfide is toxic to all denitrifiers, including nitrate-reducing chemolithoautotrophs, chemolithoheterotrophs and organoheterotrophs. The reason for this I explain a little later in the discussion section below.

Lines 73-75: This statement, which is effectively the premise of the manuscript, is simply not up to date with the state-of-the-art. As I mentioned above, it's not as though we know nothing about this topic of F-SOHD/chemolithoheterotrophs. The authors have partly relayed this information in the discussion (on lines 368-370), but choose not to interject this information here or above when background information is needed on this specialized metabolism. Again, the work is valuable not because it is the first to discover F-SOHD, but because it expands the potential for this metabolism to operate in the aquatic realm. The authors should consider reworking this paragraph to include the state-of-the-art, and perhaps frame the premise in a slightly different way.

Lines 80-83: To be specific, the study identifies new F-SOHDs at a sediment site. Somewhat misleading is the use of "ubiquitous" as this implies that the authors provided evidence of F-SOHDs widespread distribution across multiple sampling sites, although their study site is positioned at one location (45°49'12.81"N, 126°43'22.51"E). Again, the authors have a habit of overstating claims, when in fact, there is no need.

Lines 82-85: I find this overly broad and introduces a bit of speculation, especially on the climate change front; can the authors be a bit more specific?

Lines 98-101: If I have this right, this would amount to roughly two-thirds of the original 300 M of nitrate being recovered as N₂ and NH₄⁺. But roughly one-third is not recovered, hence, both complete and incomplete denitrification are evident in this

experiment.

Line 108: The with and without sulfide addition experiments are somewhat confusing. I understand that the authors added additional sulfide, but it should be perhaps reiterated that sulfide is going to be produced in the background, unless the authors removed the sulfate for sulfate reducers, in incubations like C+N, or Ci, or Co etc...

Lines 117-121: The increases of 1.3% and 3.8% don't seem to be substantial for *Thauera* and *Pseudomonas*. Thus, can the authors assume that their heterotrophs according to these values? And in the second part, how much did these members decrease when amended with $^{13}\text{C}_i$? In light of chemolithoautotrophic abundances increasing with the addition of ^{13}C , could this not artificially dilute such heterotrophs like *Thauera* and *Pseudomonas*, if it's based on relative abundances?

Lines 121-122: Can the authors comment on why *Thauera* comprised a large fraction of the microbial community in the experiment void of sulfide (C+N) when compared to the C+N+S experiments? It would appear that this member is sensitive to sulfide inhibition. In addition, an increase of 0.7% is practically negligible. Moreover, it appears that *Pseudomonas* shows a different pattern that is opposite to your statement. Also in Fig. 2d, *Pseudomonas* and *Azoarcus* both have $^{13}\text{C}_i$ which is higher than the control in grey. Hence, this goes against the idea that no ^{13}C -DIC is incorporated. Part of the problem the authors might encounter in their analysis is that heterotrophs can also assimilate ^{13}C -DIC (e.g. Erb 2011), which might complicate their analysis, or at least require a note.

Lines 125-129: *Thiobacillus* increased in Fig. 2d (C+N) when sulfide was not present, which runs counter to their chemolithoautotrophic lifestyle. And also what about other chemolithoautotrophic members such as Chromatiales (M5.bin31) and *Sulfurivermis* (M5.bin64) shown in Fig. 3?

Lines 130-154: It seems that this method distinguishes *Thiobacillus*, but what about other chemolithoautotrophic members such as Chromatiales (M5.bin31) and *Sulfurivermis* (M5.bin64) shown in Fig. 3, where do they position in Fig. 2i? In addition, organisms like *Dechloromonas*, which according to Fig. 3 is an F-SOHD, are positioned in the bottom left of Fig. 2i alongside *Thiobacillus* the chemolithoautotroph. I also have difficulties in reconciling the very negligible increases the authors reported on lines 118 and 121 and how this results in very clear patterns for *Thauera* and *Pseudomonas* in Fig. 2i. I think the authors should add that this method might not be perfect at disentangling all F-SOHDs from other community members, but it may work for more active microbes.

Lines 138-142: The exception would be *Arcobacter peruensis*, which reduces nitrate, oxidizes sulfide, and simultaneously assimilates/oxidizes organic matter into biomass for energy/growth (Callbeck et al. 2019). *Arcobacter peruensis* was also abundant in sulfidic and organic matter-rich shelf waters alongside chemolithoautotrophs like SUP05.

Line 142-144: In the methods section, on lines 488-490, the authors state that "microbes have the capacity to oxidize sulfide if they increase in relative abundances in +C+N+S compared to +C+N microcosms", but the issue with *Thauera* is that its abundance decreases, and hence, by their logic above is not a sulfide oxidizer. The authors could further clarify this.

Line 171: The authors coin the new acronym facultative sulfur-oxidizing heterotrophic denitrifiers (F-SOHDs) for a metabolic process that has been described previously as chemolithoheterotrophy (Callbeck et al. 2019). For consistency's sake with previous work, why not term F-SOHDs as chemolithoheterotrophs which include microbes that oxidize sulfide, and reduce nitrate while simultaneously utilizing organic carbon as an energy/carbon source for growth? This would reduce the number of acronyms in the literature, and make for a more comparable term to chemolithoautotrophy, which the authors readily apply in their manuscript.

Lines 204-207: The authors mention *Arcobacter*, but it was later been shown to grow as a chemolithoheterotroph (Callbeck et al., 2019). Hence, *Arcobacter peruensis* had the capacity to assimilate organic carbon, reduce nitrate, and oxidize sulfide, much like what the authors describe in their paper for *Thauera* and other members.

Lines 208-213: I checked the literature on some of the closest affiliated microbes in Figure 3 (see below), and some of them have capacities for CO_2 fixation, despite being labeled as having no capacity in Fig. 3c.

- *Azoarcus* (see https://www.kegg.jp/pathway/aza00710+AZKH_p0225)
- *Ralstonia pickettii* DTP0602 (<https://www.kegg.jp/pathway/rpj00710+M00166>)
- *Pseudomonas yangmingensis* (Wong and Lee 2014)

Thus, while I believe that some are probably chemolithoheterotrophs, the authors should stipulate some caveats and unknowns here.

Lines 222-225: As mentioned above the authors should apply some caution here. Moreover, the authors have not measured the nitrate turnover rates of the specific members, thus to say that this cluster is more biogeochemical relevant is rather speculative. It's fair to say that strong potential for chemolithoheterotrophy exists at their study site.

Lines 244-249: I find it difficult to take the position that *Thauera* is sensitive to sulfide. If the authors take this position then they have to reconcile in their discussion that both nitrate-reducing chemoorganoheterotrophs and chemolithoheterotrophs are sensitive to sulfide. Presently, they hold only the former to be true. However, I would suggest that the authors explain this differently. For example, sulfide-oxidizers like *Arcobacter*, SUP05 and *Sulfurovum* can co-exist in the presence of sulfide however *Arcobacter* and *Sulfurovum* tend to dominate over SUP05 when sulfide concentrations exceed 10 μM (Callbeck et al., 2021). So could it not be the case that *Pseudomonas* outcompetes *Thauera* at a certain sulfide threshold, thereby lowering its contribution to the overall community (based on relative abundances). Taking this position means that *Thauera*

is not necessarily sensitive but rather outcompeted under certain environmental conditions by others.

Lines 266-267: This is a bit confusing here as it is not clear why being a heterotrophic denitrifier leads to N₂O production, and also why chemolithoautotrophs do not. In my mind it's not so clear cut, for example, SUP05 identified in Peru can perform complete denitrification to N₂ (Callbeck et al. 2018), while other SUP05 species in the Saanich Inlet can only do nitrate reduction to N₂O (Walsh et al. 2009). Might also be worth mentioning here that *Arcobacter peruensis* is an exception, plus others exist as mentioned earlier.

Lines 285-287: This line contradicts the author's statement above that heterotrophic members like *Thauera* could be sensitive to sulfide.

Lines 305-321: I agree with the authors that DNA-SIP technology is important to unraveling microbial ecology-related questions. But I find a dedicated paragraph to this in the opening discussion is a little excessive, as the method is in principle the same with the difference being that the authors did an additional correlative analysis (Fig. 2i, g). In addition, organisms like *Dechloromonas*, which according to Fig. 3 is an F-SOHD, are positioned in the bottom left of Fig. 2i alongside *Thiobacillus* the chemolithoautotroph. Hence, this method is not perfect, and the authors (if they decide to keep this piece) will need to address the caveats. As a recommendation, I would instead suggest the authors shorten or drop to make way for a more detailed discussion of the specific chemolithoheterotroph members they identified in their study, which was lacking in my opinion (see below).

Line 323: I would add, "Epsilonproteobacteria and the Gammaproteobacteria SUP05 clade"

Line 331: There are more pertinent studies to reference here that discuss sulfidic events in marine environments, which is summarized in the review by Callbeck et al., 2021.

Lines 332-333: I believe this to be an overstatement by the authors "...uncovered here have evolved multiple sulfur oxidation pathways to cope with sulfide stress." The authors have not empirically demonstrated sulfide stress on the community and how it leads to the evolution of multiple sulfur oxidation pathways. Rather this statement could be framed as a hypothesis.

Lines 322-340: I see the importance of a wider description of sulfidic events, but this really appears as a zoomed-out introduction piece especially since the *nosZ* gene discussion below brings more detail to this idea. Instead, I think the opening of the discussion would be better served reiterating what defines a F-SOHDs/chemolithoheterotroph. Namely, chemolithoheterotrophs couple sulfide oxidation to nitrate reduction while simultaneously utilizing organic carbon as an energy/carbon source (hence the assimilation of ¹³C into biomass in your DNA-SIP experiments). Next, I would reiterate the key putative chemolithoheterotrophs mentioned in the results section and Fig. 3. Subsequently, the authors need to spend more time describing what is already known about these species in the literature, so that the reader can assess what new information is added in this study. For example, my cursory inspection of some of the literature finds that some of these members (identified in Fig. 3) are known sulfide oxidizers like *Pseudomonas stutzeri* (Li et al. 2022). *Thauera* is well-studied and has been isolated from sulfide-rich environments and bioreactors before, implicating it already with a role as a putative sulfide oxidizer. Others are capable of fixing CO₂ autotrophically such as *Azoarcus* (see https://www.kegg.jp/pathway/aza00710+AZKH_p0225), *Ralstonia pickettii* DTP0602 (<https://www.kegg.jp/pathway/rpj00710+M00166>) and *Pseudomonas yangmingensis* (Wong and Lee 2014). I recommend the authors incorporate a much more thorough discussion of the key members they identified.

Lines 345-346: I agree that sulfide has an influence over N₂O production, but the statement here that sulfide only affects heterotrophic denitrifiers and not denitrifiers more broadly – chemolithoautotrophs included – is not evidenced in the literature. Much of this discussion has taken place in the review by Callbeck et al., 2021, but to reiterate their points sulfide is a chelator of copper, and the *nosZ* gene is a copper-rich metalloenzyme. The sequestration of copper by its abiotic reaction with sulfide will induce copper limitations that result in nitrate reduction terminating at N₂O in microbial cultures (Granger and Ward 2003; Manconi et al. 2006; Moffett et al. 2012; Felgate et al. 2012). The accumulation of N₂O has therefore been linked to copper limitations induced by sulfide, which would affect nitrate-reducing bacteria (with a *nosZ*) irrespective of whether they are chemolithoautotrophs, chemolithoheterotrophs or heterotrophs.

Lines 346-349: I agree with this statement, but as above this is not exclusively a heterotrophic denitrifier issue but will rather affect denitrifiers broadly.

Lines 349-353: Indeed, sulfide-driven denitrification is likely responsible for major N₂O hotspots (Naqvi et al. 2000, 2006; Arevalo-Martinez et al. 2015; Kock et al. 2016; Bourbonnais et al. 2017; Arévalo-Martínez et al. 2019), which likely occur as a result of the mechanism outlined above. However, the difficulty I have with this line is the authors claim that these N₂O/H₂S-rich systems are dominated by heterotrophic denitrifiers, when in fact, the cited studies do not make that claim (mainly because they also pre-date large community metagenomic surveys). Secondly, in light of more recent analyses of these same environments, such as Peru, where we have major N₂O hotspots along with the presence of sulfide, the microbial community is comprised of a consortium of nitrate-reducing chemolithoautotrophs, chemoorganoheterotrophs and chemolithoheterotrophs. Thirdly, from a logical perspective, if such systems have high organic matter, sulfide, and nitrate, wouldn't they offer the perfect ecological niche for chemolithoheterotrophs to thrive? To that point, Callbeck et al. 2019, and this study enriched for chemolithoheterotrophs from exactly these environments.

Lines 353-356: I generally agree with this line, but perhaps clarify that this occurs because chemolithoheterotrophs can remove sulfide and therefore remove its ability to chelate copper, enabling N₂O reduction to N₂ to continue (via the Cu-rich

nosZ).

Lines 357-359: I agree, but it would make sense to mention that sulfide provides an additional electron donor (on top of organic matter) to reduce available nitrate to N₂ rather than to N₂O.

Lines 356-363: I agree with this statement, but it should be pointed out that chemolithoautotrophs have the same advantage (as your work also demonstrates in Fig. 5g). In other words, this feature is not unique to chemolithoheterotrophs.

Lines 363-365: As I stipulated above, this is complicated and not so clear-cut. The environment where *Arcobacter peruensis* was identified, was from a sulfidic event off the coast of Peru (Callbeck et al., 2019). Here, the environment was defined as being sulfide-, N₂O-, and organic matter-rich. Hence, the conditions where we find chemolithoheterotrophs are also environments where relatively high N₂O concentrations persist. To better frame this one could say that chemolithoheterotrophs/chemolithoautotrophs might contribute to suppressing some N₂O in sulfidic environments that are associated with high N₂O production rates.

Lines 368-370: I think the introduction, results, and discussion would be improved if the authors made a greater attempt to draw parallels between the microbes identified here, and the recently described *Arcobacter peruensis* as well as the other ones mentioned earlier. I found it somewhat unfortunate that the authors when discussing organic matter and sulfide-rich environments, failed to mention *Arcobacter peruensis* in that context, but rather discussed chemolithoautotrophs like SUP05.

Lines 370-375: Could the authors be a little more specific? For example, what pathways did they identify in these microbial genomes, that made them comparable to the chemolithoheterotrophs described here?

Lines 375-378: The authors did not touch on what features make chemolithoheterotrophs more adaptable over chemolithoautotrophs, so can they be more specific? Moreover, it appears that the authors have mixed up the Gibbs free energy values from their figure. According to the authors Figure 6, the authors have calculated a Gibbs free energy of -494 and -579 for F-SOHDs related processes (rxns. C1 and C2), and with chemolithoautotrophy estimated at -566 (rxn. b). But in the text, they list the Gibbs free energy of heterotrophy instead of chemolithoautotrophy. This mix-up makes it seem that F-SOHDs activity is more energetically favorable when in reality it's not according to Figure 6.

With that said the authors have overlooked an important feature that gives chemolithoheterotrophs an advantage over chemolithoautotrophs. As detailed by Callbeck et al., in 2019, the assimilation of acetate into biomass is energetically more favorable than the assimilation of CO₂ via a chemolithoautotrophic metabolism. This is because CO₂ fixation pathways require substantial amounts of ATP (Erb 2011). The favorable energetics of a chemolithoheterotrophic metabolism explain how *Arcobacter* can rapidly bloom in the sulfide- and organic-rich upper shelf waters off the coast of Peru and Namibia (Lavik et al. 2009; Callbeck et al. 2019, 2021). This is a theme the authors could consider exploring in the context of the microbes identified in their study.

Lines 378-381: Their exceptional performance hinges on their ability to rapidly bloom over chemolithoautotrophs for available sulfide, this is something to keep in mind. Overall, I find that the authors could be a bit more specific regarding their industrial application reference. Do the authors mean to use chemolithoheterotrophs to remove N₂O in sulfide, and organic matter-rich engineered systems?

Lines 381-383: Fully agree, and I think this is where you could end the discussion. I find the next line regarding climate change and its evolutionary trajectories highly speculative and rather vague.

Methods:

Line 389: What month specifically did sampling take place? What was the water depth of the study site?

Lines 390-392: Do the authors have a reference to support its biogeochemical significance to the nitrogen cycle?

Line 393: Why did the authors pick the depth between 0-20 cm? What is the nitrate and oxygen penetration depth?

Line 394: What is sulfide antioxidant buffer, please be specific.

Lines 396-398: I would convert from mg L⁻¹ to μM/mM to be consistent with the rest of the manuscript. Also, how were these nutrient concentrations measured? And by which methods? What were the detection limits?

Lines 404-407: What were the incubation temperatures? And do they differ from in situ temperatures?

Line 420: How was dinitrogen gas measured?

Lines 424: How much sediment was used for the extraction? Keep in mind that the readers need to be able to repeat the experiments.

Lines 424-428: The authors provide no mention of references either in the qPCR section or supplementary Table 2 regarding the primers. Surely, these primers were designed elsewhere and if so, the authors need to indicate this information.

Lines 428-429: Is this common to do in qPCR, because cells often have more than one gene copy? So, the cell numbers are going to be overestimated?

Lines 430-432: What qPCR chemistry was used, and how much DNA template was used? Were DNA concentrations also verified, and how? For the standards what concentration range was used in genes per uL? Was melt curve analysis also done?

Lines 434-445: Is this a completely novel technique or is there other studies that have done the same?

Lines 443-445: What was the ¹³C labeling percent for the incubation experiments?

Line 488: Change “compacity” to “capacity”

Lines 495-497: Please provide a reference to the study that designed these primers. And also indicate if these are universal primers or primers used to only amplify the Bacterial community.

Lines 497-500: Can the authors indicate the number of raw reads generated?

Lines 504-506: Please provide a reference to the study that designed these primers.

Lines 514-515: Is there a reference for this line?

Lines 551-552: Perhaps I missed this somewhere, but what is the composition of the medium in terms of salts and other nutrients. Or is the medium just filtered river water with amended substrates? Please clarify.

Lines 555-558: If CO₂ is removed, how is the system then pH buffered? Based on the principle that chemolithoautotrophs are slower growing compared to chemolithoheterotrophs I don't fully understand the reasoning behind removing CO₂ to hinder chemolithoautotrophs. As we know, chemolithoheterotrophs when supplied sulfide, nitrate, and organic matter will naturally outcompete chemolithoautotrophs for available sulfide (Callbeck et al., 2019; 2021). So, actively removing CO₂ could have downstream consequences for pH buffering in the system. Do the authors have pH measurements?

Line 580: This could be shortened to read, “Physiological experiments”.

Lines 583-585: Do the authors know which heterotrophic denitrifiers were enriched? And do they lack a capacity for sulfide oxidation and have the nosZ gene? I'm wondering how distinct they are from the sulfide-oxidizing heterotrophs.

Lines 595-596: The authors should provide more information regarding the instruments and standards used for gas analysis.

Lines 608-611: This is too vague. Please provide citations and additional information on how these sulfur species were quantified.

Figures:

Figure 6: How was Gibbs free energy calculated and where did they get the formation energies from? Under what conditions was this calculated, for example, pH, sulfide, nitrate concentrations? Reaction c2 is driven by mostly sulfide (1:5 ratio), with a little bit of organic matter, which seems more like a chemolithoautotrophy with a little bit of heterotrophy. What does this look like if the stoichiometry is a bit more balanced?

Literature cited in my review:

- Arevalo-Martinez, D. L., A. Kock, C. R. Loscher, R. A. Schmitz, and H. W. Bange. 2015. Massive nitrous oxide emissions from the tropical South Pacific Ocean. *Nat. Geosci* 8: 530–533. doi:10.1038/ngeo2469
- Arévalo-Martínez, D. L., T. Steinhoff, P. Brandt, A. Körtzinger, T. Lamont, G. Rehder, and H. W. Bange. 2019. N₂O Emissions From the Northern Benguela Upwelling System. *Geophys. Res. Lett.* 46: 3317–3326. doi:10.1029/2018GL081648
- Boltyanskaya, Y. V., V. V. Kevbrin, A. M. Lysenko, T. V. Kolganova, T. P. Tourova, G. A. Osipov, and T. N. Zhilina. 2007. *Halomonas mongoliensis* sp. nov. and *Halomonas kenyensis* sp. nov., new haloalkaliphilic denitrifiers capable of N₂O reduction, isolated from soda lakes. *Microbiology* 76: 739–747. doi:10.1134/S0026261707060148
- Bourbonnais, A., R. T. Letscher, H. W. Bange, V. Échevin, J. Larkum, J. Mohn, N. Yoshida, and M. A. Altabet. 2017. N₂O production and consumption from stable isotopic and concentration data in the Peruvian coastal upwelling system. *Global Biogeochem. Cycles* 31: 678–698. doi:10.1002/2016GB005567
- Callbeck, C. M., D. E. Canfield, M. M. M. Kuypers, P. Yilmaz, G. Lavik, B. Thamdrup, C. J. Schubert, and L. A. Bristow. 2021. Sulfur cycling in oceanic oxygen minimum zones. *Limnol. Oceanogr.* 66: 2360–2392. doi:https://doi.org/10.1002/lno.11759
- Callbeck, C. M., G. Lavik, T. G. Ferdelman, and others. 2018. Oxygen minimum zone cryptic sulfur cycling sustained by offshore transport of key sulfur oxidizing bacteria. *Nat. Commun.* 9: 1729. doi:10.1038/s41467-018-04041-x
- Callbeck, C. M., C. Pelzer, G. Lavik, and others. 2019. *Arcobacter peruensis* sp. nov., a chemolithoheterotroph isolated from sulfide and organic rich coastal waters off Peru. *Appl. Environ. Microbiol.* 85: e01344-19. doi:10.1128/AEM.01344-19

- Erb, T. J. 2011. Carboxylases in Natural and Synthetic Microbial Pathways. *Appl Env. Microbiol* 77: 8466–8477. doi:10.1128/aem.05702-11
- Felgate, H., G. Giannopoulos, M. J. Sullivan, A. J. Gates, T. A. Clarke, E. Baggs, G. Rowley, and D. J. Richardson. 2012. The impact of copper, nitrate and carbon status on the emission of nitrous oxide by two species of bacteria with biochemically distinct denitrification pathways. *Environ. Microbiol.* 14: 1788–1800. doi:10.1111/j.1462-2920.2012.02789.x
- Granger, J., and B. B. Ward. 2003. Accumulation of nitrogen oxides in copper-limited cultures of denitrifying bacteria. *Limnol. Oceanogr.* 48: 313–318. doi:10.4319/lo.2003.48.1.0313
- Kock, A., D. L. Arévalo-Martínez, C. R. Löscher, and H. W. Bange. 2016. Extreme N₂O accumulation in the coastal oxygen minimum zone off Peru. *Biogeosciences* 13: 827–840. doi:10.5194/bg-13-827-2016
- Lavik, G., T. Stührmann, V. Brüchert, and others. 2009. Detoxification of sulphidic African shelf waters by blooming chemolithotrophs. *Nature* 457: 581–584. doi:10.1038/nature07588
- Li, P., W. Yuan, Y. Huang, C. Zhang, C. Ni, Q. Lin, Z. Zhu, and J. Wang. 2022. Complete genome sequence of *Pseudomonas stutzeri* S116 owning bifunctional catalysis provides insights into affecting performance of microbial fuel cells. *BMC Microbiol.* 22: 137. doi:10.1186/s12866-022-02552-8
- Manconi, I., P. van der Maas, and P. Lens. 2006. Effect of copper dosing on sulfide inhibited reduction of nitric and nitrous oxide. *Nitric oxide Biol. Chem.* 15: 400–407. doi:10.1016/j.niox.2006.04.262
- Moffett, J. W., C. B. Tuit, and B. B. Ward. 2012. Chelator-induced inhibition of copper metalloenzymes in denitrifying bacteria. *Limnol. Oceanogr.* 57: 272–280. doi:10.4319/lo.2012.57.1.0272
- Naqvi, S. A. W., H. Naik, D. A. Jayakumar, M. S. Shailaja, and P. V Narvekar. 2006. Seasonal oxygen deficiency over the western continental shelf of India, p. 195–224. In N.L. Neretin [ed.], *Past and Present Water Column Anoxia*. Springer Netherlands.
- Naqvi, S. W. A., D. A. Jayakumar, P. V Narvekar, H. Naik, V. V. S. S. Sarma, W. D'Souza, S. Joseph, and M. D. George. 2000. Increased marine production of N₂O due to intensifying anoxia on the Indian continental shelf. *Nature* 408: 346–349. doi:10.1038/35042551
- Sorokin, D. Y. 2003. Oxidation of Inorganic Sulfur Compounds by Obligately Organotrophic Bacteria. *Microbiology* 72: 641–653. doi:10.1023/B:MIC1.0000008363.24128.e5
- Walsh, D. A., E. Zaikova, C. G. Howes, Y. C. Song, J. J. Wright, S. G. Tringe, P. D. Tortell, and S. J. Hallam. 2009. Metagenome of a versatile chemolithoautotroph from expanding oceanic dead zones. *Science* (80-.). 326: 578–582. doi:10.1126/science.1175309
- Wang, L., and Z. Shao. 2021. Aerobic Denitrification and Heterotrophic Sulfur Oxidation in the Genus *Halomonas* Revealed by Six Novel Species Characterizations and Genome-Based Analysis. *Front. Microbiol.* 12: 652766. doi:10.3389/fmicb.2021.652766
- Wong, B.-T., and D.-J. Lee. 2014. *Pseudomonas yangmingensis* sp. nov., an alkaliphilic denitrifying species isolated from a hot spring. *J. Biosci. Bioeng.* 117: 71–74. doi:https://doi.org/10.1016/j.jbiosc.2013.06.006

Reviewer #3

(Remarks to the Author)

This study identifies a previously unknown group of heterotrophic denitrifiers capable of coupling sulfur oxidation with denitrification autonomously. Traditionally, heterotrophic denitrifiers have been thought to be hampered by sulfide, which negatively affects their growth and increases nitrous oxide (N₂O) emissions, a potent greenhouse gas. However, in this study, these bacteria were found to use sulfide as an alternative electron source, enhancing denitrification and reducing nitrous oxide (N₂O) emissions. Through microcosm incubations, DNA stable-isotope probing, and metagenomics, the researchers demonstrated that these versatile heterotrophs exhibit a flexible physiology allowing them to thrive in sulfide-rich environments and compete effectively with chemolithoautotrophic denitrifiers. This discovery shifts our understanding of sulfur cycling and suggests new implications for mitigating climate change.

Specific Comments

1. There are a few grammatical errors and some sentences are lengthy and complex, which might hinder comprehension. Simplify and break down long sentences where possible.
2. Line 62-64: suggested revision: "Incomplete denitrification is a major source of N₂O, a potent greenhouse gas that contributes to climate warming. Additionally, N₂O emissions contribute to ozone layer depletion and acid rain formation."
3. Line 65-68: Please revise to "In many estuaries, coastal regions, and marine oxygen-deficient zones, the sulfur cycle plays a crucial role in regulating biogeochemical processes. This is due to the dominance of sulfate as a terminal electron acceptor and the subsequent release of hydrogen sulfide (H₂S) from benthic sediments."
4. Line 68-75: This part should include a more detailed introduction to sulfide-driven dissimilatory nitrate reduction to ammonium (DNRA) (Canfield et. al, 2010 (Science); Wang et. al, 2021 (Water Research), etc.).
5. Line 71-73: Please revise to "Sulfide is highly toxic to heterotrophic denitrifiers, adversely affecting microbial activity and potentially increasing N₂O emissions."
6. Line 76-80: "In this study, we investigate the response of heterotrophic denitrifiers to sulfide in the highly productive Songhua River Estuary, a major tributary within the largest water system in East Asia. We employed a combination of microcosm and lab culture experiments, metagenomics, and a novel community-isotope-corrected DNA stable-isotope probing (DNA-SIP) approach."
7. In sediment microcosms, how do community compositions shift over time under different treatment conditions? How is the stability and resilience of these microbial communities in response to fluctuating environmental factors?
8. How do the lab-based findings translate to natural estuarine environments? Are the conditions in the microcosm experiments reflective of those in natural settings, and how might environmental variability influence the activity of F-SOHDs? Can modelling approaches be used to predict their contributions to nutrient cycling and greenhouse gas emissions

under various environmental scenarios?

9. Given the ability of F-SOHDs to reduce N₂O emissions, how can this be applied in wastewater treatment? What are the practical steps to harness these microbes in engineered systems to mitigate greenhouse gas? Can the activity of F-SOHDs be enhanced in situ through biostimulation strategies? What types of amendments (e.g., organic carbon, sulfide) would be most effective, and what are the potential ecological side effects of such interventions?

10. While metagenomics provides a snapshot of potential metabolic capabilities, it does not confirm actual activity. How can the findings from metagenomics be integrated with other omics approaches to provide a more comprehensive understanding of microbial functions in these environments?

Version 1:

Reviewer comments:

Reviewer #1

(Remarks to the Author)

No more questions.

Reviewer #2

(Remarks to the Author)

I thank the authors, Shao et al., for making the effort to address my comments in full. The authors made several important clarifications across the manuscript regarding the discussion of chemolithoheterotrophic bacteria more generally, added a more nuanced discussion of method caveats, and a more refined assessment of results from the DNA-SIP experiments. Moreover, the authors corrected mistakes regarding the Gibbs-free energy calculations. After reviewing the revised manuscript, I did not find any other issues. Again, I commend the authors on their excellent work and look forward to seeing the manuscript published.

Reviewer #3

(Remarks to the Author)

No further comments

Response to Reviewers' Comments

We sincerely appreciate the insightful comments and suggestions from the reviewers and editor, which have greatly helped improve the quality of the manuscript (NCOMMS-24-34488A). We have now carefully revised the manuscript and addressed each comment raised by the reviewers with a point-by-point response. The reviewer comments are provided in italic black font, and our responses and the changes made within the revised manuscript are indicated in blue font color.

Responses to Reviewer #1

Reviewer #1 (Remarks to the Author):

This manuscript found the important contribution of heterotrophic denitrifiers on N₂O reduction in natural ecosystem, especially in estuaries environments. Based on authors data, heterotrophic denitrifiers seemed to exert a more significant role compared to chemolithoautotrophic denitrifiers, which was completely beyond folks's traditional understanding. These findings help folks further comprehend the interaction between C, N and S cycles. Before its publication, the followings should be addressed:

Response: We sincerely thank the reviewer for the positive assessment and insightful comments on our work! We have carefully revised the manuscript, fully addressing all the comments/suggestions provided. Please find our detailed responses below.

1. Line 88-89, why authors sampled the Estuarine sediments at the junction of the Songhua River and effluent from municipal treatment? And in Fig.S1 the detailed sample site was not provided.

Response: We conducted sampling at the estuarine junction site, a critical hotspot for element exchange, due to its key role as a regulator of nutrient flow between human communities and natural ecosystems. Annually, over 300 billion cubic meters of anthropogenic wastewater, about one-seventh of the global river volume, are generated, with around 60% being discharged into natural waters after treatment (Mateo et al. 2015; Wu et al. 2019). This contributes large amounts of nitrogen, phosphorus, sulfur, and organic matter to rivers, influencing global biogeochemical cycles (Shao et al. 2022). In this study, we focused on the junction between the Taiping Wastewater Treatment Plant, the largest wastewater treatment system in Heilongjiang Province, and the Songhua River, the largest river in East Asia. This site was chosen to examine the impact of anthropogenic activities on biogeochemical cycling. Per the reviewer's suggestion, we added detailed geographic coordinates and sampling procedures in Revised Supplementary Information Lines 42–45 and Fig. S1a.

2. line 91, what was the quality of 20 mL of surface sediments?

Response: We are unsure what the reviewer means by “quality” in this comment. If it refers to weight, the 20 mL of surface sediments weighs 24.2 grams. If the intent was to inquire about the physicochemical parameters, they were measured as follows: the collected 20 mL surface sediment contained 48.35% sand, 22.57% total organic matter, 1.34% total nitrogen, 0.07% total phosphorus, and had a pH of 7.06. Both weight and physiochemical parameters have been integrated in the main

text (see Revised Manuscript Lines 122 and 544–545).

3. In Fig. 1d, the N amounts were balanced or not?

Response: The total nitrogen remained stable at around 300 μM . In Fig. 1d, we displayed only nitrate, nitrogen gas and ammonium concentrations to focus on how much nitrate has been reduced via denitrification and DNRA, producing nitrogen gas and ammonium respectively. Intermediate products, including nitrite and nitrous oxide, were also measured (see Figure R1 below). Total nitrogen, calculated by summing different nitrogen states, ranged from 293.6 to 304.4 μM , indicating a balanced nitrogen amounts.

Figure R1. Concentration of nitrite, and nitrous oxide and calculated total nitrogen.

4. line 171-177, please provide more details on this deduction of heterotrophs may dominate sulfur oxidation and denitrification in this ecosystem.

Response: As suggested, we have added further clarification to strengthen our inference. Combined with the feedback from Reviewer #2, we have made substantial revisions to this section. We now believe there is strong evidence supporting the dominance of F-SOHDs over chemolithoautotrophs in sulfur oxidation and denitrification. The details are as follows:

1) The F-SOHDs constitute 60.8% of the denitrifying sulfur-oxidizing community, whereas chemolithoautotrophs account for only 12.1% (see Revised Manuscript Lines 219–221).

2) In the microbial community of the +C+N+S microcosm, *Azoarcus* is the most dominant genus, accounting for up to 61.5% of the relative abundance (Fig. 2c). Its relative abundance increased by 21.6% with the addition of $^{13}\text{C}_0$, but decreased by 10.1% with the addition of $^{13}\text{C}_i$. Additionally, this genus exhibited enhanced $^{13}\text{C}_0$ assimilation in response to sulfide addition, as its relative abundance increased by 64.1% in the $(^{12}\text{C}_i+^{13}\text{C}_0)+\text{N}+\text{S}$ treatment compared to $(^{12}\text{C}_i+^{13}\text{C}_0)+\text{N}$ treatment. These findings support its functional identification as an F-SOHD (see Revised Manuscript Lines 149–153 and 158–164).

3) Compared to the +C+N microcosm, the relative abundance of *Thiobacillus*, the most abundant chemolithoautotroph in the community, decreased by an average of 3.8% in the +C+N+S microcosm (Fig. 2c,d). This suggests that F-SOHDs outcompete chemolithoautotrophs when sulfide is present (see Revised Manuscript Lines 221–224).

4) Furthermore, the subsequent metagenomic analyses further confirm the functional inferences for the microbes mentioned above, specifically identifying *Azoarcus* and *Pseudomonas* as F-SOHDs and *Thiobacillus* as a chemolithoautotroph.

5. line 222-229, author mainly focused on the contribution of denitrification processes, and they also found DNRA process in the sediments. Thus, how about the DNRA process contribution on both NO₃⁻ reduction and S₂- oxidation?

Response: The DNRA process plays a relatively minor role in nitrate reduction, as evidenced by three factors: 1) qPCR experiments showed that *nrfA* gene copies accounted for only 3% of *nirS* and *nirK* gene copies (Fig. 1b); 2) ammonium release was just 16.63% of dinitrogen gas release under the same conditions (Fig. 1d); 3) metagenomic data revealed that microbes capable of DNRA have a much lower relative abundance (3.76%) compared to F-SOHDs (86.55%) (Fig. 3). Thus, F-SOHDs rather than DNRA microbes, are the primary contributors to nitrate reduction. Given the importance of F-SOHDs and to avoid distraction, we focused directly on their role in detoxification in the presence of sulfide, without assessing the contribution of DNRA microbes to sulfide oxidation.

6. line 299-303, more details information are required to support authors's deduction.

Response: Following the reviewer's suggestion, we have provided additional details to strengthen the deduction (see Revised Manuscript Lines 370–383).

7. line 351-353, please provide more similar sites, I do not believe that only 3 similar sample were enough for supporting authors's opinion.

Response: Thank you for your insightful comment. We have provided additional sample sites, including bay and freshwater sediments (Aelion et al. 2009; Miller et al. 1986), agricultural soils (Tam et al. 1979), marine and soil pure cultures (Sørensen et al. 1987; Sørensen et al., 1980), and wastewater treatment-related activated sludges (Pan et al. 2013; Schönharting et al. 1998), to further support the discussion. Relevant references have also been included (see Revised Manuscript Lines 478–482). This sentence has been rephrased in response to the comments provided by Reviewers #1 and #2.

Responses to Reviewer #2

Reviewer #2 (Remarks to the Author):

The study by Shao et al., explores the presence of facultative sulfur-oxidizing heterotrophic denitrifiers, abbreviated by the authors as F-SOHDs, in estuary sediments of the Songhua River System. Specifically, their study employed DNA-SIP experiments, in combination with metagenomic sequencing and enrichment cultures to identify F-SOHD members in a mixed microbial community comprising chemoorganoheterotrophs and chemolithoautotrophs. Using this range of techniques, they identified environmental genera affiliated with Thauera, Pseudomonas, and Azoacoccus as likely F-SOHDs in their samples. Lastly, they demonstrate using F-SOHD enriched bioreactors, that such

members are capable at better suppressing N₂O emissions than their heterotrophic denitrifying counterparts. In turn, this led to the suggestion that F-SOHDs could be useful in industrial settings to mitigate N₂O emissions.

As someone who has previously studied F-SOHDs, which I refer to as chemolithoheterotrophs, I find the work to be of high value as it expands on the potential for chemolithoheterotrophy in the environment. Moreover, the authors employed an impressive range of tools to identify these microbes *in situ*. Overall, I find the results fairly compelling, albeit the manuscript is not well-rounded at times, and does tend to overstate claims. Namely, the introduction did not provide the reader with state-of-the-art background information concerning chemolithoheterotrophs. The results I generally find compelling, although I have some points that need further clarification, and the authors should discuss some of the discrepancies in their data. While I don't disagree with all the points raised in the discussion, it does need some more work/reframing, to present a more well-rounded case for chemolithoheterotrophy in the environment. In addition, some errors in their estimates of Gibbs free energy need to be addressed in the discussion. The method section was at times thorough, but other parts were fairly vague, to the point that the reader would not be able to reproduce the experiment without additional clarification. Ending on an optimistic note, the figures are nicely polished, and the data/conclusions (while overstated at times) appear generally sound.

Response: We sincerely appreciate the reviewer's positive and encouraging feedback on our work. All the comments are highly constructive and thorough, which helped us substantially improve the quality of the manuscript. Following suggestions, we have revised the sentence description throughout the manuscript to avoid overstatements and have rewritten the Introduction to include more research background on chemolithoheterotrophs. Regarding the Gibbs free energy estimates, we have recalculated these values and corrected the relevant sentences/figures in the Revised Manuscript. To clarify, we have added more details to the Methods section as recommended, to provide a clearer explanation for readers. We have carefully revised the manuscript, fully addressing all the comments/suggestions provided. Please find our detailed responses below.

Comments

1. Lines 40-41: I find this statement to be a little vague, namely because heterotrophic denitrifiers operate in environments where sulfide is present. But what the authors could specify is that sulfide inhibits a key enzymatic reaction that catalyzes the reduction of N₂O to N₂. Hence, heterotrophic denitrifiers are sensitive unless sulfide is removed, either by cooperation with chemolithoautotrophs or with chemolithoheterotrophs. This type of denitrifying community structure is indeed what we find in organic and sulfide-rich ecosystems.

Response: We fully agree with the reviewer's comment that heterotrophic denitrifiers are active but carry out incomplete denitrification in the presence of sulfide, owing to the inhibition of NosZ. To clarify this point, we have revised the sentence as follows: "However, their inability to oxidize sulfide renders them vulnerable to this toxic molecule, which inhibits the key enzymatic reaction responsible for reducing nitrous oxide (N₂O) to dinitrogen gas, thereby raising greenhouse gas N₂O emissions." (also see Revised Manuscript Lines 40–43).

2. Line 43: *This statement comes across as a little too strong as their role in the environment, including aquatic ecosystems, has been noticed by some and even empirically demonstrated (Sorokin 2003; Boltyanskaya et al. 2007; Callbeck et al. 2019; Wang and Shao 2021). As I mentioned above, the value of the study is not in that it discovered sulfide oxidation by heterotrophs in the environment, it is that it demonstrates that numerous microbes in estuary sediments could thrive via a chemolithoheterotrophic metabolism greatly expanding its potential. I would therefore consider re-framing this part.*

Response: As suggested, we have revised the sentence to emphasize the detection of chemolithoheterotrophs and their ability to thrive, without referring to them as “previously unnoticed” (see Revised Manuscript Lines 43–48).

3. Lines 48-51: *I agree with this statement, but this largely repeats the findings of Callbeck et al., 2019, which I detail a little later in this review. Moreover, their justification of this statement based on their Gibbs free energy calculations appears to be flawed (see later discussion).*

Response: We agree that a similar phenomenon has been observed in other studies. As the reviewer pointed out, while this statement aligns with previous findings and may appear repetitive, we have kept it in the abstract and revised it to accurately represent the results of our study (see Revised Manuscript Lines 52–55). Additionally, we cited the paper the reviewer mentioned in the discussion to support our findings. Regarding the concerns about Gibbs free energy calculations, we have recalculated these values and revised the statement accordingly in Revised Manuscript Lines 514–519 and Revised Figure 6 (please see Comments 42 and 69).

4. Lines 51-53: *I'm in favor of this work generally, but this is an overstatement. Firstly, environmental examples of sulfide oxidation by heterotrophs exist, thus the use of “paradigm shift” is over the top. Secondly, I understand that if you work on N₂O you can make a link to climate change, however, to say that his work has “profound implications” for climate change is a stretch too far. For example, it is not the case that the authors have tested/modeled their reactors under various climate change scenarios. In my view, the work is a good piece because it advances the idea that chemolithoheterotrophs form a persistent part of the microbial community in an organic matter-rich estuary.*

Response: We appreciate the reviewer’s thoughtful comments and are pleased to hear that the reviewer considers our work as advancing the current understanding of chemolithoheterotrophs. As suggested, we have rephrased the sentence to avoid overstatement (see Revised Manuscript Lines 57–60).

5. Lines 62-64: *For clarity, denitrification is one of two major processes, nitrification being the other. The statement makes it seem as though denitrification is the only one.*

Response: Revised as suggested (see Revised Manuscript Lines 69–71). The sentence has been rephrased in response to the comments provided by Reviewers #2 and #3.

6. Lines 65-68: Fully agree, for a comprehensive review of sulfur cycling and its influence over the nitrogen cycle in oxygen minimum zones please see (Callbeck et al. 2021).

Response: This is a great citation. We have carefully studied this paper and have incorporated relevant key insights into the *Introduction* and *Discussion* sections of the Revised Manuscript. Please see the detailed changes in Lines 76–79, 86–91, 447–451, and 466–473.

7. Lines 71-73: The authors only superficially mention that it impacts N₂O but how is this the case? Some have speculated that sulfide also inhibits the last step of denitrification, as sulfide acts as a Cu-chelator.

Response: Per the reviewer's suggestion, we have added details regarding the specific inhibitory mechanism of sulfide in the Revised Manuscript Lines 76–79 and 466–471.

8. Lines 68-70: Certainly, chemolithoautotrophs are well-recognized, but work by Callbeck et al., 2019 has shown that both chemolithoautotrophs, and chemolithoheterotroph (aka. F-SOHD) can co-exist to oxidize sulfide during a sulfidic event off the coast of Peru.

Response: We have provided additional sentences to review previous findings regarding chemolithoheterotrophic denitrifier *Arcobacter peruensis* in coastal waters off Peru (see Revised Manuscript Lines 86–91).

9. Line 72: I understand the point the authors are making, however, for clarification sulfide is toxic to all denitrifiers, including nitrate-reducing chemolithoautotrophs, chemolithoheterotrophs and organoheterotrophs. The reason for this I explain a little later in the discussion section below.

Response: Similar to comment 6, we have revised the relevant sections to specify sulfide toxicity toward Cu-dependent enzymes, rather than limiting the description to conventional heterotrophic denitrifiers (see Revised Manuscript Lines 76–79 and 466–469).

10. Lines 73-75: This statement, which is effectively the premise of the manuscript, is simply not up to date with the state-of-the-art. As I mentioned above, it's not as though we know nothing about this topic of F-SOHD/chemolithoheterotrophs. The authors have partly relayed this information in the discussion (on lines 368-370), but choose not to interject this information here or above when background information is needed on this specialized metabolism. Again, the work is valuable not because it is the first to discover F-SOHD, but because it expands the potential for this metabolism to operate in the aquatic realm. The authors should consider reworking this paragraph to include the state-of-the-art, and perhaps frame the premise in a slightly different way.

Response: We greatly appreciate the reviewer's constructive comment. Following your suggestions, we have thoroughly rewritten this paragraph to incorporate more relevant research background on chemolithoheterotrophs and sulfide toxicity toward Cu-dependent enzymes. Additionally, we have clarified the knowledge gap and refined the research objective (see the second paragraph of Introduction section in the Revised Manuscript).

11. Lines 80-83: To be specific, the study identifies new F-SOHDs at a sediment site. Somewhat misleading is the use of “ubiquitous” as this implies that the authors provided evidence of F-SOHDs widespread distribution across multiple sampling sites, although their study site is positioned at one location (45°49'12.81"N, 126°43'22.51"E). Again, the authors have a habit of overstating claims, when in fact, there is no need.

Response: As suggested, we have deleted the word “ubiquitous” from the manuscript to avoid overstating (see Revised Manuscript Line 107).

12. Lines 82-85: I find this overly broad and introduces a bit of speculation, especially on the climate change front; can the authors be a bit more specific?

Response: We have removed the speculated description of climate change and rewritten these two sentences to make it more specific. The revised sentences are pasted below for convenience (also see Revised Manuscript Lines 108–114).

“This capability allows them to detoxify sulfide and achieve complete denitrification, thereby mitigating the N₂O emission. This work enhances our understanding of heterotrophic denitrifiers capable of sulfide detoxification in aquatic environments, with broader implications for regulating carbon, nitrogen, and sulfur cycles, including the mitigation of N₂O emissions in both natural and engineered system.” (Lines 108–114)

13. Lines 98-101: If I have this right, this would amount to roughly two-thirds of the original 300 μM of nitrate being recovered as N₂ and NH₄⁺. But roughly one-third is not recovered, hence, both complete and incomplete denitrification are evident in this experiment.

Response: We agree with the reviewer’s observation that both complete and incomplete denitrification are occurring simultaneously. As shown in Figure R1 below, 56.1% of the consumed nitrate was converted to N₂O, rather than being fully reduced to N₂. We have revised this sentence to reflect this clarification (see Revised Manuscript Lines 129–132).

Figure R1. Supplementary data of total nitrogen amounts, nitrite, and nitrous oxide for Fig.1d.

14. Line 108: The with and without sulfide addition experiments are somewhat confusing. I understand that the authors added additional sulfide, but it should be perhaps reiterated that sulfide is going to be produced in the background, unless the authors removed the sulfate for sulfate reducers, in incubations like C+N, or Ci, or Co etc...

Response: We appreciate the reviewer's concerns. In these experiments, we did not specifically exclude sulfate or sulfate reducers. Our experiment was designed to study the effect of excess sulfide on denitrification without excess manipulation of the community. While we acknowledge that sulfide produced by background sulfate reduction could potentially contribute to nitrate reduction through the re-oxidation of sulfide, the addition of excess sulfide led to a significant increase in the rate of nitrate reduction compared to the group without added sulfide (Figure 1c), suggesting that this supplemental sulfide may serve as an electron donor to enhance nitrate reduction. Thus, our results provide evidence of a sulfide-supported nitrate reduction process within the sediment community. To clarify the experimental design, we have revised the label in Revised Figure 1c from "sulfide free" to "no sulfide added".

15. Lines 117-121: The increases of 1.3% and 3.8% don't seem to be substantial for *Thauera* and *Pseudomonas*. Thus, can the authors assume that their heterotrophs according to these values? And in the second part, how much did these members decrease when amended with $^{13}\text{C}_i$? In light of chemolithoautotrophic abundances increasing with the addition of ^{13}C , could this not artificially dilute such heterotrophs like *Thauera* and *Pseudomonas*, if it's based on relative abundances?

Response: We understand that the reviewer has three concerns: 1) Whether a heterotrophic lifestyle can be inferred from relatively small changes in relative abundance in the ($^{13}\text{C}_o+^{12}\text{C}_i$)+N+S and ($^{12}\text{C}_o+^{12}\text{C}_i$)+N+S treatments; 2) The detailed changes in the relative abundances of these three genera when comparing the ($^{12}\text{C}_o+^{13}\text{C}_i$)+N+S and ($^{12}\text{C}_o+^{12}\text{C}_i$)+N+S treatments; 3) The reliability of inferring lifestyle solely from relative abundance data.

Response to point 1: The modest increases observed in fast-growing heterotrophs like *Thauera* (1.3%) and *Pseudomonas* (3.8%) are indeed a common outcome in DNA-SIP analysis, particularly within complex microbial communities, especially given relatively slow growth under anaerobic conditions and short-term isotope labeling. Similar small increases in abundance have been widely reported in other studies (e.g., Sathyamoorthy et al. 2018; Sun et al. 2021). In our study, the relatively small increases in *Thauera* and *Pseudomonas* can partially be attributed to their low background abundances (2.0% and 5.1%). These two genera exhibited 63% and 74% additional increases in relative abundance in the heavy DNA fraction, compared to the 35% increase observed in *Azoarcus*. This suggests that, despite their low initial abundance, their growth may be higher than *Azoarcus*. The dominance of *Thauera* and *Pseudomonas* in the 612-day long-term enrichment experiment further supports their active growth.

Response to point 2: Compared to the heavy fraction without ^{13}C labelling in the +C+N+S microcosm, the relative abundances of *Thauera*, *Azoarcus*, and *Pseudomonas* decreased by 0.1%, 10.1%, and 1.5% when amended with $^{13}\text{C}_i$. These values have been added for clarity (see Revised

Manuscript Lines 151–152).

Response to point 3: We acknowledge the limitations of inferring microbial lifestyles solely from relative abundance data, as changes in chemolithoautotrophic populations could dilute the apparent relative abundance of heterotrophs like *Thauera* and *Pseudomonas*. This limitation is inherent to DNA-SIP when paired with 16S rRNA amplicon sequencing. DNA-SIP alone provides preliminary insights into active microorganisms but lacks the resolution for definitive lifestyle characterization. To enhance accuracy, integrating DNA-SIP with qPCR, metagenomics, or metatranscriptomics could provide advantages. We have moderated the description regarding lifestyle inference and emphasized the methodological limitations in the Revised Manuscript Lines 153 and 404–407.

16. Lines 121-122: Can the authors comment on why *Thauera* comprised a large fraction of the microbial community in the experiment void of sulfide (C+N) when compared to the C+N+S experiments? It would appear that this member is sensitive to sulfide inhibition. In addition, an increase of 0.7% is practically negligible. Moreover, it appears that *Pseudomonas* shows a different pattern that is opposite to your statement. Also in Fig. 2d, *Pseudomonas* and *Azoarcus* both have $^{13}\text{C}_i$ which is higher than the control in grey. Hence, this goes against the idea that no ^{13}C -DIC is incorporated. Part of the problem the authors might encounter in their analysis is that heterotrophs can also assimilate ^{13}C -DIC (e.g. Erb 2011), which might complicate their analysis, or at least require a note.

Response: Thank you for your suggestion and relevant literature. The concern regarding the increased abundance of *Thauera* in C+N compared to C+N+S has been fully addressed in Comment 25 (please refer to that section for details). Regarding the concern of $^{13}\text{C}_i$ -DIC assimilation, we acknowledge that some heterotrophic microorganisms might incorporate some DIC, as noted for *Azoarcus* and *Pseudomonas*, which partially incorporated $^{13}\text{C}_i$ isotope in the $(^{13}\text{C}_i+^{12}\text{C}_o)$ +N treatment compared to $(^{12}\text{C}_i+^{12}\text{C}_o)$ +N treatment (Fig. 2d). This observation aligns with the reviewer's comment that some heterotrophs are capable of assimilating ^{13}C -DIC, as described in the reference provided (Erb, 2011). To address this potential complication, we have noted this issue explicitly in the main text and cited the relevant reference to provide readers with a balanced and comprehensive understanding of the data interpretation (see Revised Manuscript Lines 407–409).

17. Lines 125-129: *Thiobacillus* increased in Fig. 2d (C+N) when sulfide was not present, which runs counter to their chemolithoautotrophic lifestyle. And also what about other chemolithoautotrophic members such as *Chromatiales* (M5.bin31) and *Sulfurivermis* (M5.bin64) shown in Fig. 3?

Response: It is true that *Thiobacillus* was slightly less abundant and apparently slightly less active when sulfide was added in the presence of organic matter (comparing Fig. 2c and Fig. 2d). We reason that in the presence of OM, *Thiobacillus* may be outcompeted by chemolithoheterotrophs, because heterotrophy is energetically more favorable than autotrophy. But our results in Fig. 2f and Fig. 2g in comparison with each other and with Fig. 2c and Fig. 2d show a strong stimulation of

Thiobacillus when sulfide is added in the absence of added OM. This supports our inference that *Thiobacillus* is primarily chemolithoautotrophic in our experiments, a conclusion further corroborated by metagenomic data.

Due to limited space, we decided to not show low-abundance taxa in Figure 2, such as *Chromatiales* (M5.bin31) and *Sulfurivermis* (M5.bin64), which are both below 0.08%. This decision also allowed us to focus the related text on the most abundant microbes. We included them in Figure 3 because that's a more comprehensive figure. We think this decision strikes the right balance of keeping the focus on the most abundant taxa while still showing data for these other chemolithotrophs.

18. Lines 130-154: It seems that this method distinguishes *Thiobacillus*, but what about other chemolithoautotrophic members such as *Chromatiales* (M5.bin31) and *Sulfurivermis* (M5.bin64) shown in Fig. 3, where do they position in Fig. 2i? In addition, organisms like *Dechloromonas*, which according to Fig. 3 is an F-SOHD, are positioned in the bottom left of Fig. 2i alongside *Thiobacillus* the chemolithoautotroph. I also have difficulties in reconciling the very negligible increases the authors reported on lines 118 and 121 and how this results in very clear patterns for *Thauera* and *Pseudomonas* in Fig. 2i. I think the authors should add that this method might not be perfect at disentangling all F-SOHDs from other community members, but it may work for more active microbes.

Response: We agree with the reviewer's comment and acknowledge the limitations of this approach. While it effectively highlights the more abundant and active microbes within the community, it may not fully capture the functional roles of all microbial members, particularly those of low abundance. In response to these considerations, we have removed the Fig. 2h and 2i, along with the associated descriptions, and opted for a more straightforward analysis of abundance changes for functional identification in the main text (see Revised Manuscript Lines 158–200). Moreover, we have significantly shortened the overstated description of DNA-SIP (see Revised Manuscript Lines 388–404) and explicitly addressed its limitations to provide readers with a clearer understanding (see Revised Manuscript Lines 404–407). For the concerns raised in Lines 150 and 154, these have been addressed in our response to Comment 15 above.

19. Lines 138-142: The exception would be *Arcobacter peruensis*, which reduces nitrate, oxidizes sulfide, and simultaneously assimilates/oxidizes organic matter into biomass for energy/growth (Callbeck et al. 2019). *Arcobacter peruensis* was also abundant in sulfidic and organic matter-rich shelf waters alongside chemolithoautotrophs like SUP05.

Response: As stated, *Arcobacter peruensis* is indeed a very important bacterium with metabolic potential similar to F-SOHDs identified in this study. We acknowledge that in certain environments, such as organic carbon-rich waters, *Arcobacter peruensis* can play a significant role alongside chemolithoautotrophs. However, in our study, we aimed to emphasize the most general case rather than focus on specific exceptions. As the reviewer noted, we have added more detailed description of the role of *Arcobacter peruensis* in both the Introduction and Discussion sections (see Revised

Manuscript Lines 89–91 and 423–426).

20. Line 142-144: In the methods section, on lines 488-490, the authors state that “microbes have the capacity to oxidize sulfide if they increase in relative abundances in +C+N+S compared to +C+N microcosms”, but the issue with *Thauera* is that its abundance decreases, and hence, by their logic above is not a sulfide oxidizer. The authors could further clarify this.

Response: We apologize for the confusion caused by the improper sentence structure and the inaccurate inference regarding the sulfide sensitivity of *Thauera* in the manuscript. The original sentence “This appears to show that heterotrophs might also contribute to sulfur oxidation and denitrification in our system” was intended to emphasize the genera *Azoarcus* and *Pseudomonas*, rather than *Thauera*. Based on the abundance changes observed, here we could not infer that *Thauera* act as a sulfide oxidizer in this context; this will be discussed in detail in Comment 25 below. Following the reviewer’s comment, we have rephrased the relevant sentences to improve clarity (see Revised manuscript Lines 158–165).

21. Line 171: The authors coin the new acronym facultative sulfur-oxidizing heterotrophic denitrifiers (F-SOHDs) for a metabolic process that has been described previously as chemolithoheterotrophy (Callbeck et al. 2019). For consistency’s sake with previous work, why not term F-SOHDs as chemolithoheterotrophs which include microbes that oxidize sulfide, and reduce nitrate while simultaneously utilizing organic carbon as an energy/carbon source for growth? This would reduce the number of acronyms in the literature, and make for a more comparable term to chemolithoautotrophy, which the authors readily apply in their manuscript.

Response: We appreciate the reviewer’s suggestion. However, we believe that the term “chemolithoheterotrophy” is somewhat broad and may not accurately convey the specific characteristics of the microorganisms we are studying. While chemolithoheterotrophs are defined as microbes that derive energy from oxidizing inorganic compounds while using organic matter as a carbon source, this category includes a wide variety of microbes that utilize diverse inorganic electron donors, such as hydrogen, manganese, or iron (Miroshnichenko et al. 2003; Kompantseva et al. 2017; Price et al. 2018), and is not limited to sulfur. Moreover, the term chemolithoheterotrophy does not necessarily imply a connection to denitrification or the fact that these microorganisms can also grow without added reduced sulfur (this “facultative”). Our findings indicate that microbes detected in our study represent a subset of heterotrophic denitrifiers with a unique ability to oxidize sulfur, distinct from microbes that utilize other organic matter. More importantly, they catalyze complete denitrification and mitigate N₂O emissions in the presence of sulfide. To capture these specific features, we propose the term “facultative sulfur-oxidizing heterotrophic denitrifiers” (F-SOHDs), providing a clearer and more precise description for readers.

22. Lines 204-207: The authors mention *Arcobacter*, but it was later been shown to grow as a chemolithoheterotroph (Callbeck et al., 2019). Hence, *Arcobacter peruensis* had the capacity to assimilate organic carbon, reduce nitrate, and oxidize sulfide, much like what the authors describe

in their paper for *Thauera* and other members.

Response: Following the suggestion, we have removed “*Arcobacter*” here to ensure the accuracy of the results (see Revised manuscript Lines 256).

23. Lines 208-213: I checked the literature on some of the closest affiliated microbes in Figure 3 (see below), and some of them have capacities for CO₂ fixation, despite being labeled as having no capacity in Fig. 3c.

- *Azoarcus* (see https://www.kegg.jp/pathway/aza00710+AZKH_p0225)

- *Ralstonia pickettii* DTP0602 (<https://www.kegg.jp/pathway/rpj00710+M00166>)

- *Pseudomonas yangmingensis* (Wong and Lee 2014)

Thus, while I believe that some are probably chemolithoheterotrophs, the authors should stipulate some caveats and unknowns here.

Response: We acknowledge that some of our inferences might be biased due to the incomplete nature of the reconstructed genomes, although the majority are of high quality. Regarding the possibility that some of these organisms may be chemolithoautotrophs, it is important to note that significant metabolic differences can exist even among closely related microbes. For instance, *Azoarcus* sp. CIB and *Azoarcus* sp. BH72 have been reported as a heterotroph (Durante et al. 2019; Hurek et al. 1996). To address this uncertainty and potential biases, we have added a sentence clarifying these limitations in our findings (see Revised manuscript Lines 266–269).

24. Lines 222-225: As mentioned above the authors should apply some caution here. Moreover, the authors have not measured the nitrate turnover rates of the specific members, thus to say that this cluster is more biogeochemical relevant is rather speculative. It's fair to say that strong potential for chemolithoheterotrophy exists at their study site.

Response: As suggested, we revised the sentence to reflect the presence of F-SOHDs in present study (see Revised Manuscript Lines 279–282).

25. Lines 244-249: I find it difficult to take the position that *Thauera* is sensitive to sulfide. If the authors take this position then they have to reconcile in their discussion that both nitrate-reducing chemoorganoheterotrophs and chemolithoheterotrophs are sensitive to sulfide. Presently, they hold only the former to be true. However, I would suggest that the authors explain this differently. For example, sulfide-oxidizers like *Arcobacter*, SUP05 and *Sulfurovum* can co-exist in the presence of sulfide however *Arcobacter* and *Sulfurovum* tend to dominate over SUP05 when sulfide concentrations exceed 10 μ M (Callbeck et al., 2021). So could it not be the case that *Pseudomonas* outcompetes *Thauera* at a certain sulfide threshold, thereby lowering its contribution to the overall community (based on relative abundances). Taking this position means that *Thauera* is not necessarily sensitive but rather outcompeted under certain environmental conditions by others.

Response: We sincerely appreciate the reviewer's insightful comments. From a microbial ecophysiology perspective, the reviewer's theory on competition among different microorganisms under high-sulfide conditions provides a more reasonable explanation for reconciling the observed

decrease in abundance with the detected sulfur oxidation capacity of *Thauera*. Given the insufficient evidence, we cannot definitively conclude that *Thauera* is a sulfide-sensitive microorganism. After thorough discussion, we have fully incorporated the reviewer's suggestion and revised the sentences to present the results in a more reasonable manner (see Revised Manuscript Lines 302–313).

26. Lines 266-267: *This is a bit confusing here as it is not clear why being a heterotrophic denitrifier leads to N₂O production, and also why chemolithoautotrophs do not. In my mind it's not so clear cut, for example, SUP05 identified in Peru can perform complete denitrification to N₂ (Callbeck et al. 2018), while other SUP05 species in the Saanich Inlet can only do nitrate reduction to N₂O (Walsh et al. 2009). Might also be worth mentioning here that Arcobacter peruensis is an exception, plus others exist as mentioned earlier.*

Response: According to the Comments 33 and 34 raised by the reviewer, we agree that sulfide not only poisons heterotrophs but also impacts chemolithoautotrophs by interacting with Cu-dependent enzymes, such as NosZ. However, we interpret that sulfide-oxidizing chemolithotrophs may be less sensitive to sulfide toxicity due to their ability to oxidize sulfide into less toxic compounds. To avoid confusion, we have deleted this sentence (see Revised Manuscript Lines 333–334) and added more details (see Comments 33 and 34 below) regarding the sulfide sensitivity of complete denitrification in conventional heterotrophic denitrifiers (see Revised Manuscript Lines 466–476). Also, we have expanded the revised Discussion section to elaborate on the relationships between F-SOHDs, heterotrophic denitrifiers and broader chemolithoautotrophic denitrifiers, including *Arcobacter peruensis*.

27. Lines 285-287: *This line contradicts the author's statement above that heterotrophic members like Thauera could be sensitive to sulfide.*

Response: As mentioned above, several Cu-dependent enzymes such as NosZ, encoded by heterotrophic denitrifiers, are sensitive to sulfide. Therefore, the previous description was inaccurate, and we have revised it accordingly. The revised sentence reflects that the sulfide oxidation capacity of F-SOHDs leads to the consumption of sulfide, thereby mitigating or preventing its toxic effects (see Revised Manuscript Lines 352–356).

28. Lines 305-321: *I agree with the authors that DNA-SIP technology is important to unraveling microbial ecology-related questions. But I find a dedicated paragraph to this in the opening discussion is a little excessive, as the method is in principle the same with the difference being that the authors did an additional correlative analysis (Fig. 2i, g). In addition, organisms like Dechloromonas, which according to Fig. 3 is an F-SOHD, are positioned in the bottom left of Fig. 2i alongside Thiobacillus the chemolithoautotroph. Hence, this method is not perfect, and the authors (if they decide to keep this piece) will need to address the caveats. As a recommendation, I would instead suggest the authors shorten or drop to make way for a more detailed discussion of the specific chemolithoheterotroph members they identified in their study, which was lacking in my opinion (see below).*

Response: Following the reviewer's suggestion, we have substantially shortened this paragraph and highlighted the method's limitations (see Revised Manuscript Lines 387–404). Additionally, to sharpen the study's focus, we have concentrated the Discussion section on F-SOHDs identified in our study rather than on the DNA-SIP methodology. We have also expanded on the ecophysiological significance and distribution of F-SOHDs in the main text (see Revised Manuscript Lines 417–441).

29. Line 323: *I would add, "Epsilonproteobacteria and the Gammaproteobacteria SUP05 clade"*

Response: Done as suggested (see Revised Manuscript Lines 411).

30. Line 331: There are more pertinent studies to reference here that discuss sulfidic events in marine environments, which is summarized in the review by Callbeck et al., 2021.

Response: Following the reviewer's valuable suggestion and relevant reference, we have revised the citations with the work published by Callbeck et al. 2021, and Caffrey et al. 2018. Additionally, the more pertinent studies of marine sulfidic events and their impacts have also been incorporated in Revised Manuscript Lines 443–451.

31. Lines 332-333: *I believe this to be an overstatement by the authors "...uncovered here have evolved multiple sulfur oxidation pathways to cope with sulfide stress." The authors have not empirically demonstrated sulfide stress on the community and how it leads to the evolution of multiple sulfur oxidation pathways. Rather this statement could be framed as a hypothesis.*

Response: Done as suggested. The revised sentence is pasted below for convenience (also see Revised Manuscript Lines 451–453).

"Remarkably, the F-SOHDs may have developed multiple sulfur oxidation pathways to cope with sulfide stress, a hypothesis that merits further exploration." (Lines 451–453)

32. Lines 322-340: *I see the importance of a wider description of sulfidic events, but this really appears as a zoomed-out introduction piece especially since the nosZ gene discussion below brings more detail to this idea. Instead, I think the opening of the discussion would be better served reiterating what defines a F-SOHDs/chemolithoheterotroph. Namely, chemolithoheterotrophs couple sulfide oxidation to nitrate reduction while simultaneously utilizing organic carbon as an energy/carbon source (hence the assimilation of 13C into biomass in your DNA-SIP experiments). Next, I would reiterate the key putative chemolithoheterotrophs mentioned in the results section and Fig. 3. Subsequently, the authors need to spend more time describing what is already known about these species in the literature, so that the reader can assess what new information is added in this study. For example, my cursory inspection of some of the literature finds that some of these members (identified in Fig. 3) are known sulfide oxidizers like Pseudomonas stutzeri (Li et al. 2022). Thauera is well-studied and has been isolated from sulfide-rich environments and bioreactors before, implicating it already with a role as a putative sulfide oxidizer. Others are capable of fixing CO2 autotrophically such as Azoarcus (see https://www.kegg.jp/pathway/aza00710+AZKH_p0225),*

Ralstonia pickettii DTP0602 (<https://www.kegg.jp/pathway/rpj00710+M00166>) and *Pseudomonas yangmingensis* (Wong and Lee 2014). I recommend the authors incorporate a much more thorough discussion of the key members they identified.

Response: We sincerely appreciate the reviewer for the insightful comments, which are greatly important in improving the logical flow and clarity of the discussion in our manuscript. Following your suggestion, we have revised the beginning of the discussion section to better define the concept of F-SOHDs. Additionally, we have incorporated more comprehensive comparisons between the F-SOHDs identified in our study and previously characterized species reported in the literature. With these updates, we aim to emphasize the significance of our findings as suggested, particularly the discovery that F-SOHDs represent a persistent part of the microbial community in organic matter-rich estuaries, with important implications for the sulfur cycle and N₂O reduction. Please refer to the detailed revisions in the Revised Manuscript Lines 415–441.

33. Lines 345-346: I agree that sulfide has an influence over N₂O production, but the statement here that sulfide only affects heterotrophic denitrifiers and not denitrifiers more broadly – chemolithoautotrophs included – is not evidenced in the literature. Much of this discussion has taken place in the review by Callbeck et al., 2021, but to reiterate their points sulfide is a chelator of copper, and the *nosZ* gene is a copper-rich metalloenzyme. The sequestration of copper by its abiotic reaction with sulfide will induce copper limitations that result in nitrate reduction terminating at N₂O in microbial cultures (Granger and Ward 2003; Manconi et al. 2006; Moffett et al. 2012; Felgate et al. 2012). The accumulation of N₂O has therefore been linked to copper limitations induced by sulfide, which would affect nitrate-reducing bacteria (with a *nosZ*) irrespective of whether they are chemolithoautotrophs, chemolithoheterotrophs or heterotrophs.

Response: After carefully reviewing the relevant literature, we fully agree with the reviewer's suggestion. We have revised our discussion to more accurately address the broader impacts of sulfide on denitrifiers and its abiotic copper chelation mechanism (see Revised Manuscript Lines 466–473).

34. Lines 346-349: I agree with this statement, but as above this is not exclusively a heterotrophic denitrifier issue but will rather affect denitrifiers broadly.

Response: Revised as suggested. Please see the changes in Revised Manuscript Lines 466–468.

35. Lines 349-353: Indeed, sulfide-driven denitrification is likely responsible for major N₂O hotspots (Naqvi et al. 2000, 2006; Arevalo-Martinez et al. 2015; Kock et al. 2016; Bourbonnais et al. 2017; Arévalo-Martínez et al. 2019), which likely occur as a result of the mechanism outlined above. However, the difficulty I have with this line is the authors claim that these N₂O/H₂S-rich systems are dominated by heterotrophic denitrifiers, when in fact, the cited studies do not make that claim (mainly because they also pre-date large community metagenomic surveys). Secondly, in light of more recent analyses of these same environments, such as Peru, where we have major N₂O hotspots along with the presence of sulfide, the microbial community is comprised of a consortium

of nitrate-reducing chemolithoautotrophs, chemoorganoheterotrophs and chemolithoheterotrophs. Thirdly, from a logical perspective, if such systems have high organic matter, sulfide, and nitrate, wouldn't they offer the perfect ecological niche for chemolithoheterotrophs to thrive? To that point, Callbeck et al. 2019, and this study enriched for chemolithoheterotrophs from exactly these environments.

Response: As the reviewer pointed out, the cited studies do not explicitly indicate that these microbial communities are dominated by heterotrophic denitrifiers. Instead, such environments likely host a diverse consortium of microorganisms, including chemolithoautotrophs, chemoorganoheterotrophs, chemolithoheterotrophs, and others. To prevent potential misinterpretation, we have removed the inaccurate speculative statement and clarified the text accordingly (see Revised Manuscript Lines 476–482).

We fully agree with the reviewer's insight regarding ecological niche favoring F-SOHDs (or chemolithoheterotrophs) in environments rich in OM, sulfide, and nitrate. These conditions indeed create an ideal habitat for F-SOHDs to thrive, as exemplified by the *Arcobacter peruensis* identified in Peruvian coastal waters and the microorganisms observed in our estuarine sediments. This highlights an intriguing research avenue deserving of further exploration. Future studies employing large-scale metagenomic sequencing could help determine the persistence and abundance of F-SOHDs within microbial communities in these typical ecosystems.

36. Lines 353-356: I generally agree with this line, but perhaps clarify that this occurs because chemolithoheterotrophs can remove sulfide and therefore remove its ability to chelate copper, enabling N₂O reduction to N₂ to continue (via the Cu-rich nosZ).

Response: Done as suggested. The revised sentence is pasted below for convenience (also see Revised Manuscript Lines 485–486).

“By oxidizing sulfide, F-SOHDs prevent it from chelating the copper within Cu-rich NosZ, thereby enabling N₂O reduction to continue.” (Lines 485–486)

37. Lines 357-359: I agree, but it would make sense to mention that sulfide provides an additional electron donor (on top of organic matter) to reduce available nitrate to N₂ rather than to N₂O.

Response: Thank you for your kind suggestion. We agree and we have revised the sentence accordingly. We pasted as below for convenience (also see Revised Manuscript Lines 486–489).

“Sulfide oxidation could not only rescue the sulfide-sensitive NosZ [Cu:S] active sites but may simultaneously provide an additional electron donor beyond organic matter to reduce available nitrate to dinitrogen gas rather than to N₂O.” (Line 486–489)

38. Lines 356-363: I agree with this statement, but it should be pointed out that chemolithoautotrophs have the same advantage (as your work also demonstrates in Fig. 5g). In other words, this feature is not unique to chemolithoheterotrophs.

Response: Done as suggested (see Revised Manuscript Lines 491–493).

39. Lines 363-365: *As I stipulated above, this is complicated and not so clear-cut. The environment where *Arcobacter peruensis* was identified, was from a sulfidic event off the coast of Peru (Callbeck et al., 2019). Here, the environment was defined as being sulfide-, N₂O-, and organic matter-rich. Hence, the conditions where we find chemolithoheterotrophs are also environments where relatively high N₂O concentrations persist. To better frame this one could say that chemolithoheterotrophs/chemolithoautotrophs might contribute to suppressing some N₂O in sulfidic environments that are associated with high N₂O production rates.*

Response: We agree that the relationship between F-SOHD activity and N₂O emissions is complex and likely depends on specific environmental conditions, such as sulfidic events and N₂O-rich environments. In cases like the sulfidic event off the coast of Peru (Callbeck et al., 2019), where *Arcobacter peruensis* was identified, high levels of N₂O coexist with sulfide and organic matter, providing conditions conducive to both chemolithoheterotrophic and chemolithoautotrophic activity. Given these dynamics, we have rephrased our conclusion to emphasize that F-SOHDs could play a role in mitigating N₂O emissions, particularly in certain sulfidic environments, rather than as a general mechanism across all estuarine systems. This aligns with the idea that chemolithoheterotrophs and chemolithoautotrophs may help suppress N₂O in environments where both high N₂O production rates and sulfide stress coexist. Since chemolithoautotrophs are not the primary focus of our study, we have not specifically mentioned their contribution to N₂O reduction. The revised sentence is pasted below for convenience (also see Revised Manuscript Lines 496–499).

“Thus, our results suggest that F-SOHDs may contribute to the reduction of N₂O emissions under certain sulfidic conditions that are associated with high N₂O production rates, although their role might be context-dependent and potentially overlooked in estuarine systems.” (Lines 496–499)

40. Lines 368-370: *I think the introduction, results, and discussion would be improved if the authors made a greater attempt to draw parallels between the microbes identified here, and the recently described *Arcobacter peruensis* as well as the other ones mentioned earlier. I found it somewhat unfortunate that the authors when discussing organic matter and sulfide-rich environments, failed to mention *Arcobacter peruensis* in that context, but rather discussed chemolithoautotrophs like SUP05.*

Response: Thank you for your comment. Per the reviewer’s insightful suggestion, we have expanded the Introduction and Discussion sections to include more detailed descriptions of *Arcobacter peruensis* and other relevant species discussed earlier (see Revised Manuscript Lines 86–96 and 423–441).

41. Lines 370-375: *Could the authors be a little more specific? For example, what pathways did they identify in these microbial genomes, that made them comparable to the chemolithoheterotrophs described here?*

Response: Done as suggested. These genomes were identified as heterotrophs, possessing genes for complete denitrification, OM mineralization and sulfide/sulfur/thiosulfate oxidation (see Revised Manuscript Lines 509–512).

42. Lines 375-378: *The authors did not touch on what features make chemolithoheterotrophs more adaptable over chemolithoautotrophs, so can they be more specific? Moreover, it appears that the authors have mixed up the Gibbs free energy values from their figure. According to the authors Figure 6, the authors have calculated a Gibbs free energy of -494 and -579 for F-SOHDs related processes (rxns. C1 and C2), and with chemolithoautotrophy estimated at -566 (rxn. b). But in the text, they list the Gibbs free energy of heterotrophy instead of chemolithoautotrophy. This mix-up makes it seem that F-SOHDs activity is more energetically favorable when in reality it's not according to Figure 6.*

*With that said the authors have overlooked an important feature that gives chemolithoheterotrophs an advantage over chemolithoautotrophs. As detailed by Callbeck et al., in 2019, the assimilation of acetate into biomass is energetically more favorable than the assimilation of CO₂ via a chemolithoautotrophic metabolism. This is because CO₂ fixation pathways require substantial amounts of ATP (Erb 2011). The favorable energetics of a chemolithoheterotrophic metabolism explain how *Arcobacter* can rapidly bloom in the sulfide- and organic-rich upper shelf waters off the coast of Peru and Namibia (Lavik et al. 2009; Callbeck et al. 2019, 2021). This is a theme the authors could consider exploring in the context of the microbes identified in their study.*

Response: We sincerely thank the reviewer for your insightful comments and suggestions, which have greatly helped to refine and clarify our manuscript. We understand the reviewer has three main concerns: 1) Adaptability of F-SOHDs; 2) Correction of Gibbs Free Energy Values; 3) ATP Energetics and Carbon Assimilation Pathways.

1) Adaptability of F-SOHDs

Regarding this point, we believe that F-SOHDs may be more adaptable than chemolithoautotrophs due to their dual capacity to thrive in environments with or without sulfide and their ability to metabolize under both OM-rich and OM-limited conditions. This adaptability enables F-SOHDs to respond rapidly to fluctuating environmental conditions. Per the reviewer's suggestion, we have added this more specific details in the Revised Manuscript Lines 514–516.

2) Correction of Gibbs free energy values

We sincerely apologize for the error in reporting Gibbs free energy values in the previous version of the manuscript. Upon reviewing the calculations and incorporating the reviewer's feedback (see Comment 69 below), we found that the Gibbs free energy values for the four reactions were incorrectly listed. We have now recalculated the standard Gibbs free energy changes for each reaction, and the corrected values are as follows (Revised Figure 6):

- Reaction a: $\Delta G^\circ = -565.8 \text{ kJ mol}^{-1}$
- Reaction b: $\Delta G^\circ = -480.1 \text{ kJ mol}^{-1}$
- Reaction c1: $\Delta G^\circ = -579.3 \text{ kJ mol}^{-1}$

- Reaction c2: $\Delta G^\circ = -522.9 \text{ kJ mol}^{-1}$

These results show that F-SOHD-associated reactions (c1 and c2) exhibit a lower Gibbs free energy change compared to chemolithoautotrophic denitrification (b), indicating a thermodynamic advantage. We have revised **Figure 6** accordingly and corrected the text regarding the Gibbs free energy values (see Revised Manuscript Lines 517–519).

3) ATP energetics and carbon assimilation pathways

We greatly appreciate the reviewer for highlighting the importance of ATP energetics between different carbon assimilation pathways. We fully agree that the assimilation of acetate into biomass by chemolithoheterotrophs is energetically more favorable than the ATP-intensive CO_2 fixation pathways used by chemolithoautotrophs. This provides a compelling explanation for the metabolic advantages of F-SOHDs, such as their ability to proliferate rapidly in sulfide- and organic-rich environments, similar to those described by Callbeck et al. (2019, 2021) off the coasts of Peru and Namibia. Per the reviewer's suggestion, we have incorporated this discussion into the Revised Manuscript and cited the relevant references to strengthen the context (see Revised Manuscript Lines 519–522).

Revised Figure 6. Proposed ecophysiological advantages and environmental adaptations of F-SOHDs.

43. Lines 378-381: Their exceptional performance hinges on their ability to rapidly bloom over chemolithoautotrophs for available sulfide, this is something to keep in mind. Overall, I find that the authors could be a bit more specific regarding their industrial application reference. Do the authors mean to use chemolithoheterotrophs to remove N₂O in sulfide, and organic matter-rich engineered systems?

Response: We believe that F-SOHD microorganisms hold significant potential for industrial application in municipal wastewater treatment plants (WWTPs), particularly within anoxic, OM-rich denitrification tanks in engineered systems. Toxic sulfide, commonly around 10 mg L^{-1} , is frequently generated during sewage transport and inevitably introduced into the core anoxic biological treatment tanks (Sharma et al. 2008; Liang et al. 2019; Schönharting et al. 1998). These tanks are generally dominated by heterotrophic denitrifiers that lack sulfide oxidation capacity, leading to increased N_2O emissions and reduced nitrogen-removal efficiency (Pan et al. 2013; Schönharting et al. 1998). Our findings suggest that F-SOHDs, as heterotrophic denitrifiers capable of reducing N_2O , could be effectively utilized through the engineered bioaugmentation. By introducing F-SOHDs into anoxic denitrification tanks of WWTPs, we believe it is possible to enhance the microbial resilience to sulfide toxicity, maintain high efficiency in nitrogen and organic matter removal, and significantly mitigate N_2O emissions.

Following the suggestion, we have added additional details on the specific industrial applications to clarify this point, along with relevant citations (see Revised Manuscript Lines 525–530).

44. Lines 381-383: Fully agree, and I think this is where you could end the discussion. I find the next line regarding climate change and its evolutionary trajectories highly speculative and rather vague.

Response: Thank you for your positive feedback. As suggested, we have removed the following sentences on climate change and its evolutionary trajectories (see Revised Manuscript Line 532–535).

Methods:

45. Line 389: What month specifically did sampling take place? What was the water depth of the study site?

Response: Sediment samples from the study site were collected in August 2019, at a depth of 3.7 m. We have incorporated the information into the sentence accordingly (see Revised Manuscript Lines 539 and 543–544).

46. Lines 390-392: Do the authors have a reference to support its biogeochemical significance to the nitrogen cycle?

Response: We acknowledge that the sentence overstated its global effect, and we therefore deleted it in the Revised Manuscript Lines 540–543.

47. Line 393: Why did the authors pick the depth between 0-20 cm? What is the nitrate and oxygen penetration depth?

Response: In this study, we selected a sampling depth of 0–20 cm as it is a commonly used depth in similar research (Wang et al. 2024; Laing et al. 2009) which can effectively capture the main microbial processes occurring in surface sediments. During samplings, we measured the oxygen

penetration depth, which was approximately 2.04 cm. However, we were unable to obtain the nitrate penetration depth due to the lack of an available microelectrode in our lab.

48. Line 394: *What is sulfide antioxidant buffer, please be specific.*

Response: The sulfide antioxidant buffer is a well-established chemical reagent developed by Kellerlehmman and his colleagues (Kellerlehmman et al. 2006), used to prevent sulfide oxidation during short-term storage. As suggested, this reference has been cited in the Revised Manuscript Line 546.

49. Lines 396-398: *I would convert from mg L-1 to uM/mM to be consistent with the rest of the manuscript. Also, how were these nutrient concentrations measured? And by which methods? What were the detection limits?*

Response: Per the reviewer's suggestion, we have adjusted the units accordingly and included additional details regarding the measurement of physicochemical parameters and detection limits (see Revised Manuscript Lines 548–556).

50. Lines 404-407: *What were the incubation temperatures? And do they differ from in situ temperatures?*

Response: The sediment microcosms were incubated at 22 °C, which closely matches the in situ measured temperature, ranging from 21.2–23.6 °C. This information has been added in Revised Manuscript Line 563.

51. Line 420: *How was dinitrogen gas measured?*

Response: Dinitrogen gas in the headspace was measured using an Agilent 7890A gas chromatography (USA) equipped with thermal conductivity detector and J&W G3591-81121 packed column, with a detection limit of 2.6 ppm. As suggested, we have added this information in Revised Manuscript Lines 769–772.

52. Lines 424: *How much sediment was used for the extraction? Keep in mind that the readers need to be able to repeat the experiments.*

Response: Approximately 0.25 g sediment was used for the DNA extraction in our study (see Revised Manuscript Line 582).

53. Lines 424-428: *The authors provide no mention of references either in the qPCR section or supplementary Table 2 regarding the primers. Surely, these primers were designed elsewhere and if so, the authors need to indicate this information.*

Response: Per the reviewer's suggestion, relevant references for the primers has been provided in the Revised Supplementary Table 2 below (also see Revised Supplementary Information).

Revised supplementary Table 2. Primers and references of the eleven genes selected for qPCR

quantification.

Target gene	Primer	Primer sequence (5'-3')	References
dsrA	DSR-1F+	ACSCACTGGAAGCACGCCGG	(Kondo et al. 2004)
	DSR-R	GTGGMRCCTGTCAKRTTGG	
sqr	Forward	GCTCGGCAGCCTCAATAC	(Yin et al. 2014)
	Reverse	GGTCGGACGGTGGTTACTG	
soxB	710F	ATCGGYCAGGCYTTYCCSTA	(Tourna et al. 2014)
	1184R	MAVGTGCCGTTGAARTTGC	
nrfA	nrfAF2aw	CARTGYCAYGTBGARTA	(Welsh et al. 2014)
	nrfAR1	TWNGGCATRTGRCARTC	
napA	V17m	TGGACVATGGGYTTYAAYC	(Bru et al. 2007)
	napA-4r	ACYTCRCGHGCVGTRCCRCA	
narG	narG-f	TCGCCSATYCCGGCSATGTC	(Bru et al. 2007)
	narG-r	GAGTTGTACCAGTCRGCSGAYTCSG	
nirS	nirS1F	TACCACCCSGARCCGCGCGT	(Braker et al. 1998)
	nirS3R	GCCGCCGTCRTGVAGGAA	
nirK	876F	ATYGGCGGVCAYGCGCA	(Henry et al. 2004)
	1040R	GCCTCGATCAGRTRTGGTT	
norB	pF	CATGGCGCTGATAACGGG	(Dandie et al. 2007)
	pR	CTTIACCATGCTGAAGGCG	
nosZ	Z2F	CGCRACGGCAASAAGGTSMSST	(Henry et al. 2006)
	Z2R	CAKRTGCAKSGCRTGGCAGAA	
16S	341F	CCTACGGGNGGCWGCAG	(Herlemann et al. 2011)
rRNA	805F	GACTACHVGGGTATCTAATCC	

54. Lines 428-429: Is this common to do in qPCR, because cells often have more than one gene copy? So, the cell numbers are going to be overestimated?

Response: Yes, it is common to use this approach in qPCR and is widely applied in many DNA-SIP studies (Wasmund et al., 2021; He et al., 2022; Thomas et al., 2019; Yin et al., 2019). The analysis of the 16S rRNA gene across DNA buoyant density fractions helps to determine microbial distribution and the extent of ¹³C incorporation. While microbial cells can have multiple 16S rRNA gene copies, potentially leading to overestimation, this error impacts all density fractions uniformly. Therefore, the method remains reliable for comparing shifts in microbial communities along isotope gradients under different experimental conditions.

55. Lines 430-432: What qPCR chemistry was used, and how much DNA template was used? Were DNA concentrations also verified, and how? For the standards what concentration range was used in genes per μ L? Was melt curve analysis also done?

Response: All qPCR assays were performed in a 20 μ L reaction, containing 10 μ L of TB-Green

Premix Ex Taq (Takara, Japan), 0.4 μL of ROX Reference Dye, 0.4 μL of each forward/reverse primer (10 μM), 2 μL of standardized DNA template, and 6.8 μL of DNAase-free water. DNA concentrations were measured using the Quant-iT PicoGreen dsDNA kit (Thermo Scientific, USA), and standardized to 17.5 ng μL^{-1} for each DNA template. A melt curve analysis was conducted for each qPCR experiment. As suggested, these details have been added in Revised Manuscript Lines 588–596.

56. Lines 434-445: *Is this a completely novel technique or is there other studies that have done the same?*

Response: The washing process conducted inside the anaerobic workstation is a customized sediment pre-treatment method in our lab. This approach effectively remove undesirable impurities and sand from sediments, thereby preserving the biomass-rich portions for subsequent microcosm experiments. To support this procedure, we have cited a recent published work (see Revised Manuscript Line 600). Regarding the construction of DNA-SIP-based microcosms, this technique is relatively common and aligns with approaches used in other studies (Dunford and Neufeld, 2010; Sathyamoorthy et al. 2018; Thomas et al., 2019).

57. Lines 443-445: *What was the ^{13}C labeling percent for the incubation experiments?*

Response: The percents of $^{13}\text{C}_i$ and $^{13}\text{C}_o$ labeling in the incubation experiments were 98–99% and 99%, respectively. This information has been added in the Revised Manuscript Lines 608–609.

58. Line 488: *Change “compacity” to “capacity”*

Response: Revised as suggested (see Revised Manuscript Line 653).

59. Lines 495-497: *Please provide a reference to the study that designed these primers. And also indicate if these are universal primers or primers used to only amplify the Bacterial community.*

Response: As suggested, we have provided the reference (Herlemann et al. 2011) for the primers specifically designed for amplifying bacterial communities (see Revised Manuscript Lines 660–662).

60. Lines 497-500: *Can the authors indicate the number of raw reads generated?*

Response: Done as suggested (see Revised Manuscript Lines 663–664).

61. Lines 504-506: *Please provide a reference to the study that designed these primers.*

Response: Done as suggested (Singer et al. 2016; see Revised Manuscript Line 672).

62. Lines 514-515: *Is there a reference for this line?*

Response: Done as suggested (Grob et al. 2015; see Revised Manuscript Line 681).

63. Lines 551-552: *Perhaps I missed this somewhere, but what is the composition of the medium in*

terms of salts and other nutrients. Or is the medium just filtered river water with amended substrates? Please clarify.

Response: In addition to the target substrates, such as inorganic carbon (CO₂ and bicarbonate), composite OM (glucose, propionate, acetate, pyruvate, and fermentative DL-lactate), nitrate, and sulfide, the medium also contains essential salts and nutrients like ammonium (NH₄Cl), sulfate (Na₂SO₄), disodium phosphate, HEPES buffer, and trace elements. The medium composition has already been described in the “Isotope incubation experiments” section in Methods (see Revised Manuscript Lines 603–607). While filtered river water was used for *in-situ* sediment incubation and DNA-SIP microcosm experiments, sterile deionized water was employed in the long-term enrichment experiments to maintain greater control and prevent impurities from interfering with the growth of F-SOHDs. Additional details on the medium composition have been added for clarification (see Revised Manuscript Lines 721–725).

64. Lines 555-558: If CO₂ is removed, how is the system then pH buffered? Based on the principle that chemolithoautotrophs are slower growing compared to chemolithoheterotrophs I don't fully understand the reasoning behind removing CO₂ to hinder chemolithoautotrophs. As we know, chemolithoheterotrophs when supplied sulfide, nitrate, and organic matter will naturally outcompete chemolithoautotrophs for available sulfide (Callbeck et al., 2019; 2021). So, actively removing CO₂ could have downstream consequences for pH buffering in the system. Do the authors have pH measurements?

Response: Thank you for your question. In our experiments, we aimed to enrich the highly pure F-SOHDs, which have been confirmed to exhibit an obligate heterotrophic lifestyle through our preliminary experiments. While it is understood that chemolithoheterotrophs like F-SOHDs naturally outcompete chemolithoautotrophs in the presence of organic matter, chemolithoautotrophs may still persist by utilizing CO₂ produced by chemolithoheterotrophs, albeit at a slower rate. To prevent this, we actively removed CO₂ to suppress the growth of chemolithoautotrophs and enhance the purity of F-SOHDs.

To stabilize the pH after CO₂ removal, we employed HEPES buffer, which is effective in the neutral pH range (6.8 to 8.2) and commonly applied in microbial cultivation (Kraft et al. 2014), instead of carbonate/bicarbonate buffering. Throughout the enrichment process, pH was consistently monitored and remained stable at 7.10 ± 0.17.

65. Line 580: This could be shortened to read, “Physiological experiments”.

Response: Done as suggested (see Revised Manuscript Line 750).

66. Lines 583-585: Do the authors know which heterotrophic denitrifiers were enriched? And do they lack a capacity for sulfide oxidation and have the nosZ gene? I'm wondering how distinct they are from the sulfide-oxidizing heterotrophs.

Response: We previously performed 16S rRNA gene amplicon sequencing to examine the microbial composition of HD-enriched community, which revealed that it is predominantly composed of the

genus *Pseudomonas* (97.9% in relative abundance) (Figure R2 below). To assess the metabolic potential for sulfur oxidation and denitrification, we conducted PCR amplification targeting the key genes *sqr* and *nosZ*. Electrophoresis results showed a broad and bright band for the N₂O-reducing *nosZ* gene, while no amplification was observed for the sulfide-oxidizing *sqr* gene (Figure R3 below). This suggests that the HD community comprises non-sulfur-oxidizing conventional heterotrophic denitrifiers capable of reducing nitrate to dinitrogen gas but lacking the capacity to detoxify sulfide. This finding is further corroborated by the results of reaction kinetic experiments (Figure R4a–d below). Moreover, earlier studies have demonstrated that *Pseudomonas fluorescens* and *Pseudomonas aeruginosa* are similarly inhibited by sulfide, leading to increased N₂O emissions (Sørensen et al. 1980; Tam et al. 1979). Taken together, our findings from microbial composition, PCR amplification, and reaction kinetics consistently indicate the presence of non-sulfur-oxidizing heterotrophic denitrifiers in the HD community.

Figure R2. Amplicon sequencing of 16S rRNA gene of the enriched HD microbial community. Only the genera with a relative abundance higher than 0.05% were shown.

Figure R3. Ultraviolet imaging of PCR products after electrophoresis for *nosZ* (484 bp) and *sqr* (136 bp) genes.

Figure R4. Reaction kinetics of the HD microbial community under sulfide-free (a, b) and sulfide-present conditions (c, d), respectively.

67. Lines 595-596: The authors should provide more information regarding the instruments and standards used for gas analysis.

Response: Done as suggested (see Revised Manuscript Lines 766–772).

68. Lines 608-611: This is too vague. Please provide citations and additional information on how these sulfur species were quantified.

Response: We have incorporated additional information regarding sulfur determination as suggested, and relevant references have been cited accordingly (see Revised Manuscript Lines 786–791).

Figures:

69. Figure 6: How was Gibbs free energy calculated and where did they get the formation energies from? Under what conditions was this calculated, for example, pH, sulfide, nitrate concentrations? Reaction c2 is driven by mostly sulfide (1:5 ratio), with a little bit of organic matter, which seems more like a chemolithoautotrophy with a little bit of heterotrophy. What does this look like if the stoichiometry is a bit more balanced?

Response: The standard Gibbs free energy change (ΔG^0) for each reaction was calculated using the standard method, which involves subtracting the total formation free energies of the reactants from the total formation free energies of the products. The formation energies for each substance were obtained from the standard thermodynamic database provided in “Section 5 Thermochemistry, Electrochemistry, and Solution Chemistry” of the *CRC Handbook of Chemistry and Physics* (William M. Haynes ed. 2014). These calculations were performed under standard conditions, which

typically assume a temperature of 298 K, pressure of 1 atm, and reactant/product concentrations of 1 M. Factors such as pH, sulfide, and nitrate concentrations were not explicitly considered in the calculation, as the formation energies used for the calculation of reaction's ΔG^0 are independent of these variables. To improve clarity for readers, we have added an explanation of the calculation methods in Revised Manuscript Lines 1223–1227.

Per the reviewer's suggestion, we have adjusted the stoichiometry of organic matter and sulfide in the reaction c2 to achieve a more balanced representation and recalculated the corresponding ΔG^0 . The updated results show that the revised reaction c2 has a more negative Gibbs energy change ($\Delta G^0(\text{c2}) = -522.9 \text{ kJ mol}^{-1}$) compared to that of chemolithoautotrophic denitrifiers ($\Delta G^0(\text{b}) = -480.1 \text{ kJ mol}^{-1}$). This indicates a thermodynamic advantage in reaction free energy for F-SOHDs, supporting their metabolic potential in this system (Fig. 6).

Reaction c2:

Literature cited in my review:

Arevalo-Martinez, D. L., A. Kock, C. R. Loscher, R. A. Schmitz, and H. W. Bange. 2015. Massive nitrous oxide emissions from the tropical South Pacific Ocean. *Nat. Geosci* 8: 530–533. doi:10.1038/ngeo2469

Arévalo-Martínez, D. L., T. Steinhoff, P. Brandt, A. Körtzinger, T. Lamont, G. Rehder, and H. W. Bange. 2019. N₂O Emissions From the Northern Benguela Upwelling System. *Geophys. Res. Lett.* 46: 3317–3326. doi:10.1029/2018GL081648

Boltyanskaya, Y. V, V. V Kevbrin, A. M. Lysenko, T. V Kolganova, T. P. Tourova, G. A. Osipov, and T. N. Zhilina. 2007. *Halomonas mongoliensis* sp. nov. and *Halomonas kenyensis* sp. nov., new haloalkaliphilic denitrifiers capable of N₂O reduction, isolated from soda lakes. *Microbiology* 76: 739–747. doi:10.1134/S0026261707060148

Bourbonnais, A., R. T. Letscher, H. W. Bange, V. Échevin, J. Larkum, J. Mohn, N. Yoshida, and M. A. Altabet. 2017. N₂O production and consumption from stable isotopic and concentration data in the Peruvian coastal upwelling system. *Global Biogeochem. Cycles* 31: 678–698. doi:10.1002/2016GB005567

Callbeck, C. M., D. E. Canfield, M. M. M. Kuypers, P. Yilmaz, G. Lavik, B. Thamdrup, C. J. Schubert, and L. A. Bristow. 2021. Sulfur cycling in oceanic oxygen minimum zones. *Limnol. Oceanogr.* 66: 2360–2392. doi:https://doi.org/10.1002/lno.11759

Callbeck, C. M., G. Lavik, T. G. Ferdelman, and others. 2018. Oxygen minimum zone cryptic sulfur cycling sustained by offshore transport of key sulfur oxidizing bacteria. *Nat. Commun.* 9: 1729. doi:10.1038/s41467-018-04041-x

Callbeck, C. M., C. Pelzer, G. Lavik, and others. 2019. *Arcobacter peruensis* sp. nov., a chemolithoheterotroph isolated from sulfide and organic rich coastal waters off Peru. *Appl. Environ. Microbiol.* 85: e01344-19. doi:10.1128/AEM.01344-19

- Erb, T. J. 2011. *Carboxylases in Natural and Synthetic Microbial Pathways*. *Appl Env. Microbiol* 77: 8466–8477. doi:10.1128/aem.05702-11
- Felgate, H., G. Giannopoulos, M. J. Sullivan, A. J. Gates, T. A. Clarke, E. Baggs, G. Rowley, and D. J. Richardson. 2012. *The impact of copper, nitrate and carbon status on the emission of nitrous oxide by two species of bacteria with biochemically distinct denitrification pathways*. *Environ. Microbiol.* 14: 1788–1800. doi:10.1111/j.1462-2920.2012.02789.x
- Granger, J., and B. B. Ward. 2003. *Accumulation of nitrogen oxides in copper-limited cultures of denitrifying bacteria*. *Limnol. Oceanogr.* 48: 313–318. doi:10.4319/lo.2003.48.1.0313
- Kock, A., D. L. Arévalo-Martínez, C. R. Löscher, and H. W. Bange. 2016. *Extreme N₂O accumulation in the coastal oxygen minimum zone off Peru*. *Biogeosciences* 13: 827–840. doi:10.5194/bg-13-827-2016
- Lavik, G., T. Stührmann, V. Brüchert, and others. 2009. *Detoxification of sulphidic African shelf waters by blooming chemolithotrophs*. *Nature* 457: 581–584. doi:10.1038/nature07588
- Li, P., W. Yuan, Y. Huang, C. Zhang, C. Ni, Q. Lin, Z. Zhu, and J. Wang. 2022. *Complete genome sequence of Pseudomonas stutzeri S116 owning bifunctional catalysis provides insights into affecting performance of microbial fuel cells*. *BMC Microbiol.* 22: 137. doi:10.1186/s12866-022-02552-8
- Manconi, I., P. van der Maas, and P. Lens. 2006. *Effect of copper dosing on sulfide inhibited reduction of nitric and nitrous oxide*. *Nitric oxide Biol. Chem.* 15: 400–407. doi:10.1016/j.niox.2006.04.262
- Moffett, J. W., C. B. Tuit, and B. B. Ward. 2012. *Chelator-induced inhibition of copper metalloenzymes in denitrifying bacteria*. *Limnol. Oceanogr.* 57: 272–280. doi:10.4319/lo.2012.57.1.0272
- Naqvi, S. A. W., H. Naik, D. A. Jayakumar, M. S. Shailaja, and P. V Narvekar. 2006. *Seasonal oxygen deficiency over the western continental shelf of India*, p. 195–224. In N.L. Neretin [ed.], *Past and Present Water Column Anoxia*. Springer Netherlands.
- Naqvi, S. W. A., D. A. Jayakumar, P. V Narvekar, H. Naik, V. V. S. S. Sarma, W. D'Souza, S. Joseph, and M. D. George. 2000. *Increased marine production of N₂O due to intensifying anoxia on the Indian continental shelf*. *Nature* 408: 346–349. doi:10.1038/35042551
- Sorokin, D. Y. 2003. *Oxidation of Inorganic Sulfur Compounds by Obligately Organotrophic Bacteria*. *Microbiology* 72: 641–653. doi:10.1023/B:MICI.0000008363.24128.e5
- Walsh, D. A., E. Zaikova, C. G. Howes, Y. C. Song, J. J. Wright, S. G. Tringe, P. D. Tortell, and S. J. Hallam. 2009. *Metagenome of a versatile chemolithoautotroph from expanding oceanic dead zones*. *Science* (80-.). 326: 578–582. doi:10.1126/science.1175309
- Wang, L., and Z. Shao. 2021. *Aerobic Denitrification and Heterotrophic Sulfur Oxidation in the Genus Halomonas Revealed by Six Novel Species Characterizations and Genome-Based Analysis*. *Front. Microbiol.* 12: 652766. doi:10.3389/fmicb.2021.652766
- Wong, B.-T., and D.-J. Lee. 2014. *Pseudomonas yangmingensis sp. nov., an alkaliphilic denitrifying species isolated from a hot spring*. *J. Biosci. Bioeng.* 117: 71–74. doi:https://doi.org/10.1016/j.jbiosc.2013.06.006

Responses to Reviewer #3

Reviewer #3 (Remarks to the Author):

This study identifies a previously unknown group of heterotrophic denitrifiers capable of coupling sulfur oxidation with denitrification autonomously. Traditionally, heterotrophic denitrifiers have been thought to be hampered by sulfide, which negatively affects their growth and increases nitrous oxide (N₂O) emissions, a potent greenhouse gas. However, in this study, these bacteria were found to use sulfide as an alternative electron source, enhancing denitrification and reducing nitrous oxide (N₂O) emissions. Through microcosm incubations, DNA stable-isotope probing, and metagenomics, the researchers demonstrated that these versatile heterotrophs exhibit a flexible physiology allowing them to thrive in sulfide-rich environments and compete effectively with chemolithoautotrophic denitrifiers. This discovery shifts our understanding of sulfur cycling and suggests new implications for mitigating climate change.

Response: We sincerely appreciate the reviewer's positive and encouraging feedback on our work! We have revised the manuscript by fully considering and addressing all the comments as suggested. Please see our detailed responses below.

Specific Comments

1. *There are a few grammatical errors and some sentences are lengthy and complex, which might hinder comprehension. Simplify and break down long sentences where possible.*

Response: Thank you for your comment. Following the reviewer's subsequent suggestions, we are committed to making the manuscript clear and digestible to the audience. Additionally, we have done our best to simplify long and complicated sentences throughout the manuscript. Please see Revised Manuscript Lines 69–72, 73–76, 80–82, 102–108, 202, 211, 232, 234–237, 279–283, 302–304, 346, 394–400, 442–447, 462, 495, 507, 509, 514–519, 568, 594–596, 653, and 1219–1223.

2. *Line 62-64: suggested revision: "Incomplete denitrification is a major source of N₂O, a potent greenhouse gas that contributes to climate warming. Additionally, N₂O emissions contribute to ozone layer depletion and acid rain formation."*

Response: Done as suggested. This sentence has been rephrased in response to the comments provided by Reviewers #2 and #3 (see Revised Manuscript Lines 69–72).

3. *Line 65-68: Please revise to "In many estuaries, coastal regions, and marine oxygen-deficient zones, the sulfur cycle plays a crucial role in regulating biogeochemical processes. This is due to the dominance of sulfate as a terminal electron acceptor and the subsequent release of hydrogen sulfide (H₂S) from benthic sediments."*

Response: Done as suggested (see Revised Manuscript Lines 73–76).

4. *Line 68-75: This part should include a more detailed introduction to sulfide-driven dissimilatory nitrate reduction to ammonium (DNRA) (Canfield et. al, 2010 (Science); Wang et. al, 2021 (Water Research), etc.).*

Response: Thank you for the valuable suggestion and references. We have revised as suggested (see Revised Manuscript Lines 82–85). The sentence is pasted below for convenience.

“Some of these chemolithoautotrophs are also involved in dissimilatory nitrate reduction to ammonium (DNRA), thereby supplying ammonium to support microbial growth of indigenous bacteria^{20,21}.” (Lines 82–85)

5. Line 71-73: Please revise to “Sulfide is highly toxic to heterotrophic denitrifiers, adversely affecting microbial activity and potentially increasing N₂O emissions.”

Response: Revised as suggested (see Revised Manuscript Lines 76–79). This sentence has been rephrased in response to the comments provided by Reviewers #2 and #3.

6. Line 76-80: “In this study, we investigate the response of heterotrophic denitrifiers to sulfide in the highly productive Songhua River Estuary, a major tributary within the largest water system in East Asia. We employed a combination of microcosm and lab culture experiments, metagenomics, and a novel community-isotope-corrected DNA stable-isotope probing (DNA-SIP) approach.”

Response: Revised as suggested (see Revised Manuscript Lines 102–108).

7. In sediment microcosms, how do community compositions shift over time under different treatment conditions? How is the stability and resilience of these microbial communities in response to fluctuating environmental factors?

Response: Thank you for your comment. Unfortunately, we do not have data on the dynamic shifts in community composition. Whether a time-series investigation is necessary depends on the scientific question. In our study, the primary goal was to identify the key contributors in response to sulfide stimulation under organic matter-rich conditions, which we addressed using DNA-SIP technique. The timeframe for DNA-SIP experiments is typically determined by the time required for effective isotope incorporation. Shorter durations risk incomplete labelling, while extended incubation can lead to cross-feeding, complicating accurate detection of isotope-assimilating microbes. Prior to the DNA-SIP-based incubation, our preliminary experiments showed that substrates were fully consumed within 65–83 hours in the +C+N+S, +C+N, and +C_i+N+S treatments. Based on this and literature, we chose a 10–12 day incubation (~ three generations), a widely accepted timeframe in previous DNA-SIP-based studies (Cai et. al, 2016; Zhang et. al, 2021; Wilhelm et. al, 2019; Dunford and Neufeld, 2010). This effectively labels most microorganisms, while avoiding cross-feeding. Although we did not track microbial community composition over time, we recognize that such data could provide valuable insights into how communities respond to fluctuating environmental factors. However, we feel this question is somewhat broad and may be beyond the scope of our current study.

8. How do the lab-based findings translate to natural estuarine environments? Are the conditions in the microcosm experiments reflective of those in natural settings, and how might environmental

variability influence the activity of F-SOHDs? Can modelling approaches be used to predict their contributions to nutrient cycling and greenhouse gas emissions under various environmental scenarios?

Response: We acknowledge the complexities in translating lab-based findings to natural estuarine environments. In our study, we aimed to simulate *in situ* conditions by using samples directly collected from estuarine environments for microcosm incubations. While this approach allows us to closely mimic natural settings, we recognize that natural estuarine systems are far more complex and variable compared to controlled lab environments. As a result, microcosm experiments may not fully capture the unpredictable dynamics of natural ecosystems.

Environmental variability, such as fluctuating oxygen levels, temperature, salinity, and nutrient availability, can significantly influence the activity of F-SOHDs *in situ*, which may differ from the stable conditions observed in microcosms. In future research, we plan to deploy an on-site chamber and conduct *in situ* incubations with various substrates provided to assess whether F-SOHDs are the main contributors in natural environments. During the incubation, we will leverage static respiration chamber measurements, qPCR, DNA-SIP based microcosm assays and metagenomic sequencing to examine nitrate reduction, organic matter consumption and sulfur oxidation, further exploring the potential coupling of carbon, nitrogen and sulfur cycling. To better understand these dynamics, future studies incorporating time-series field investigations could also provide valuable insights into how F-SOHDs adapt and respond over time in natural environments.

Regarding modeling approaches, they offer a potential tool to bridge the gap between lab-based experiments and real-world conditions. By integrating data from microcosm studies, time-series field observations, and environmental variables, models can help predict the contributions of F-SOHDs to nutrient cycling and greenhouse gas emissions under different environmental scenarios. These predictive models could improve our understanding of how microbial activity fluctuates across various estuarine conditions and inform broader ecosystem management strategies.

*9. Given the ability of F-SOHDs to reduce N₂O emissions, how can this be applied in wastewater treatment? What are the practical steps to harness these microbes in engineered systems to mitigate greenhouse gas? Can the activity of F-SOHDs be enhanced *in situ* through biostimulation strategies? What types of amendments (e.g., organic carbon, sulfide) would be most effective, and what are the potential ecological side effects of such interventions?*

Response: It is a very insightful comment. F-SOHDs offer significant potential for reducing N₂O emissions in wastewater treatment processes. In practical settings, particularly in long-distance sewage transport pipelines, sulfide can be generated biologically or chemically at concentrations of around 10 mg L⁻¹ (Sharma et. al, 2008; Liang et. al, 2019; Schönharting et. al, 1998). This sulfide often enters the anoxic denitrification tanks in wastewater treatment plants, where it can inhibit conventional heterotrophic denitrifiers, leading to increased N₂O emissions (Pan et al., 2013; Schönharting et al., 1998).

Our findings demonstrate that F-SOHDs, which are essentially heterotrophic denitrifiers with the additional ability to detoxify sulfide, can mitigate these emissions. Rather than biostimulation,

we believe bioaugmentation might be a more effective strategy for harnessing their potential in engineered systems. Introducing F-SOHDs into anoxic denitrification tanks could unlock their ability to reduce N₂O emissions while maintaining efficient nitrogen and COD removal. Furthermore, F-SOHDs could enhance the microbial community's resistance to sulfide toxicity, significantly reducing the overall greenhouse gas footprint.

In terms of amendments, moderate levels of sulfide could be beneficial, helping to regulate competition between F-SOHDs and conventional heterotrophic denitrifiers, thus ensuring the stable presence and activity of F-SOHDs during long-term operation. However, one potential ecological side effect is the increase in sulfate concentrations in the effluent due to the sulfur-oxidizing capabilities of F-SOHDs. Careful monitoring and adjustment of sulfide levels may be necessary to balance N₂O reduction with the prevention of excessive sulfate accumulation.

Further research into optimal amendment strategies and the long-term effects of F-SOHDs through bioaugmentation in wastewater systems would provide valuable insights into mitigating both environmental impact and operational challenges.

Following the suggestion, we have added additional details on the potential industrial applications of F-SOHDs in wastewater treatment to clarify this point (see Revised Manuscript Lines 525–530).

10. While metagenomics provides a snapshot of potential metabolic capabilities, it does not confirm actual activity. How can the findings from metagenomics be integrated with other omics approaches to provide a more comprehensive understanding of microbial functions in these environments?

Response: Thank you for your comment. We acknowledge that relying solely on metagenomics provides only a snapshot of microbial communities and their potential metabolic capabilities. To gain a more comprehensive understanding of microbial activity, incorporating other omics techniques, such as metatranscriptomics, metaproteomics, and metabolomics, would be beneficial. However, we want to emphasize that our study employed a combined approach of metagenomics and DNA-SIP. DNA-SIP directly identifies active microorganisms by tracking isotope incorporation into their DNA, ensuring that only metabolically active microbes are detected. By integrating DNA-SIP with metagenomics, we can not only identify potential functions but also confirm actual metabolic activity under specific conditions. This, together with various nutrient treatments and isotope labeling, allows us to accurately pinpoint key contributors to element cycling in these environments.

References

- Aelion, C.M., Wartinger, U. 2009. Low sulfide concentrations affect nitrate transformations in freshwater and saline coastal retention pond sediments. *Soil. Biol. Biochem.* 41 (4), 735-741.
- Caffrey, J. M., Bonaglia, S., Conley, D. J. 2018. Short exposure to oxygen and sulfide alter nitrification, denitrification, and DNRA activity in seasonally hypoxic estuarine sediments. *FEMS*

Microbiol. Lett. 366.

Cai, Y., Zheng, Y., Bodelier, P.L., Conrad, R., Jia, Z. 2016. Conventional methanotrophs are responsible for atmospheric methane oxidation in paddy soils. *Nat. Commun.* 7, 11728.

Canfield, D.E., Stewart, F.J., Thamdrup, B., De Brabandere, L., Dalsgaard, T., Delong, E.F., Revsbech, N.P., Ulloa, O. 2010. A cryptic sulfur cycle in oxygen-minimum-zone waters off the Chilean coast. *Science* 330 (6009), 1375-1378.

Dunford, E.A., Neufeld, J.D. 2010. DNA stable-isotope probing (DNA-SIP). *J Vis. Exp.* (42), e2027-e2027.

Durante Rodríguez, G., Fernández-Llamas, H., Alonso-Fernandes, E., Fernández-Muñiz, M. N., Muñoz-Olivas, R., Díaz, E., & Carmona, M. 2019. ArxA from *Azoarcus* sp. CIB, an anaerobic arsenite oxidase from an obligate heterotrophic and mesophilic bacterium. *Front. Microbiol.* 10, 1699.

Grob, C., Taubert, M., Howat, A.M., Burns, O.J., Dixon, J.L., Richnow, H.H., Jehmlich, N., von Bergen, M., Chen, Y., Murrell, J.C. 2015. Combining metagenomics with metaproteomics and stable isotope probing reveals metabolic pathways used by a naturally occurring marine methyloph. *Environ Microbiol.* 17 (10), 4007-4018.

He, R., Wang, J., Pohlman, J.W., Jia, Z., Chu, Y.-X., Wooller, M.J., Leigh, M.B. 2022. Metabolic flexibility of aerobic methanotrophs under anoxic conditions in Arctic lake sediments. *ISME J.* 16 (1), 78-90.

Herlemann, D.P.R., Labrenz, M., Jürgens, K., Bertilsson, S., Waniek, J.J., Andersson, A.F. 2011. Transitions in bacterial communities along the 2000 km salinity gradient of the Baltic Sea. *ISME J.* 5 (10), 1571-1579.

Hurek, T., Van Montagu, M., Kellenberger, E., & Reinhold Hurek, B. 1995. Induction of complex intracytoplasmic membranes related to nitrogen fixation in *Azoarcus* sp. BH72. *Mol. Microbiol.* 18(2), 225-236.

Kellerlehmann, B., Corrie, S., Ravn, R., Yuan, Z., Keller, J. 2006. Preservation and simultaneous analysis of relevant soluble sulfur species in sewage samples, Vienna, Austria

Kleinjan, W.E., Keizer, A.D., Janssen, A. 2005. Equilibrium of the reaction between dissolved sodium sulfide and biologically produced sulfur. *Colloids Surf B Biointerfaces* 43 (3-4), 228-237.

Kompantseva, E. I. et al. 2017. *Calorithrix insularis* gen. nov., sp. nov., a novel representative of the phylum Calditrachaeota. *International Journal of Systematic and Evolutionary Microbiology.* 67, 1486-1490.

Kraft, B., Tegetmeyer, H.E., Sharma, R., Klotz, M.G., Ferdelman, T.G., Hettich, R.L., Geelhoed, J.S., Strous, M. 2014. The environmental controls that govern the end product of bacterial nitrate respiration. *Science.* 345(6197), 676-679.

Laing, G.D., Meers, E., Dewispelaere, M., Vandecasteele, B., Rinklebe, J.R., Tack, F.M.G., Verloo,

- M.G. 2009. Heavy metal mobility in intertidal sediments of the Scheldt estuary. *Sci. Total. Environ* 407 (8), 2919-2930
- Liang, Zhen-Sheng, Liang Zhang, Di Wu, Guang-Hao Chen, and Feng Jiang. 2019. Systematic evaluation of a dynamic sewer process model for prediction of odor formation and mitigation in large-scale pressurized sewers in Hong Kong. *Water Res.* 154, 94-103.
- Mateo-Sagasta, J., Raschid-Sally, L. & Tebo, A. 2015. Global wastewater and sludge production, treatment and use. 15–38 (Springer)
- Miller, L.G., Oremland, R.S., Paulsen, S. 1986. Measurement of Nitrous Oxide Reductase Activity in Aquatic Sediments. *Appl. Environ. Microbiol.* 51 (1), 18-24.
- Miroshnichenko, M. L. et al. 2003. *Oceanithermus profundus* gen. nov., sp. nov., a thermophilic, microaerophilic, facultatively chemolithoheterotrophic bacterium from a deep-sea hydrothermal vent. *International Journal of Systematic and Evolutionary Microbiology.* 53, 747-752.
- Naqvi, S.W et al., 2000. Increased marine production of N₂O due to intensifying anoxia on the Indian continental shelf. *Nature* 408 (6810), 346-349.
- Pan, Y., Ye, L., Yuan, Z. 2013. Effect of H₂S on N₂O reduction and accumulation during denitrification by methanol utilizing denitrifiers. *Environ. Sci. Technol.* 47 (15), 8408-8415.
- Price, A., Pearson, V. K., Schwenzer, S. P., Miot, J. & Olsson-Francis, K. 2018. Nitrate-Dependent Iron Oxidation: A Potential Mars Metabolism. *Frontiers in Microbiology.* 9, 513.
- Sathyamoorthy, S., Hoar, C., Chandran, K. 2018. Identification of bisphenol A-assimilating microorganisms in mixed microbial communities using ¹³C-DNA stable isotope probing. *Environ Sci Technol* 52 (16), 9128-9135.
- Schönharting, Rehner, Metzger, Krauth, Rizzi. 1998. Release of nitrous oxide from denitrifying activated sludge caused by H₂S-containing wastewater: quantification and application of a new mathematical model. *Water. Sci. Technol.* 38 (1), 237-246.
- Senga, Y., Mochida, K., Fukumori, R., Okamoto, N., Seike, Y., 2006. N₂O accumulation in estuarine and coastal sediments: The influence of H₂S on dissimilatory nitrate reduction. *Estuar. Coast. Shelf. Sci.* 67 (1), 231-238.
- Shao, B et. al. 2022. Cryptic Sulfur and Oxygen Cycling Potentially Reduces N₂O-Driven Greenhouse Warming: Underlying Revision Need of the Nitrogen Cycle. *Environ. Sci. Technol.* 56 (9), 5960-5972.
- Sharma, K.R., Yuan, Z., Haas, D.D., Hamilton, G., Corrie, S., Keller, J. 2008. Dynamics and dynamic modelling of H₂S production in sewer systems. *Water Res.* 42, 2527-2538.
- Singer, E., Bushnell, B., Coleman-Derr, D., Bowman, B., Woyke, T. 2016. High-resolution phylogenetic microbial community profiling. *ISME J.* 10 (8), 2020-2032.
- Sun, W., Sun, X., Häggblom, M.M., Kolton, M., Lan, L., Li, B., Dong, Y., Xu, R., Li, F. 2021. Identification of antimonate reducing bacteria and their potential metabolic traits by the combination of stable isotope probing and pangenomic analysis. *Environ. Sci. Technol.* 55 (20), 13902-13912.

- Sørensen, J., Rasmussen, L.K., Koike, I. 1987. Micromolar sulfide concentrations alleviate acetylene blockage of nitrous oxide reduction by denitrifying *Pseudomonas fluorescens*. *Can. J. Microbiol.* 33 (11), 1001-1005.
- Sørensen, J., Tiedje, J.M., Firestone, R.B. 1980. Inhibition by sulfide of nitric and nitrous oxide reduction by denitrifying *Pseudomonas fluorescens*. *Appl. Environ. Microbiol.* 39 (1), 105-108.
- Tam, T.-Y., Knowles, R. 1979. Effects of sulfide and acetylene on nitrous oxide reduction by soil and by *Pseudomonas aeruginosa*. *Can. J. Microbiol.* 25 (10), 1133-1138.
- Thomas, F., Corre, E., Cebon, A. 2019. Stable isotope probing and metagenomics highlight the effect of plants on uncultured phenanthrene-degrading bacterial consortium in polluted soil. *ISME J.* 13(7), 1814-1830.
- Walter, W.G. 1998. APHA standard methods for the examination of water and wastewater. *Health Lab. Sci.* 4 (3), 137.
- Wang, S., Lan, B., Yu, L., Xiao, M., Jiang, L., Qin, Y., Jin, Y., Zhou, Y., Armanbek, G., Ma, J., Wang, M., Jetten, M.S.M., Tian, H., Zhu, G., Zhu, Y.-G. 2024. Ammonium-derived nitrous oxide is a global source in streams. *Nat. Commun.* 15 (1), 4085.
- Wasmund, K., Pelikan, C., Schintlmeister, A., Wagner, M., Watzka, M., Richter, A., Bhatnagar, S., Noel, A., Hubert, C.R.J., Rattei, T., Hofmann, T., Hausmann, B., Herbold, C.W., Loy, A. 2021. Genomic insights into diverse bacterial taxa that degrade extracellular DNA in marine sediments. *Nat. Microbiol.* 6(7), 885-898.
- Wilhelm, R.C., Singh, R., Eltis, L.D., Mohn, W.W. 2019. Bacterial contributions to delignification and lignocellulose degradation in forest soils with metagenomic and quantitative stable isotope probing. *ISME J* 13 (2), 413-429.
- William M. Haynes. (Ed.) 2014. *CRC Handbook of Chemistry and Physics (95th ed.)*. CRC Press. (<https://www.taylorfrancis.com/books/mono/10.1201/b17118>).
- Wu, L et. al. 2019. Global diversity and biogeography of bacterial communities in wastewater treatment plants. *Nat. Microbiol.* 4 (7), 1183-1195.
- Yin, X., Wu, W., Maeke, M., Richter-Heitmann, T., Kulkarni, A.C., Oni, O.E., Wendt, J., Elvert, M., Friedrich, M.W. 2019. CO₂ conversion to methane and biomass in obligate methylophilic methanogens in marine sediments. *ISME J.* 13 (8), 2107-2119.
- Zhang, M., Li, Z., Hggblom, M.M., Young, L., Sun, W. 2021. Bacteria responsible for nitrate-dependent antimonite oxidation in antimony-contaminated paddy soil revealed by the combination of DNA-SIP and metagenomics. *Soil. Biol. Biochem.* 156, 108194.